# Global protein turnover quantification in *Escherichia coli* reveals cytoplasmic recycling under nitrogen limitation

Meera Gupta [1,2,3,7], Alex N. T. Johnson[1,2,3,7], Edward R. Cruz[2,3], Eli J. Costa [2], Randi L. Guest[3], Sophia Hsin-Jung Li[3], Elizabeth M. Hart[3,4], Thao Nguyen [1,2,3], Michael Stadlmeier [2,3], Benjamin P. Bratton [2,3,5,6], Thomas J. Silhavy [3], Ned S. Wingreen [2,3], Zemer Gitai [3] & Martin Wühr [2,3] ✉

Protein turnover is critical for proteostasis, but turnover quantification is challenging, and even in well-studied *E. coli*, proteome-wide measurements remain scarce. Here, we quantify the turnover rates of ~3200 *E. coli* proteins under 13 conditions by combining heavy isotope labeling with complement reporter ion quantification and find that cytoplasmic proteins are recycled when nitrogen is limited. We use knockout experiments to assign substrates to the known cytoplasmic ATP-dependent proteases. Surprisingly, none of these proteases are responsible for the observed cytoplasmic protein degradation in nitrogen limitation, suggesting that a major proteolysis pathway in *E. coli* remains to be discovered. Lastly, we show that protein degradation rates are generally independent of cell division rates. Thus, we present broadly applicable technology for protein turnover measurements and provide a rich resource for protein half-lives and protease substrates in *E. coli*, complementary to genomics data, that will allow researchers to study the control of proteostasis.

Protein degradation is central to protein homeostasis (proteostasis) and is critical in most cellular pathways[1–3]. As environments change, modification of degradation rates can rapidly adapt protein abundances to desired levels. Even if protein levels are modulated via transcription or translation, the time it takes for a protein to reach its new steady state is set by its turnover rate[4,5]. Unsurprisingly, many signaling and transcriptional regulatory proteins exhibit short half-lives[6–9]. Protein degradation is important in health and disease, such as cancer and neurodegenerative disorders[10,11]. Additionally, protein degradation plays an important metabolic role. It has been shown that bacteria and yeast cells increase their proteome turnover rates under starvation conditions, presumably generating and recycling scarce amino acids[12–15].

Quantitative models have been developed to describe the dependence of global protein expression on cells' physiological characteristics, most notably cell doubling times. These models are probably the most well-developed for the model bacterium *Escherichia coli*[16,17]. The cell cycle time in *E. coli* varies from 20 minutes in rich media to the cessation of division under starvation. Transcription rates typically increase with cell division rates[18,19]. Knowing how global parameters scale with physiological cell states allows for remarkable quantitative predictions for gene expression changes across different growth conditions[16,17]. However, active protein degradation by proteolysis is typically ignored in these models. Instead, proteins are assumed to be completely stable and only diluted via cell growth and

[1]Department of Chemical and Biological Engineering, Princeton University, Princeton, NJ, USA. [2]Lewis-Sigler Institute for Integrative Genomics, Princeton University, Princeton, NJ, USA. [3]Department of Molecular Biology, Princeton University, Princeton, NJ, USA. [4]Department of Microbiology, Harvard Medical School, Boston, MA, USA. [5]Vanderbilt Institute of Infection, Immunology and Inflammation, Nashville, TN, USA. [6]Department of Pathology, Microbiology, and Immunology, Vanderbilt University Medical Center, Nashville, TN, USA. [7]These authors contributed equally: Meera Gupta, Alex N. T. Johnson. ✉e-mail: wuhr@princeton.edu

division. This simplification is likely due to a lack of reliable genome-wide degradation rate measurements under varying growth conditions. It is still unclear how active degradation rates scale with changing cell cycle times and how this affects global gene expression regulation. Knowledge of protein degradation rates and how they scale with the physiological characteristics of cells would improve predictive models of protein expression across various cell states.

Cells have developed sophisticated mechanisms to recognize and degrade specific proteins. While eukaryotes utilize the ubiquitin-proteasome pathway, in bacteria, selective proteolysis is executed by ATP-dependent proteases[1]. While many proteases can digest unfolded proteins and peptides, unfolding a protein for degradation requires energy. In *E. coli*, four ATP-dependent proteases are known: ClpP, Lon, HslV, and FtsH. Pulldown experiments with inactivated protease mutants or protein-array studies have allowed the proteome-wide identification of putative substrates[20–23]. Orthogonally, individual substrates have been assigned to the four proteases by measuring the degradation of individual proteins in protease knockout strains or via in vitro assays[24,25]. Several example proteins (e.g., RpoH, LpxC, and SoxS) have been shown to be degraded by multiple proteases, demonstrating remarkable redundancy[26–28]. But it is still unclear to what extent substrates overlap between different proteases.

In most biological systems, protein degradation is balanced by the synthesis of new protein, making measurements of degradation rates challenging. An easy way to overcome this complication is by using translational inhibitors like cycloheximide or chloramphenicol[29–31]. Assuming that the addition of the drug does not perturb the cells aside from blocking the translation of new proteins, protein degradation can be conveniently measured by assaying changes in protein abundances over time via western blots or quantitative proteomics. However, when we performed such experiments in *E. coli*, we found that many proteins whose abundances rapidly decreased were periplasmic (Supplementary Fig. 1). Further investigation revealed that these periplasmic proteins were not degraded but rather were accumulating in the bacterial growth medium (Supplementary Fig. 1). Presumably, this was due to protein leakage through the outer membrane. We concluded that translation inhibitor experiments in *E. coli* could lead to major perturbations, and, thus, interpreting such studies might be challenging.

A classic method to measure the unperturbed turnover of biological molecules uses radioactive isotope tracking or the combination of heavy isotope labeling and quantitative mass spectrometry[32,33]. Isotopic labels can be introduced with heavy nutrients (e.g., ammonium, glucose, or amino acids) or by incubation in heavy water. Most proteomic turnover studies have been performed with heavy amino acid labeling (dynamic stable isotope labeling by amino acids in cell culture (SILAC))[34,35], but the small number of labeled residues limits sensitivity for short-time SILAC labeling, and missing values can hinder the coverage of multiple time points in complex systems. A further advance has been the combination of SILAC experiments and isobaric tag labeling[36]. However, these measurements tend to suffer from the inherent ratio compression of multiplexed proteomics[37–39]. Heavy ammonium, glucose, and water are comparatively cheap but result in overly complex MS1 spectra, which are difficult to interpret, particularly for lower abundance proteins[40–42]. For a more detailed discussion of the advantages and limitations of various global protein turnover measurement techniques, please see the recent review by Ross et al., particularly Supplementary Table 1[43].

Despite the central role of protein degradation in nearly every aspect of biological regulation, reliable and large-scale measurements are still scarce. Even fewer studies have compared turnover rates between multiple conditions[44,45]. Here, we measure protein turnover in *E. coli* by combining heavy isotope labeling via [15]N ammonium with the accurate multiplexed proteomics method TMTproC[46]. We provide a rich resource of protein turnover rates for ~3.2k *E. coli* proteins (77% of

all genes in *E. coli*) measured across 13 different growth conditions with replicates. When comparing turnover rates among various nutrient limitations, we found that *E. coli* recycles its cytoplasmic proteins when nitrogen-limited, and we assign substrates to proteases by measuring the change of protein turnover in knockout strains. Lastly, we show that active turnover rates are typically independent of cell division rates.

## Results

### Combining heavy isotope labeling with complement reporter ion quantification enables high-quality protein turnover measurements

We wanted to measure protein turnover rates and evaluate how these rates vary across growth conditions. To simplify our measurements, we grow *E. coli* in chemostats, where we can control the cell doubling time and enforce steady state (Fig. 1A). After cells reach steady state, we change the inlet medium from unlabeled nutrients to [15]N-labeled ammonium. Over time, the [15]N-ammonium concentration in the reactor increases, and newly synthesized proteins incorporate more heavy isotopes. We can monitor the shift in the isotopic envelope of peptides by taking samples after the media switch using proteomics (Fig. 1B). With the knowledge of a peptide's chemical composition and the fraction of heavy isotopes over time, we can calculate the turnover rate of the corresponding protein. In practice, however, obtaining such measurements of isotopic envelopes in the MS1 spectrum is quite challenging, particularly at later time points when the isotopic envelopes spread out and overlap with those of other peptides. MS1-based label-free quantification relies on accurate assignment of peaks to peptide elution profiles, which is made more challenging by the complexity of spectra. Additionally, traditional search algorithms struggle to identify peptides with a significant fraction of heavy isotopes, so missing values at later time points are a severe limitation of such approaches[42]. To overcome these limitations, we labeled samples at each of the acquired eight-time points with TMTpro isobaric tags and combined them for co-injection into the mass spectrometer[46,47]. While MS1 spectra are still extremely complex using this approach, by combining isotope envelopes of peptides with and without [15]N, we ensure that the pseudo-monoisotopic peak ($M_0$) is always present for isolation and fragmentation in the MS2, increasing peptide identification rates and alleviating the missing value problem. Multiplexing experiments with highly complex MS1 spectra inevitably increase co-isolation of multiple peptides in a single MS2 spectra. Analyzing these extremely complex samples with standard low *m/z* reporter ion quantification would lead to severe ratio distortion and measurement artifacts[37,38,48]. We overcame this limitation by quantifying the balancer-peptide conjugates (complement reporter ions) that remain after reporter ions are cleaved off in the MS2 spectra. Complement ions have peptide-dependent *m/z* ratios that are typically slightly different than co-isolated peptides. Therefore, using the complementary ions for quantification reduces ratio distortion effects compared to both MS2 and multi-notch MS3 reporter ion quantification[46,47](Fig. 1C).

Figure 1D shows the peptide-level quantification for a stable protein, OmpF, and an unstable protein, RpoS, from carbon-limited chemostats with 6-h doubling times[24,49,50]. We built a differential equation model[51] for the dynamics of $M_0$ with a parameter of $k_D$ (active degradation rate) and the known variable $D$ (dilution rate) (Supplementary Note, Supplementary Data 8). For every protein, $k_D + D$ (total turnover rate) is obtained by fitting this model to the experimentally measured signal of peptides. For a protein that is not actively degrading ($k_D = 0$) and only diluting with $D$ set entirely by the chemostat, we can estimate the dynamics of $M_0$. This is called the dilution curve (dotted). Fitting $k_D$ to the measured signal of OmpF peptides yields $k_D + D \approx D$ which indicates that this protein is not actively degrading. In contrast, the deduced total half-life for RpoS is much shorter than the cell doubling

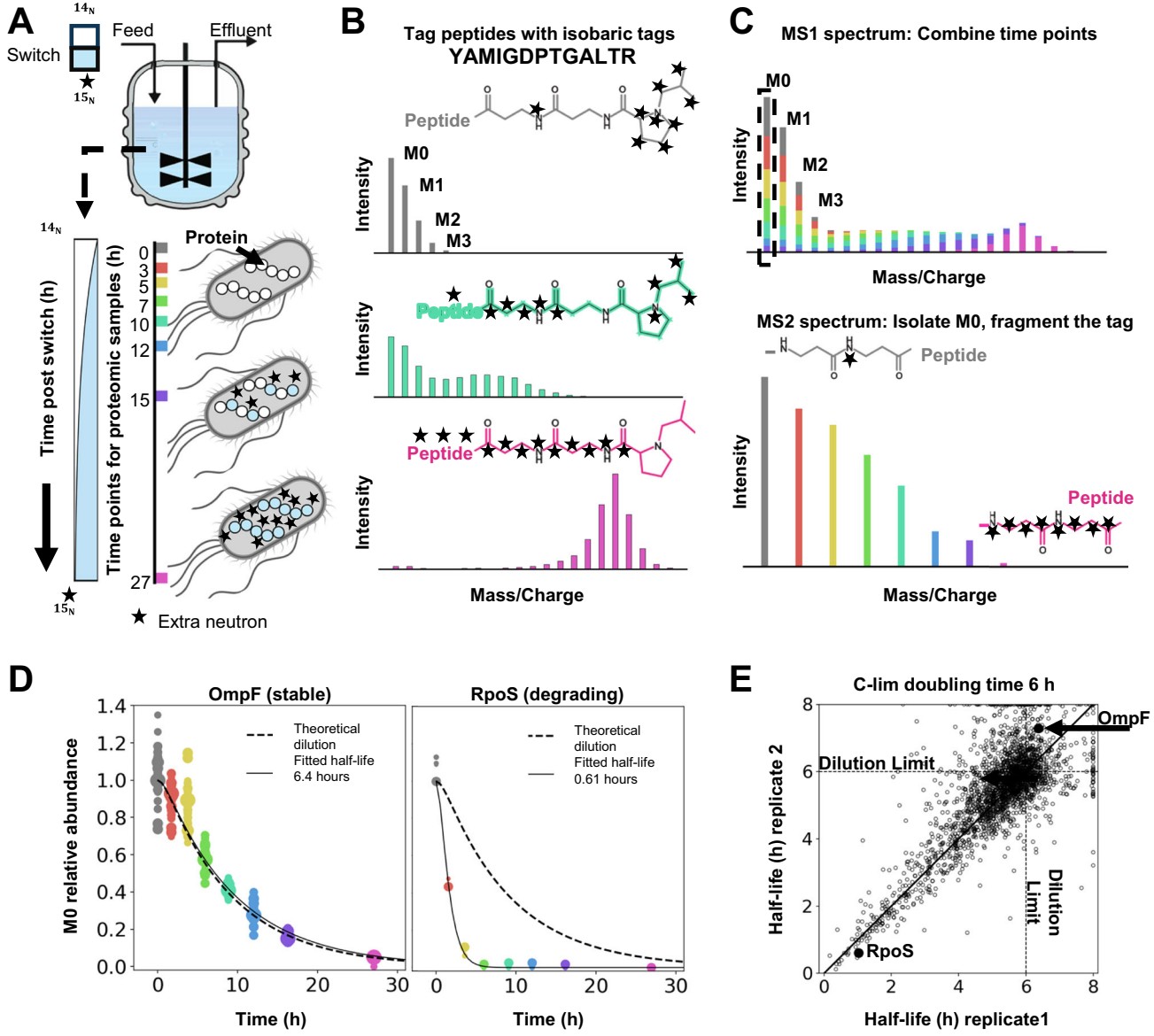

**Fig. 1 | Combining heavy isotope labeling with an accurate multiplexed proteomics method (TMTproC) enables high-quality measurements of unperturbed protein turnover. A** Experimental setup. *E. coli* cells were grown in chemostats with a defined doubling time. After reaching steady state, the chemostat feed was switched to a medium with $^{15}N$-labeled ammonium. Newly synthesized proteins will increasingly incorporate heavy isotopes. Proteomics samples were collected at various time points to determine the protein turnover rate[98]. **B** Theoretical isotopic envelopes of an example tryptic peptide, which is assumed to be stable (protein is removed from the vessel only through dilution). Over time, the increasing fraction of heavy ammonium in the peptide's structure shifts the isotopic envelope to higher masses. Peptides were labeled with isobaric tags (TMTpro) to encode different time points. **C** Top: theoretical MS1 spectrum for a single peptide species after combining labeled peptides from all the time points. The mass spectrometer was set to isolate the monoisotopic peak ($M_0$) and fragment the peptide. Bottom: the resulting complement reporter ions (peptide plus broken tag) enable accurate quantification of the relative abundance within the $M_0$ peak over time. **D** Example measurements for the stable OmpF protein and rapidly degrading RpoS protein. Each dot indicates the peptide quantification relative to the median level measured when the feed was switched. The size of each point is proportional to the number of measured ions. Fitting the observed data with the theoretical decay profile for $M_0$, we can extract the total half-life for each protein (solid curve). The dotted curve shows the theoretical decay for a stable protein ($n = 8$ time points). **E** Scatter plot of measured protein total half-lives for biological replicates of carbon-limited *E. coli* grown with a 6-h doubling time. Dotted lines indicate the cell doubling times. The solid line marks the 1:1 line. The total half-lives for each protein were calculated from the fits shown in (**D**). Median standard deviation for the total half-lives between the replicates is 0.3 h.

time. We obtain half-lives for ~2.6k *E. coli* proteins per experiment with a median standard deviation of 0.3 h (Fig. 1E, Supplementary Table 1). Having established this technology, we acquired similar measurements for 13 different growth conditions, each with two biological replicates, quantifying the turnover rates of ~3.2k proteins in at least one condition (Table 1, Supplementary Data 1, Supplementary Data 7). We then used this resource to investigate how *E. coli* adapts protein turnover under various growth conditions.

## *E. coli* recycles its cytoplasmic proteins under nitrogen limitation

Building on our method to measure protein turnover, we wanted to compare protein turnover rates under various nutrient limitations. To this end, we compared carbon (C-lim), phosphorus (P-lim), and nitrogen (N-lim) limitation measurements from chemostats with 6-h doubling times. We found that most proteins in C-lim are stable with a measured total half-life close to the theoretical dilution time (Fig. 2A).

**Table 1 | Summary of the 13 growth conditions for which we measured protein turnover rates**

| | Strain NCM3722 | Condition | Reactor type | Doubling time | Replicates | # of proteins |
|---|---|---|---|---|---|---|
| 1. | Wild type | Minimal media | Batch | 42 min | 2 | 2555 |
| 2. | Wild type | C-lim | Chemostat | 3 h | 2 | 2665 |
| 3. | Wild type | C-lim | Chemostat | 6 h | 2 | 2651 |
| 4. | Wild type | C-lim | Chemostat | 12 h | 2 | 2697 |
| 5. | Wild type | P-lim | Chemostat | 6 h | 2 | 2469 |
| 6. | Wild type | P-lim | Chemostat | 12 h | 2 | 2619 |
| 7. | Wild type | N-lim | Chemostat | 6 h | 2 | 2467 |
| 8. | Wild type | N-lim | Chemostat | 12 h | 2 | 2460 |
| 9. | Δ hslV | N-lim | Chemostat | 6 h | 2 | 2393 |
| 10. | Δ lon | N-lim | Chemostat | 6 h | 2 | 2390 |
| 11. | Δ clpP | N-lim | Chemostat | 6 h | 2 | 2491 |
| 12. | Δ hslV Δ lon Δ clpP | N-lim | Chemostat | 6 h | 2 | 2809 |
| 13. | Δ smpB | N-lim | Chemostat | 6 h | 2 | 2535 |
| | | | | | Union | 3262 |

Using biological replicates to identify degrading proteins with high confidence (Fig. 2B), we found that 15% of the proteome is actively degraded in C-lim (*p*-values < 0.05). Protein half-lives under P-lim have a similar distribution and a similar percentage of proteins that degrade with high confidence. However, in N-lim we found that 43% of proteins are actively degraded (Fig. 2B, *p*-value < 0.05).

We found that the increase in protein degradation in N-lim could be attributed to the active degradation of a wide range of cytoplasmic proteins (Fig. 2C, D). The mode protein total half-life for membrane and periplasmic proteins in all three conditions is very close to the theoretical dilution limit. In contrast, the mode protein total half-life for cytoplasmic proteins is significantly shorter under N-lim than C-lim or P-lim. We estimate that 56% of cytoplasmic proteins are actively degraded in N-lim, while only 13% of membrane proteins and 4% of periplasmic proteins undergo active degradation in this condition. Due to measurement noise and low sample sizes, we expect to be statistically underpowered and that these estimates are likely lower bounds of the true extent of protein degradation in N-lim.

We then tested whether cytoplasmic protein degradation in N-lim chemostats extends to the more physiologically relevant case of batch starvation. We grew *E. coli* cells in minimal medium until they reached an $OD_{600}$ of ~0.4. We then switched the exponentially growing cells into medium depleted of nitrogen (Fig. 2E). Once again, many cytoplasmic proteins are degraded under nitrogen starvation, and membrane/periplasmic proteins are largely stable. Thus, *E. coli* cells slowly degrade their cytoplasmic proteins when nitrogen is scarce in both chemostats and batch cultures. About 2/3 of the cell's nitrogen is stored in proteins[52]. The degradation of proteins upon nitrogen starvation likely allows the regeneration and recycling of scarce amino acids and enables *E. coli* to produce new proteins to adapt to new environments.

**Measuring protein turnover in knockout mutants allows the identification of protease substrates**

Next, we were interested in discovering the protease(s) responsible for the large-scale turnover of cytoplasmic proteins in N-lim. Combining protein-turnover measurements with genetic protease knockouts allows us to investigate protease-substrate relationships on a proteome-wide level. Since unfolding and degrading stably folded cytoplasmic proteins requires energy, we focused on assigning substrates to the ATP-dependent proteases. In *E. coli*, there are four known cytoplasmic ATP-dependent protease complexes: ClpP (in complex with ClpX or ClpA), Lon, HslV (in complex with HslU), and FtsH[1]. We identify putative substrates for the first three of these proteases by comparing the protein half-lives in protease knockout (KO) with

wildtype (WT) cells (Fig. 3A). We were able to validate several known protease-substrate targets and identify degradation pathways using these experiments. For example, the unfoldase ClpA is completely stabilized by knocking out *clpP*, consistent with previous literature[53]. We identified Tag and UhpA as putative substrates of Lon and HslV, respectively. However, many proteins still degrade in the three protease KO lines, e.g., the phosphatase YbhA—which contributes to Vitamin B6 homeostasis—still rapidly turns over with a total half-life of ~1 h in each knockout strain[54]. Surprisingly, even the proteins that increase their half-lives in single KOs are often not completely stabilized. Additionally, bulk cytoplasmic proteins are still degraded in all three single KOs.

Deleting *ftsH* is more complicated than the other proteases. One of its substrates, LpxC, catalyzes the committed step in the lipid A synthesis pathway. Lipid A is the hydrophobic anchor of lipopolysaccharides (LPS), a critical outer membrane component. Deletion of *ftsH* leads to increased levels of LpxC, causing an accumulation of LPS that makes the cells nonviable[55]. *ftsH* null cells can be rescued with a mutation of FabZ (L85P), which slows LPS synthesis and compensates for the increased LpxC levels[55]. Interestingly, we were only able to generate the Δ*ftsH fabZ (L85P)* strain in DY378 background[56]. Our attempts to knock out *ftsH* in the NCM3722 background used for the remainder of this paper failed. We are currently investigating which other modifications in DY378 might make Δ*ftsH fabZ (L85P)* viable. These Δ*ftsH fabZ (L85P)* cells are viable, though unfortunately, they grow too slowly on minimal media and are washed out of the chemostat. Therefore, we could not measure protein turnover in a *ftsH* mutant in a similar manner to the other proteases. Instead, we repeated the batch nitrogen starvation experiments (Fig. 2E). Similar to the WT cells, cells lacking *ftsH* degraded cytoplasmic proteins. In contrast, membrane proteins are mostly stable (Fig. 3B). This indicates that none of the four known ATP-dependent proteases in *E. coli* are individually responsible for the large-scale cytoplasmic recycling that occurs under nitrogen limitation.

We then asked if proteases might act redundantly, i.e., multiple proteases share a substrate, which could mask the effects of deleting individual proteases. To this end, we measured protein turnover in a triple KO line (Δ*hslV* Δ*lon* Δ*clpP*) in nitrogen limitation. A quantitative comparison of protein turnover rates between the triple KO and the individual KOs allows us to assign the contribution of each protease in turning over a substrate (Fig. 3D). We can classify the substrates into six groups: those being degraded predominantly by a single protease, those where the effects of the individual proteases are additive, those that are stabilized more in the triple KO than the combined effect of individual KOs (redundantly degraded), and those that are still actively

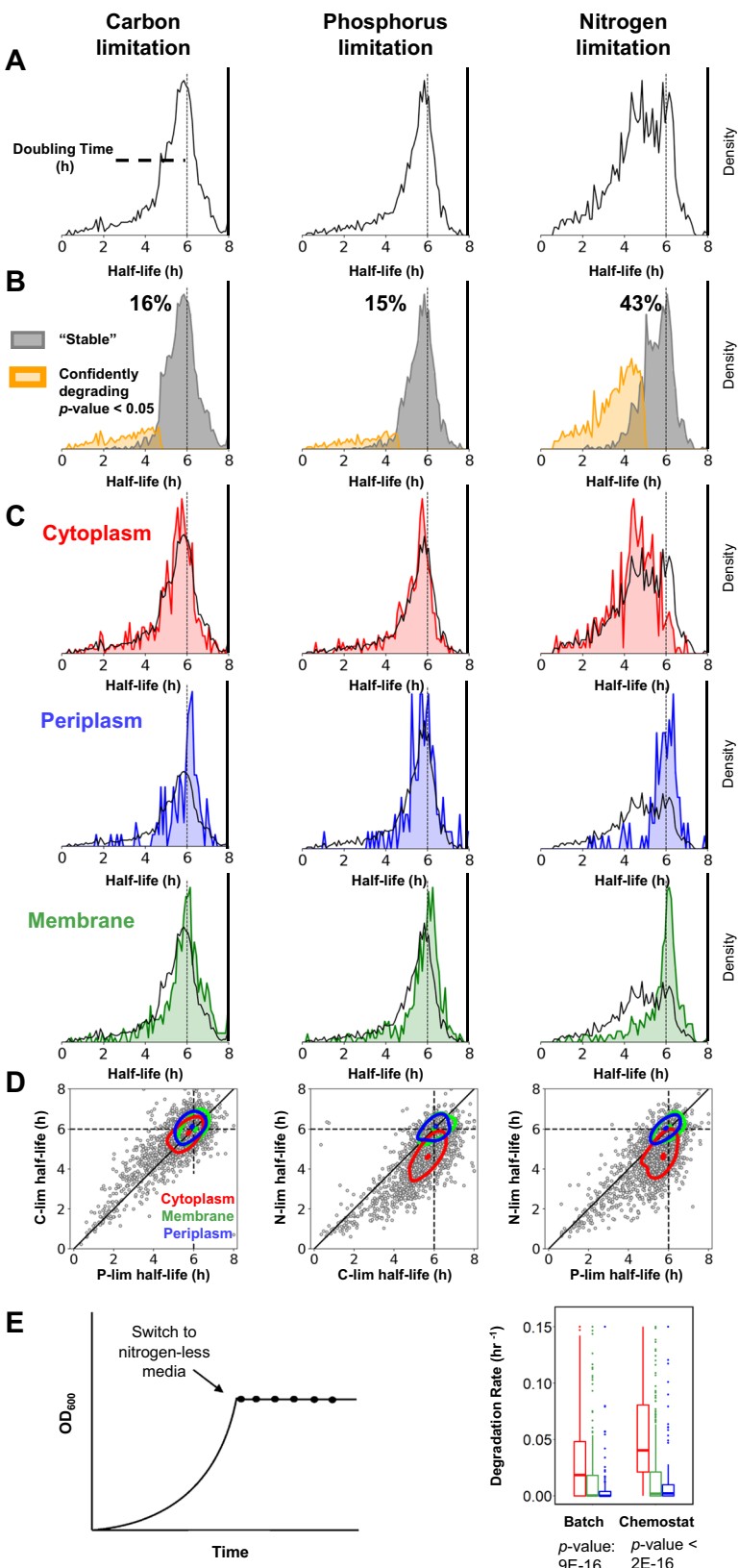

degraded in the triple KO (Supplementary Data 2, Supplementary Fig. 7).

We classified 64 and 14 substrates to be predominantly degraded by ClpP and Lon, respectively. We only assigned one substrate uniquely to HslV: UhpA, a transcriptional regulator that activates the transcription of genes involved in transporting phosphorylated sugars[57]. Eighty-two proteins are degraded additively, a notable example of which is IbpA, a small chaperone. Previous studies have proposed that Lon degrades free IbpA/IbpB and bound client proteins[58]. We found that ClpP and Lon contribute approximately equally to the degradation of IbpA, and their contribution is additive.

**Fig. 2 | *E. coli* recycles its cytoplasmic proteins when nitrogen is limited.**
**A** Histogram of protein total half-lives for *E. coli* grown in chemostats under C-lim, P-lim, and N-lim. The vertical line marks the dilution limit set by the 6-h doubling time. Total half-lives greater than doubling time indicate measurement noise. Under C-lim and P-lim, most proteins have total half-lives equal to the doubling time, suggesting they are stable. However, under N-lim many proteins are actively degraded. **B** Separation of the proteome into stable proteins (grey) and actively degrading proteins (yellow). All proteins with *p*-value < 0.05 are called confidently degrading (*t*-test, one-sided, *n* = 2). In C-lim and P-lim, 15% of the proteome turns over with high confidence. In contrast, under N-lim, 43% of the proteome turns over. **C** Distribution of total half-lives for proteins from different subcellular localizations overlayed against the entire proteome. Most proteins are stable under C-lim and P-lim, irrespective of localization. However, nearly all cytoplasmic proteins slowly degrade under N-lim while the membrane and periplasmic proteomes are largely stable. **D** Scatter plots of protein total half-lives in different nutrient limitations. The dotted black lines mark the dilution limit, the solid black line denotes perfect agreement. Contour plots contain 50% of the probability mass for each subcellular compartment. The contour plots of membrane and periplasmic proteins are centered around the dilution limit in all the binary comparisons, indicating that most of these proteins are stable under all limitations. However, the shift in the contour plots of the cytoplasmic proteins on comparing N-lim with P-lim and C-lim suggests that the cytoplasmic proteins are degraded in N-lim.
**E** Measurement of protein turnover rates under complete nutrient starvation in batch. In batch, like the N-lim chemostat, the cytoplasmic proteins are degrading with high confidence as compared to the membrane and periplasmic proteins (ANOVA, *p*-value = 9E-16, *n* = 2114 proteins). The box extends from the first quartile to the third quartile of the data, with a line at the median. The whiskers extend from the box to the farthest data point lying within 1.5× the inter-quartile range from the box.

We classify 41 proteins as being redundantly degraded by two or more proteases. For example, YbhA is rapidly degraded in all single KO strains but stabilized in the triple KO, indicating that at least two of these proteases act redundantly. Interestingly, the majority of cytoplasmic proteins are still slowly degraded in the triple KO under nitrogen limitation, and we classified ~100 proteins as still being actively degraded (Fig. 3E). LexA, an SOS repressor, auto-degrades itself under stress and unperturbed growth[59,60]. Consistent with this, LexA still undergoes degradation in the triple KO. It will be interesting to investigate if other proteins with short half-lives in the triple KO are auto-degrading, degraded by FtsH, or if other mechanisms are at play.

To validate our classifications, we compared our protease-substrate relationships with previous proteome-wide measurements. We see a significant overlap (*p*-value = 6E-9) of our identified ClpP substrates with substrates identified via a trap mutant (Fig. 3F)[20]. However, we do not observe an overlap of our putative Lon substrates with a previous Lon-trap experiment (*p*-value = 0.22)[21]. This lack of overlap is most likely caused by our separating the Lon trap substrates into the different classifications, indicated by a more significant overlap with the substrates that were stabilized in any of our KO strains (*p*-value = 0.05). This is consistent with previous observations that Lon substrates are often shared with other proteases[61]. Interestingly, the putative substrates of HslV identified through a microarray study show a strong overlap with the proteins we classify as additive or redundant (*p*-value = 0.001)[23]. This is consistent with previous reports that HslV substrates are shared with other proteases[26,62]. We also found enrichment (*p*-value = 0.002) between substrates identified in a previous FtsH trap[63] study and additive or redundant substrates, consistent with findings that FtsH often degrades proteins that are also substrates for other proteases[27]. The lack of significant overlap between the proteins still degrading in the triple KO and FtsH-trap substrates implies that FtsH is likely not involved in the degradation of these substrates.

Surprisingly, 40% of active protein degradation in nitrogen limitation in wild-type cells persists upon knocking out the three canonical ATP-dependent cytoplasmic proteases (Fig. 3G, details of calculation in the supplementary note and supplementary data 6). We could not generate a viable quadruple KO with *ftsH* deletion, so we cannot rule out the possibility that all four proteases act redundantly as an explanation of the remaining protein degradation. However, the results from the individual *ftsH* knockout (Fig. 3B) and the lack of overlap between degrading proteins and the FtsH-trap experiment (Fig. 3F) are evidence against FtsH being responsible for the remaining degradation. Regardless, a major pathway for degrading proteins in *E. coli* remains to be discovered: either FtsH plays a much bigger role than is currently believed, or a completely new mechanism degrades cytoplasmic proteins under nitrogen starvation.

## Analyzing features of rapidly turning over proteins

We found that most short-lived proteins have similar half-lives regardless of nutrient limitation (Fig. 4A, Supplementary Data 3). With gene-set enrichment[64], we found that rapidly degraded proteins were enriched in transcriptional regulators (Benjamini–Hochberg adjusted *p*-value = 4E-4). A protein's response time depends on its turnover rate[5]. Proteins involved in transcriptional regulation might need to rapidly adjust their levels to changing growth conditions.

Using our data set, we validated examples of degradational regulation that had previously been reported and also uncovered targets. Of the 24 proteins with the fastest average turnover rates, 17 were previously reported to be degraded (Fig. 4B). Seven proteins—ThiH, YgaC, SixA, YciW, CbI, ThiG, EpmB—had no prior evidence in the literature for degradation. Interestingly, six rapidly degrading proteins—ThiH, BioB, IscA, IscR, EpmB, Fnr—contain Fe–S clusters, which is significantly higher than expected by random chance (BH *p*-value = 0.048). Flynn et al. previously proposed that Fe–S binding proteins are degraded under aerobic conditions, likely because the Fe–S clusters are oxidized, destabilizing the protein[20].

Multiple metabolic enzymes such as PatA, LpxC, and HemA are also rapidly degraded. Rapid degradation allows for immediate and direct control over intracellular protein levels based on cellular demand. PatA (Putrescine-Aminotransferase) is involved in putrescine (polyamine) degradation ($K_M$ = 9 mM) and is unstable under standard growth conditions with high putrescine levels, in which another enzyme (PuuA - glutamate-putrescine ligase) dominates usage of putrescine. PatA is expected to stabilize in specific growth conditions with low putrescine concentrations[65]. LpxC, a protein required for lipid A synthesis, is rapidly degraded under slower growth to balance LPS production with cellular demand[55]. HemA, involved in porphyrin biosynthesis, is degraded when the media lacks heme as an iron source[66]. DnaQ, the proofreading exonuclease of the stable DNA polymerase III core enzyme [DnaE][DnaQ][HolE], is rapidly degraded ($t_{1/2}$ = 1.2 h). DnaE is more stable with a total half-life of 5 h, whereas HolE is undetected in our data set, likely due to its short length. Free DnaQ is unstable but stabilized on complexation with HolE[67].

We next tested whether these rapidly degrading proteins share attributes such as their physiochemical properties, sequence features, or structural characteristics. We found that smaller proteins (MW < 10 kDa) have significantly shorter half-lives regardless of the nutrient limitation (Fig. 4C, Supplementary Fig. 2A). This enrichment was more pronounced at lower total half-life cutoffs (Fig. 4C). On the other hand, charge and isoelectric point were not significantly correlated with half-lives under P-lim and C-lim (Supplementary Fig. 2B, C). However, both the charge and the isoelectric point of a protein were correlated with half-lives under N-lim (Supplementary Fig. 2B, C *p*-value = E-20). This is most likely because cytoplasmic proteins are short-lived under N-lim while membrane proteins are typically stable. Membrane proteins tend

to have higher isoelectric points and more positive charge due to their interaction with negatively charged phospholipids[68,69].

One obvious sequence feature to investigate is the N-end rule, which relates a protein's stability to its amino-terminal residue[70]. Amino-terminal arginine, lysine, leucine, phenylalanine, tyrosine, tryptophan, and formylated N-terminal methionine (fMet) are believed to be destabilizing residues, whereas the other residues are believed to be stabilizing[70,71]. First, we determined the in vivo N-terminal residue of

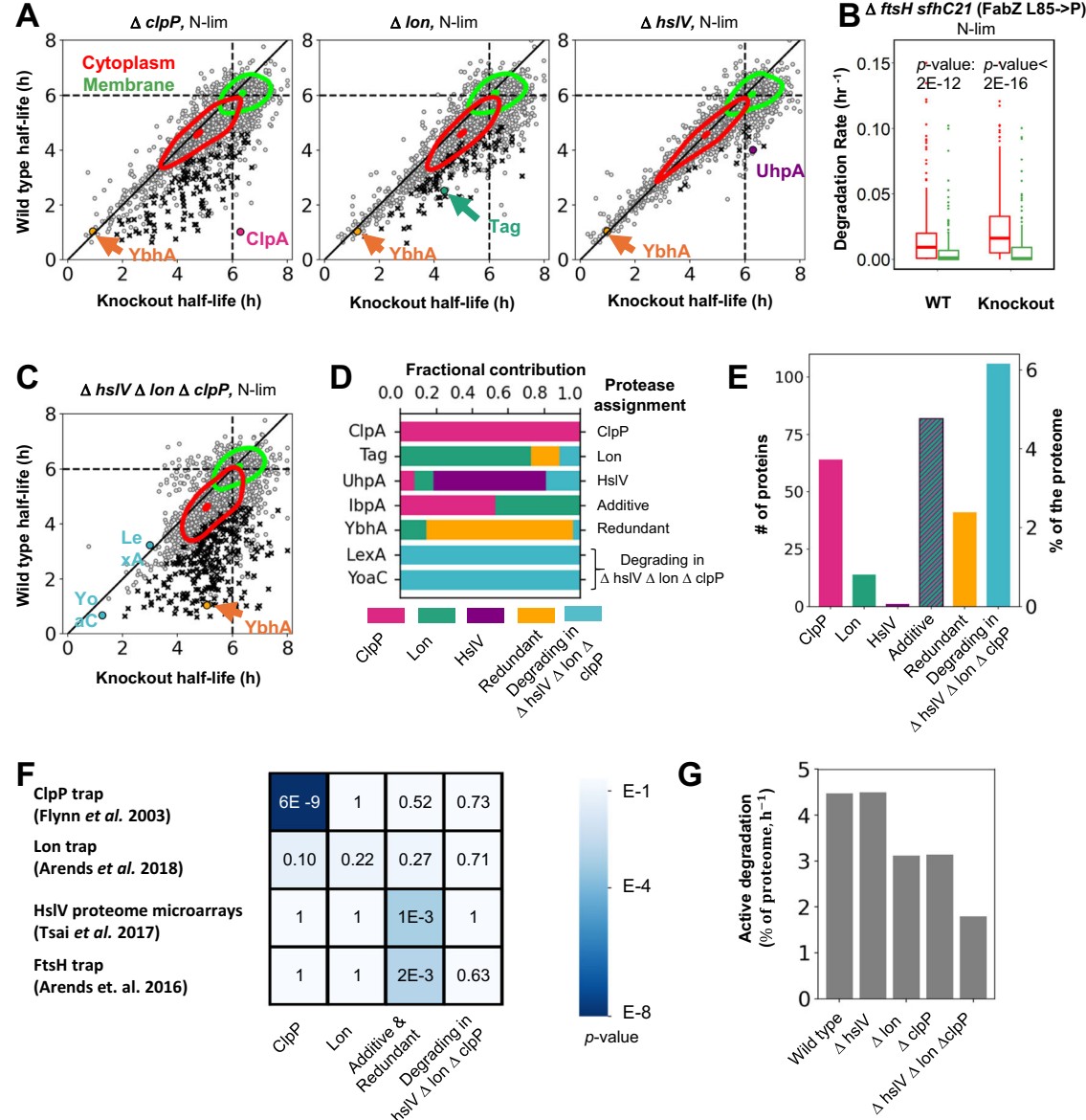

**Fig. 3 | Measurement of protein turnover in protease knockout strains enables proteome-wide identification of protease substrates. A** Scatter plots of protein total half-lives of N-limited wild type (WT) compared to ΔclpP, Δlon, and ΔhslV knockout (KO) cells. Dotted lines mark the dilution limit, the solid black line indicates perfect agreement. Substrates (black x) increase their total half-lives in KOs with high confidence (t-test, one-sided, p-value < 0.10). ClpA (pink), Tag (teal), and UhpA (purple) are the substrates of ΔclpP, Δlon, and ΔhslV, respectively. However, the protein YbhA (orange) is still degraded in individual KOs. Contour plots containing 50% of the probability mass for the cytoplasmic (red) and membrane (green) proteins indicate that individual KOs degrade bulk cytoplasmic proteins. **B** Since ΔftsH cells cannot grow in chemostats, we repeated the batch starvation assay as in Fig. 3E. The box extends from the first quartile to the third quartile, with a line at the median. The whiskers indicate 1.5× the inter-quartile range from the box. Results indicate that the ΔftsH cells, like the WT, also degrade their cytoplasmic proteins under nitrogen starvation (t-test, two-sided, p-value = 2E−12 for WT and p-value < 2E−16 for ΔftsH, n = 1519). **C** Scatter plots of protein total half-lives of WT and ΔclpP Δlon ΔhslV cells in N-lim. The substrates (black x) increase their total half-lives in the triple KO (t-test, one-sided, p-value < 0.09). Many proteins are

still degrading in the triple KO, e.g., LexA and YoaC (blue). In fact, the bulk cytoplasm is still degraded. However, many more proteins are stabilized in the triple KO compared to the individual KOs, indicating redundancy among substrates, e.g., YbhA (orange). **D** Comparing the shifts in the WT and KO strains' total half-lives, we assign each protease's contribution to active protein turnover. The bar graph represents examples from each of the six categories–turnover explained predominantly by ClpP, Lon, HslV, additive contributions, redundant contributions, and actively degrading proteins in the triple KO. **E** Bar graph for the number of substrates and the % of the proteome assigned to each of the six categories described in (**D**). **F** Comparison of the substrates from our categories in E with previous proteome-wide substrate-protease assignment studies. ClpP trapped substrates significantly overlap with the identified ClpP substrates (Fisher test, two-sided, p-value = 6E−9), and previously identified substrates of HslV and FtsH show a significant overlap with redundant and additive substrates (Fisher test, two-sided, p-value = 1E−3). **G** Comparison of the percentage of active turnover per hour across the protease KOs under N-lim. Even after knocking out hslV, lon, and clpP simultaneously, 40% of the WT proteome turnover remains, suggesting that a major pathway of protein degradation in E. coli remains to be discovered.

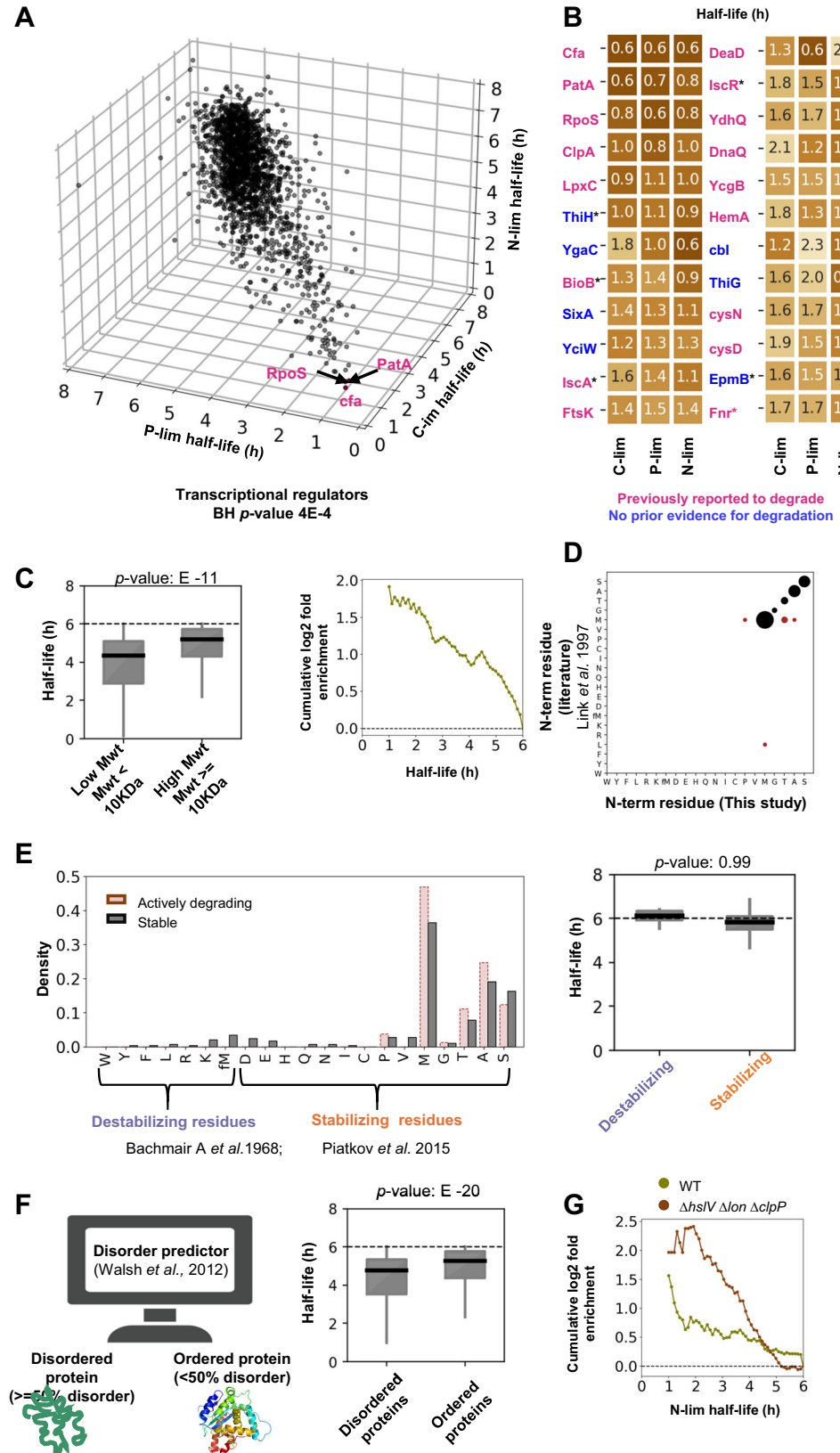

~600 proteins using a label-free proteomics data set (Supplementary Data 4). MS2 spectra were searched with the Sequest algorithm[72] considering all possible N-terminal tryptic subfragments for a protein. Encouragingly, when we compared a small subset of the identified N-termini with previous data in literature obtained using Edman Degradation[73] we found nearly perfect agreement (55/61 proteins)

(Fig. 4D). Surprisingly, however, our rapidly degrading proteins showed no enrichment for the previously reported destabilizing N-terminal amino residues (Fig. 4E). Interestingly, few proteins detected in either dataset had destabilizing N-terminal residues. It is possible that proteins with destabilizing N-termini are immediately degraded and therefore difficult to detect. Proteins which had destabilizing

**Fig. 4 | Features of proteins with short total half-lives. A** Scatter plot of protein total half-lives for C-lim, P-lim, and N-lim conditions. The total half-lives of rapidly degrading proteins are typically similar under different nutrient limitations. Three proteins with the shortest average total half-lives are marked. Short-lived proteins are enriched for transcriptional regulators (BH $p$-value = 4E−4). **B** The 24 proteins with the shortest mean total half-lives. For eight of these proteins (in blue), we could not find any prior literature evidence for degradation, and six (marked with *) contain Fe-S clusters (BH $p$-value = 0.048). (**C, E, F**) For all the boxplots, the box extends from the first quartile to the third quartile of the data, with a line at the median. The whiskers extend from the box to the farthest data point lying within 1.5× the interquartile range from the box. **C** Smaller proteins have shorter total half-lives. Left: Box plot of total half-lives averaged over all the nutrient limitations for low and high molecular weight (MW) proteins ($n$ = 2864 total proteins, $p$-value = 1E−11, Mann–Whitney U, one-sided). Right: Fold enrichment of low versus high MW proteins as a function of total half-life. **D** Comparison of the proteome-wide N-terminus amino acid residues obtained from this study and prior literature. The size of the marker indicates the number of proteins with a particular residue. **E** No correlation between the N-terminal protein residue and protein total half-lives (N-end rule). Left: Distribution of the N-terminus residues for actively degrading and stable proteins. Right: Box plots of total half-lives for the destabilizing and stabilizing residues on their N-terminus ($p$-value: 0.99, Mann–Whitney U, one-sided, $n$ = 373 total proteins). **F** Disordered proteins tend to have shorter total half-life[99]. Left: The Espritz algorithm classified proteins as ordered or disordered. Right: Box plot of total half-lives for disordered and ordered proteins ($n$ = 2865 total protein, $p$-value = E−18, Mann–Whitney U, one-sided). **G** Fold enrichment of disordered versus ordered proteins as a function of total half-life for the WT (olive) and triple protease KO (brown) cells. Rapidly turning over proteins are enriched for the disordered category. This enrichment becomes more pronounced when three proteases are knocked out.

residues exposed via cleavage would similarly be short-lived and low abundant. In this case, the main-determinant of protein half-life would be the rate at which destabilizing N-termini are exposed. Either way, our results suggest that the N-terminus of *E. coli* proteins is not the primary determinant of proteins' in vivo stability.

The SsrA-tagging system is another known degron used for marking polypeptides for degradation whose translation has stalled[74]To investigate how much of the observed protein degradation could be attributed to the system, we knocked out the *smpB* gene (codes for a protein in the SsrA tagging complex[75]), and measured gene-by-gene protein turnover in nitrogen limitation with a 6-h doubling time in duplicate. We did not observe significant changes in protein turnover compared to the wild-type strain (Supplementary Fig. 10). It's possible that the primary targets for the SsrA-tagging system are low abundant relative to stably-fold proteins, or that another pathway (e.g., ArfA-mediated[76]) for releasing stalled ribosomes is redundant with the SsrA system.

Another sequence feature previously shown to affect protein stability in bacterial and eukaryotic cells is intrinsically disordered protein segments[77,78]. To this end, we determined the percentage disorder for all the proteins using the Espritz algorithm[79]. Disordered proteins had significantly shorter half-lives than ordered proteins (Fig. 4F, $p$-value = E−208). Interestingly, this enrichment further increases when we use protein half-lives measured in the triple protease knockout cells (Δ*hslV* Δ*lon* Δ*clpP*) (Fig. 4G). This is consistent with ATP-dependent proteases being able to unfold and digest structured proteins. Once these proteases are removed, the remaining proteins with short half-lives should be enriched for those that are unstructured and therefore prone to degradation by energy-independent proteases.

### Analysis of turnover for functionally related proteins

Next, we investigated protein turnover for functionally related protein modules, such as multiprotein complexes, operons, and metabolic pathways. We calculated each module's coefficient of variation (CV) and compared this to the CV distribution when proteins were randomly assigned to sets. We observe that the functionally associated modules exhibit significantly lower variance than if the proteins were randomly assigned to each module (Fig. 5A), suggesting that functionally associated proteins tend to exhibit similar half-lives. For example, twelve of the fourteen proteins involved in phosphonate metabolism and transport are rapidly degraded (average total half-life = 0.7 h) under P-lim (Fig. 5B). These proteins were 16-fold more abundant in P-lim compared to N-lim and C-lim (Supplementary Data 5). Therefore, we were unable to measure their half-lives in C-lim or N-lim, so it's unclear if they turn over in these limitations as well.

One pathway where this is particularly noteworthy is the *rut* pathway for pyrimidine degradation. The pathway is transcribed by a single operon of seven genes (*rutABCDEFG*) and has one adjacent transcriptional repressor, *rutR*. In our database, we detected five of the seven proteins plus the repressor. Of these five proteins, four are rapidly degraded in the wild-type strain and the single protease knockouts. The fifth protein, RutG, is found in the membrane, which likely explains its stability. However, when we knock out *lon*, *clpP*, and *hslV* simultaneously, the four proteins are all significantly stabilized (Fig. 5C).

The study that originally characterized the *rut* pathway showed that when the *rut* genes are transcriptionally upregulated, *E. coli* was able to grow on thymidine as the sole nitrogen source in minimal media at room temperature[80]. We reasoned that by removing their degradation, we could similarly increase protein concentrations and observe the same growth on thymidine. Indeed, the triple protease knock-out strain grows to an $OD_{600}$ of nearly 0.28 on thymidine while the wild-type strain reaches an $OD_{600}$ of 0.03 (Fig. 5D).

Additionally, proteins associated with flagella show correlated expression levels and half-lives. Surprisingly, most of the proteins forming the basal flagellar body are stable, but the filament (FliC, FliD), motor (MotA, MotB), and sensory proteins (CheA, CheW) are rapidly degraded (Fig. 5E). Future work will be required to decipher the underlying mechanisms and functional relevance.

In general, proteins that form a complex tend to exhibit similar half-lives (Fig. 5F). Several complexes whose subunits are degraded at different rates are known to interact weakly or transiently or have subunits which are expressed non-stoichiometrically, suggesting that at least some of these discrepancies might be due to annotation details. For example, ClpA and ClpX are the unfoldases in complex with ClpP. Autodegradation of ClpA is used to regulate the number of ClpAP complexes in the cell and the flow of substrates to ClpAP[53]. Finally, antitoxins like PrlF are subject to regulated degradation while their toxin counterparts are stable[81,82].

The ribosome is one of the heterocomplexes which exhibits unanticipated patterns under different nutrient limitations (Fig. 5G). Under both C-lim and N-lim, ribosomal proteins are slightly more stable than the median cytoplasmic protein. However, under P-lim, ribosomal proteins are less stable than typical cytoplasmic proteins. rRNA contains about 50% of the cellular phosphorus[83]. Therefore, cells likely recycle the phosphorus stored in rRNA when phosphorus is scarce, and associated ribosomal proteins might become unstable once their binding partners are lost.

### Active protein degradation rates typically do not scale with division rates

So far, we have compared turnover rates under various nutrient limitations but with the same cell division time. We wanted to determine how active protein degradation scales with cell cycle time. The total turnover rate ($k_{total}$) of a protein is a combination of active degradation ($k_{active}$) and dilution ($k_{dilution}$) due to cell division (Fig. 6A). We consider two simple and reasonable models of the relationship between these

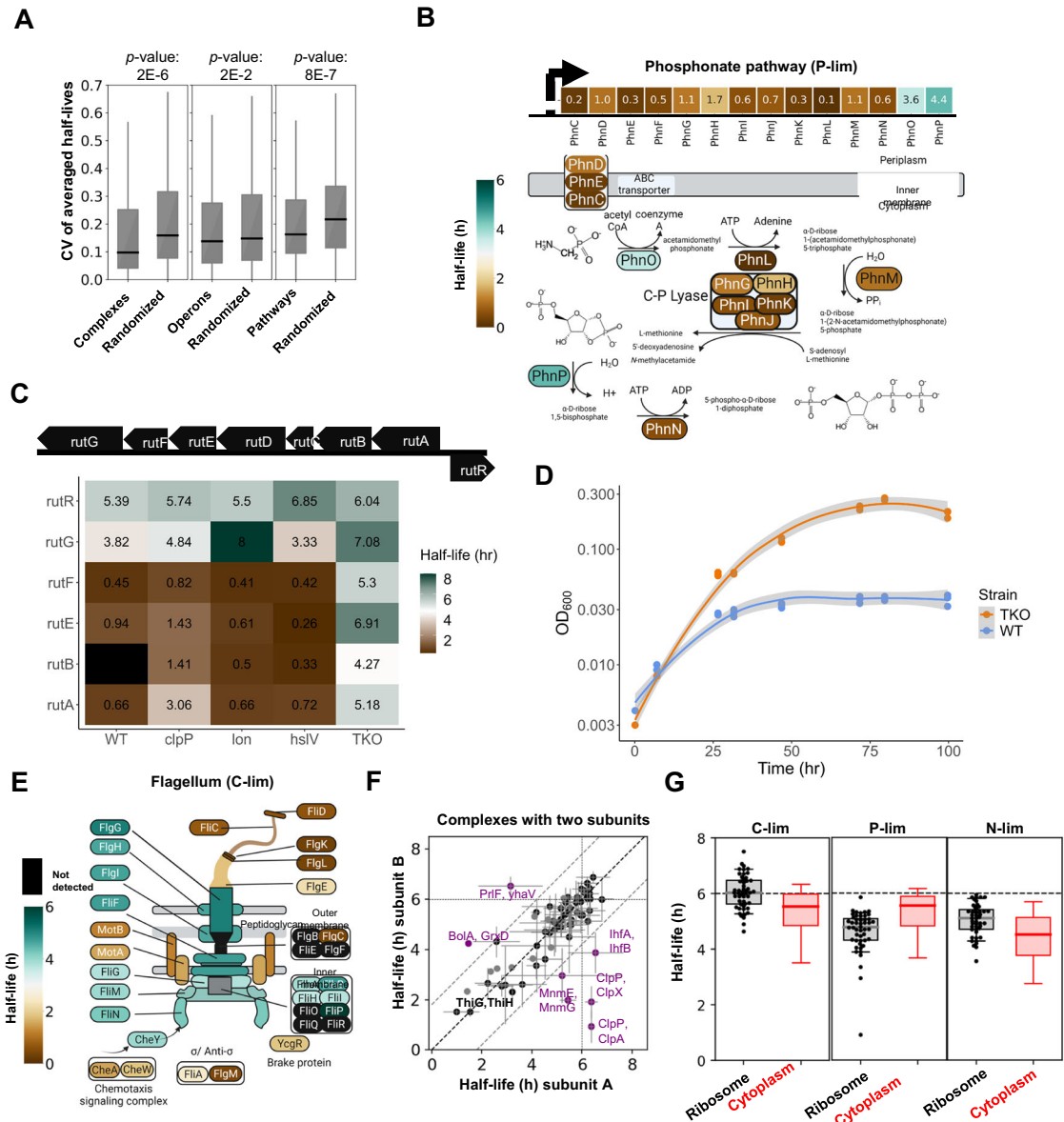

**Fig. 5 | Protein total half-lives of functionally related proteins.** For all the box plots in this figure, the box extends from the first quartile to the third quartile of the data, with a line at the median. The whiskers extend from the box to the farthest data point lying within 1.5× the inter-quartile range from the box. **A** Functionally related proteins tend to have similar total half-lives. Box plots of CVs of total half-lives for functionally related proteins compared with a randomized set of proteins (Mann–Whitney $U$, one-sided, $p$ value 2E−6, 2E−2, 8E−7 for complexes, operons, and pathways respectively). **B** Map of the total half-lives for proteins involved in the phosphonate pathway under P-lim. The phosphonate pathway is encoded in a single operon and is involved in the transport and metabolism of organophosphonates (C–P bond). Most of these proteins degrade rapidly under P-lim. **C** The rut operon consists of seven genes and one repressor ($n = 5$ detected). Four of the five proteins were rapidly degraded in the wild-type strain and ΔclpP, Δlon, and ΔhslV. These four proteins were stabilized in a triple knockout ΔclpP Δlon ΔhslV.

**D** The wild-type and TKO strain were grown on glucose minimal media with thymidine as the sole nitrogen source. The mutant strain grows to a higher maximum OD600 than the wild-type strain. A LOESS curve was fit to the data with a 99% confidence interval. **E** Map of the half-lives of flagellar proteins under C-lim. The basal body elements of the flagellum are largely stable, while filament and motor structure components are rapidly degrading. Flagellum schematic adapted from ref. 100. **F** Scatter plot of protein total half-lives for protein complexes containing two distinct subunits. Data is presented as mean +/− standard deviation. Highlighted in purple are complexes for which the total half-lives of subunits disagree, potentially because their interaction is transient, e.g., ClpA/P or BolA, GrxD. **G** Box plots of total half-lives for ribosomal proteins compared to all cytoplasmic proteins. Ribosomal proteins are more stable than cytoplasmic proteins in C-lim and N-lim but degrade faster than the median of cytoplasmic proteins in P-lim ($n = 30$ for ribosomal proteins).

two parameters. In the first model, $k_{active}$ scales with $k_{dilution}$, i.e., the protein total half-life remains a constant fraction of the cell cycle time. In the second model, active degradation rates are independent of growth rate, i.e., the active degradation rate of each protein remains constant regardless of cell doubling time.

The two models have distinct predictions on how the total protein half-time ($t_{1/2, total}$) should scale with changing cell cycle times. In the scaled model, $t_{1/2, total}$ for each protein linearly increases

with cell cycle time (Fig. 6B). In contrast, in the constant model, the dilution rate dominates for rapidly dividing cells while the contribution from active degradation becomes more relevant for slower dividing cells.

To test the models' predictions, we grew *E. coli* cells with a range of doubling times, including rapidly doubling cells with unlimited growth in minimal medium (0.7 h) and slower doubling cells in carbon-limited chemostats (3 h, 6 h, and 12 h). We found that the data favored

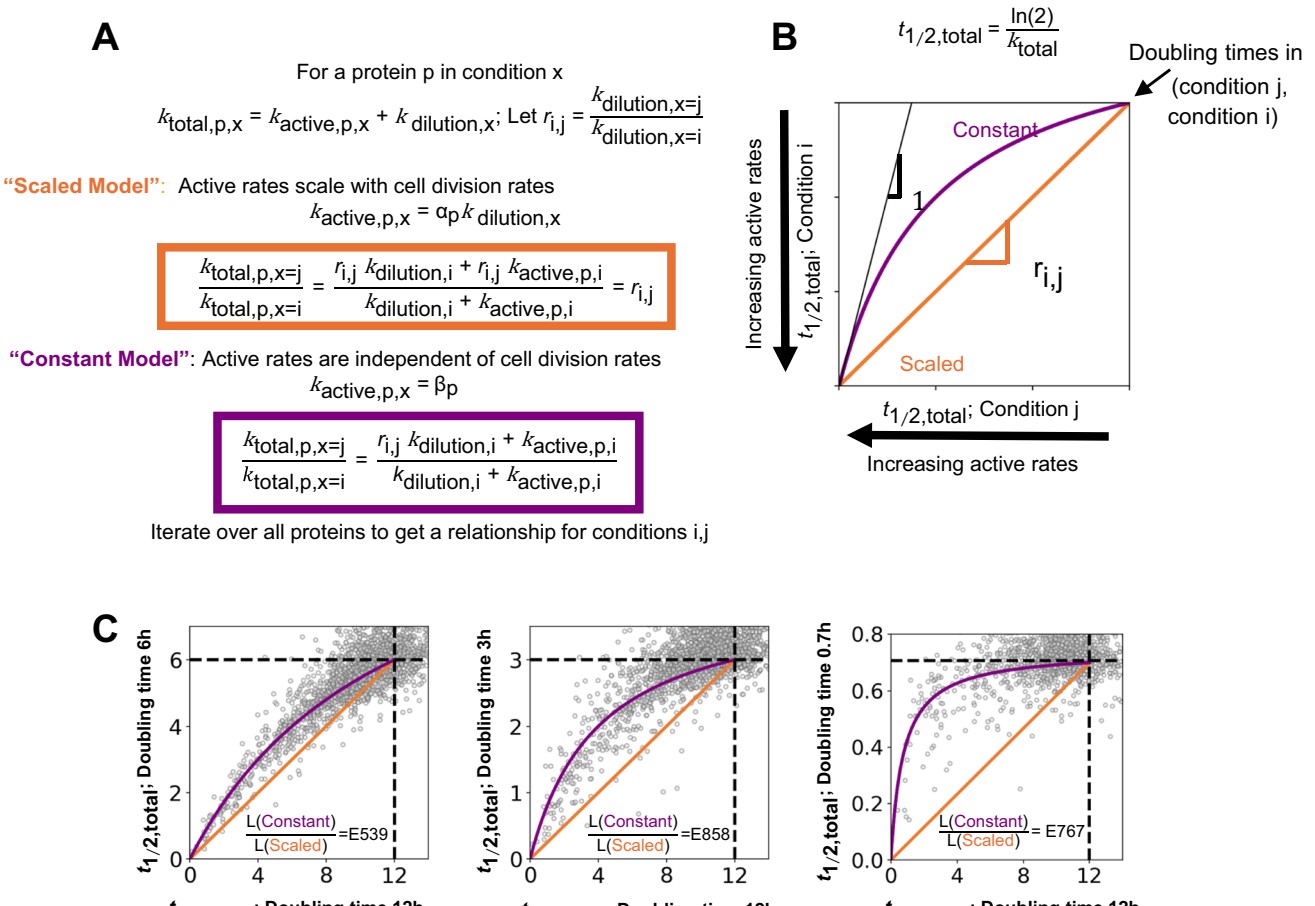

**Fig. 6 | Active degradation rates are generally uncoupled from cell division rates. A** Two simple models describing the relationship between cell division and protein-specific turnover rates. The total protein turnover rate ($k_{total}$) is the sum of the active degradation rate ($k_{active}$) and the dilution rate due to cell division ($k_{dilution}$). In the "scaled model," active degradation rates increase proportionally to division rates with a protein-specific constant ($\alpha_p$), i.e., active degradation remains a constant fraction of the total protein turnover rate. In the "constant model," protein-specific active degradation rates are constant ($\beta_p$), regardless of changing division rates. In this case, for slower-dividing cells, the contribution of active degradation increases relative to dilution. **B** $t_{1/2,\ total}$ is the time taken to replace half the protein. A theoretical plot of $t_{1/2,\ total}$ from two conditions ($i$, $j$) where cell division rates change by a factor of $r_{i,j} < 1$. In the scaled

model, $t_{1/2,\ total}$ values for all the proteins lie on a straight line with slope $r_{i,j}$ (orange). In the constant model, the $t_{1/2,\ total}$ values follow a nonlinear relationship between the two doubling times (purple). For proteins with very high active degradation rates, the constant model predicts that $t_{1/2,\ total}$ will approach the same value for both doubling times, indicated by the slope 1 line (black). For diluting proteins with no active degradation, both models converge to the doubling times of conditions $i$ and $j$. **C** Scatter plots of protein $t_{1/2,\ total}$ for *E. coli* grown at doubling times of 6 h (C-lim), 3 h (C-lim), and 0.7 h (defined minimal media batch) compared to 12 h (C-lim). The dotted lines represent the dilution limit. We observe a strong statistical preference for the "constant model," in which active degradation rates are uncoupled from cell cycle duration. Shown are the likelihood ratios ($L$) of the constant models compared to the scaled models assuming normally distributed errors.

the constant model regardless of which cell cycle times we compared (Fig. 6C). This indicates that active protein degradation rates typically remain constant regardless of cell division rates.

We noted interesting exceptions to the model (Supplementary Fig. 8), however, particularly when comparing slower-growing cells in the chemostat to cells growing without nutrient limitation. For example, RpoS degrades faster in unlimited growth conditions than the non-scaled model would predict based on chemostat measurements. This is consistent with the previous finding that RpoS is rapidly degraded in exponentially growing cells but becomes stabilized when nutrient-limited[84].

Because $k_{active}$ rates are generally constant across cell division rates, we can more accurately measure $k_{active}$ when $k_{dilution}$ is small. Importantly, the observed constancy of degradation rates, regardless of cell cycle times, allows us to extrapolate active degradation rates from conditions with long cell division times (e.g., chemostats with 6-h doubling times) to conditions with more rapid cell division times, in

which separating between active degradation rates and dilution rates is experimentally difficult. Thus, the protein half-lives in this manuscript, primarily obtained in the chemostat, are a valuable resource that can be extrapolated to arbitrary cell division rates.

## Discussion

This paper introduces a technique for the global measurement of protein turnover on a gene-by-gene basis by combining complement reporter ion quantification with heavy isotope labeling of nutrients (Fig. 1). Applying our method to measure protein turnover across multiple nutrient limitations, we found that most cytoplasmic proteins slowly degrade in nitrogen-limited conditions (Fig. 2). By contrast, in phosphorus-limited and carbon-limited conditions, proteins are mostly stable. We observe this phenomenon in a nitrogen-limited chemostat and in a nitrogen-starved batch culture. The slow degradation of cytoplasmic proteins is likely a strategy *E. coli* has developed to keep scarce amino acids available, which could be critical to various

metabolic processes, including the ability to synthesize new proteins and adapt the proteome to changing environments. To test this hypothesis, we measured growth curves following nutrient upshift from nitrogen starvation to LB media for both the wild-type strain and the mutant lacking *lon*, *clpP*, and *hslV* which has reduced cytoplasmic protein turnover. We find that the triple knock-out is less able to adapt to the new conditions and has a significant growth delay (~80 min) compared to the wild-type (Supplementary Fig. 9). Bulk protein turnover measurements in the 1950s showed that *Saccharomyces cerevisiae* also increases overall protein turnover when starved of nitrogen[15], suggesting that a similar strategy might apply to eukaryotes.

We assigned protein substrates to proteases by measuring the change in protein turnover rates in protease knockout strains (Fig. 3). We were surprised by how little protein degradation changed in the knockout strains, particularly when deleting the canonical proteases ClpP and Lon. We showed that in these knockout strains, only a few proteins have a slower degradation rate, and the observed degradation of cytoplasmic proteins continues. Even when we knocked out *clpP*, *lon*, and *hslV* simultaneously, 40% of total protein turnover remains, including the cytoplasmic recycling and the degradation of many short-lived proteins. However, we observe remarkable additive and redundant effects when comparing protein turnover rates in the individual knockouts with the triple knockout. This suggests that many proteins are substrates for more than one protease.

We could not extend these approaches to identify substrates for FtsH, as its deletion is lethal due to the accumulation of LPS. However, when combined with a *fabZ* mutation[55], we could show that the degradation in nitrogen-starved batch culture continues when *ftsH* is deleted. The lack of significant overlap between proteins that are still degraded in the Δ*clpP*Δ*lon*Δ*hslV* strain and proteins pulled down from an FtsH-trap[63] also suggests that FtsH is likely not responsible for the remaining degradation. So far, we have not been able to generate viable quadruple knockout cells for all four known ATP-dependent proteases in *E. coli*. We, therefore, cannot completely rule out that FtsH is responsible for the remaining cytoplasmic degradation when the other three proteases are deleted. Regardless, a major protein degradation pathway in *E. coli* still needs to be discovered: either FtsH plays a much more significant role than generally anticipated, or there is an entirely different pathway outside the four known ATP-dependent proteases. While protein degradation itself is energetically favorable, unfolding a protein requires energy. *E. coli* encodes many non-ATP-dependent proteases[85], but the rapid turnover of proteins in the triple knockout line and the fact that most cytoplasmic proteins are structured suggest that some ATP-dependent unfoldase is involved. Perhaps, an adapter like ClpX or a chaperone unfolds proteins and allows those substrates to be degraded by one of the proteases believed to be energy-independent[85].

We found that many proteins have short half-lives regardless of nutrient limitations (Fig. 4). Among those, we see an over-representation of transcriptional regulators. Rapid turnover might enable a quick response to changing growth conditions to rapidly adjust transcription rates to the new environment. Surprisingly, we found no correlation between a protein's total half-life and its N-terminal residue suggesting that the N-end rule is a poor predictor for protein stability in vivo. In contrast, disordered proteins are drastically enriched among proteins with rapid turnover (Fig. 4F). This might suggest that many short-lived proteins could be degraded in an energy-independent way. This is further supported by our finding that the enrichment of disordered proteins is further increased among proteins that are degraded when the three ATP-dependent proteases ClpP, Lon, and HslV are deleted. We observe highly significant correlation among protein turnover for functionally related proteins like complexes and those expressed from operons. We found striking examples of this regulation in phosphonate metabolism and flagella but can currently only speculate about the underlying regulatory

mechanisms and functional importance. Further studies will be required to follow up on these intriguing observations.

When we compared protein turnover across *E. coli* growth rates, we found that rates of active protein degradation remain constant (Fig. 6). Based on this finding, the relative contribution of active degradation compared to dilution due to cell division must change as a function of growth rate. Therefore, relative protein levels of actively degrading proteins must change with differing cell growth rates, or cells must compensate by adjusting transcription and/or translation rates. Protein expression regulation combines gene-specific effects with such global parameter changes[16]. Our insight will help to improve genome-wide protein abundance regulation models and could help to better engineer gene expression circuits with desired properties.

The discoveries in this manuscript have been enabled by introducing a method to measure protein turnover. We chose to use heavy ammonium to label newly synthesized proteins to avoid the use of mutants and to boost the signal for short labeling times. The resulting MS1 and MS2 spectra are extremely complex, making standard quantification approaches challenging[42]. We have overcome these challenges by taking advantage of the outstanding ability of the complement reporter ion strategy (TMTproC) to distinguish signals from the chemical background[46]. The introduced methods are applicable widely beyond *E. coli*. Our ability to use comparatively cheap heavy isotope labels opens up the possibility of performing similar studies on larger animals, e.g., after $D_2O$ intake, which would be cost-prohibitive with heavy amino acid labeling[42,86]. Unlike many other cutting-edge multiplexed proteomics approaches, the applied technology is compatible with comparatively simple and widely distributed instrumentation, such as quadrupole-Orbitrap instruments, as we avoid the need for an additional gas-phase isolation step. The required analysis software is available on our lab's GitHub site (https://github.com/wuhrlab/TMTProC).

We have generated a broad resource of protein turnover rates in thirteen different growth conditions, each with biological replicates. The investigated conditions include varying cell cycle times from 40 min to 12 h, nitrogen-, carbon-, phosphorus-limitation, and various protease knockout strains. We expect this resource to allow researchers to complement their data sets with protein turnover information. Our finding that active degradation rates are typically constant regardless of division rates will allow researchers to extrapolate protein half-lives to arbitrary conditions. Our measurements of how protein turnover rates change in protease knockout strains will help refine protease-substrate relationships. Unlike studies relying on trap experiments or protein microarrays, we could start to deduce the redundant nature of these connections. We have shown the power of the provided resource by demonstrating cytoplasmic recycling in nitrogen limitation and by finding a scaling law for active protein degradation rates with varying cell cycle times. Thus, we advance protein turnover measurement technology, provide a resource for ~3.2k *E. coli* protein half-lives under various conditions, and provide fundamental insight into global protein expression regulation strategies.

## Methods

### Turnover measurements

*E. coli* strain NCM3722 was grown in continuous culture chemostats at 37 °C. The chemostat (Sixfors, HT) volume was 300 mL with oxygen and pH probes to monitor the culture. pH was maintained at 7.2 ± 0.1. 40 mM MOPS media (M2120, Teknova) was used with glucose (0.4% w/v, Sigma G8270), ammonium (9.5 mM $NH_4Cl$, Sigma A9434) and phosphate (1.32 mM $K_2HPO_4$, Sigma P3786) added separately. For C- and N-limiting media, glucose and ammonium concentrations were reduced by fivefold (0.08% and 1.9 mM, respectively). The P-limiting medium contains 0.132 mM $K_2HPO_4$. Cells were grown up to steady state in a media containing light ammonium for ~8–10 generations.

**Table 2 | The time point used for the sample collection for all doubling times analyzed in this study**

| Doubling time | Reactor type | Time points for sample collection (min) |
|---|---|---|
| 42 min | Batch | 0, 5, 12, 20, 30, 45, 60,175 |
| 3 h | Chemostat | 0, 45, 105, 186, 270, 366, 510, 729 |
| 6 h | Chemostat | 0, 107, 229, 376, 548, 735, 1024, 1612 |
| 12 h | Chemostat | 0, 173, 302, 444, 649, 873, 1230, 2166 |

This is equivalent to running the 6 h doubling time chemostat for ~2.5 days before switching the feed to a media containing heavy ammonium (CIL, NLM-107-10-10). Based on the $OD_{600}$, ~4–6 mL of effluent is collected after the switch for proteomics analysis. The $OD_{600}$ for the chemostats are as follows: P-lim OD ~ 0.71, N-lim OD ~0.51, C-lim OD ~0.68. The effluent is immediately frozen in liquid nitrogen.

For the minimal media no limitation batch cultures, NCM3722 cells were grown at 37 °C in 40 mM MOPS media with glucose (0.4%), ammonium (9.5 mM) and phosphate (1.32 mM) added separately. The cells are grown overnight in the medium containing light nitrogen and are diluted 500x in the fresh media containing light ammonium. The cells are grown up to an OD ~0.4 before diluting by 10x into medium containing heavy nitrogen. We used $^{15}$N-ammonium rather than labeled amino acids because of the higher signal we obtain with a low labeling fraction. For example, when 10% of the nutrient pool is labeled: If using heavy arginine, 90% of newly synthesized tryptic peptides ending in arginine are unlabeled. In contrast, when using heavy ammonium, a peptide having 15 nitrogens, only $(90\%)^{15} = 21\%$ will be unlabeled. Furthermore, we would have to work with auxotroph mutants when using amino acid labeling. Subsequently, samples of ~200 mg of protein per time point were collected. To maintain the cells in steady state, they are repeatedly diluted into fresh media after it reaches the OD ~0.4.

The time points for proteomics collection depend on the doubling time of the bacteria. Table 2 details the time point used for the sample collection for all doubling times analyzed in this study:

## Chloramphenicol translation inhibition assay
*E. coli* strain NCM3722 was grown in batch cultures at 37 °C. 40 mM MOPS media (M2120, Teknova) was used with glucose (0.4%, Sigma G8270), L-arginine (2.37 mM, Sigma A5006) and phosphate (1.32 mM $K_2HPO_4$, Sigma P3786) added separately. The doubling time was 2 h. As the cells reached exponential growth, OD ~ 0.3, translation inhibition drugs were added to arrest the protein synthesis. Chloramphenicol was added at 200 mg/mL. *E. coli* cells (OD: 0.3 4 mL culture gave ~200 mg of protein) were harvested by centrifuging at $5000 \times g$ for ~ 2 min. 10 samples were collected post the addition of drugs at [0, 10, 20, 40, 60, 80, 100, 120, 180, 240] min for the proteomic analysis.

## Batch starvation assay
An overnight culture of *E. coli* grown in Luria-Bertani (LB) broth was diluted 1:100 in minimal media (40 mM MOPS media (M2120, Teknova), 0.4% glucose (Sigma G8270), 9.5 mM ammonium chloride (Sigma A9434), and 1.32 mM potassium phosphate dibasic (Sigma P3786)). Cultures were shaken at 30 °C until an OD of ~0.5 was reached. The culture was collected and centrifuged at $5000 \times g$ for 10 min. The supernatant was discarded, and cells were washed three times with 40 mM MOPS, centrifuging between each wash. Cells were then resuspended in identical minimal media but lacking either glucose or ammonium chloride. The culture was then shaken at 30 °C and samples were taken at the following times (in minutes): 0, 151, 226, 407, 408, 944. The samples were prepared for proteomic analysis using the protocol described in Sample Preparation.

**Table 3 | Information for all the strains and plasmids**

| Name | Description | Source or reference |
|---|---|---|
| Strains | | |
| DY378 | W3110 λcI857 Δ(cro-bioA) | 56 |
| RLG986 | DY378 *fabZ*$_{L85P}$ *cdaR*-IG-*yaeH::cat*; Cam$^R$ | This study and[55,96] |
| RLG1017 | DY378 *fabZ*$_{L85P}$ *cdaR*-IG-*yaeH::cat* D*ftsH::kan*; Cam$^R$ Kan$^R$ | This study and[55,96] |
| Plasmids | | |
| pKD4 | Template plasmid containing a FRT-flanked kanamycin resistance cassette; Amp$^R$ Kan$^R$ | 97 |

## Strain construction
The Δ*clpP*, Δ*lon*, Δ*hslV* single mutants were generated by P1 transduction from the Keio collection[87] into *E. coli* strain NCM3722. The Δ*clpP*Δ*lon*Δ*hslV* triple knockout was provided by the Basan lab[88].

The Δ*ftsH* strain was generated as follows. First, *fabZ*$_{L85P}$ was moved into the l Red strain DY378 with *cadR*-IG-*yaeH::cat* by linkage transduction[89]. Transductants were selected for on LB supplemented with 20 mg mL$^{-1}$ chloramphenicol and screened for the *fabZ*$_{L85P}$ mutation by DNA sequencing (Genewiz, South Plainfield, NJ). The *ftsH* coding sequence in the *fabZ*$_{L85P}$ mutant was deleted and replaced with the kanamycin resistance cassette and flanking flippase recognition target sites from pKD4 using l Red-mediated recombination[56].

**Strains and plasmids.** The information for all the strains and plasmids is contained in Table 3.

**Primers.** The information for all the primers used is contained in Table 4.

## Sample preparation and data analysis for quantitative proteomics
Samples were mostly prepared as previously described[90]. Briefly, each sample containing ~200 mg of total protein was lyophilized to remove the water and then resuspended in 200 mL of lysis buffer containing 50 mM HEPES pH 7.2, 2% CTAB (hexadecyltrimethylammonium bromide), 6 M GuHCl (guanidine hydrochloride), and 5 mM DTT. Cells were lysed by sonication: 10 pulses, 30 s, at 60% amplitude and further heating the lysate at 60 °C for 20 min. Next, 200 mL of lysate from every condition was methanol-chloroform precipitated. Protein concentration was determined using the bicinchoninic acid (BCA) protein assay (Thermo Fisher). Samples were then diluted to 2 M GuHCl with 10 mM EPPS pH 8.5 and digested with 20 ng μL$^{-1}$ LysC (Wako) at room temperature overnight. Samples were further diluted to 0.5 M GuHCl with 10 mM EPPS pH 8.5 and digested with an additional 20 ng μL$^{-1}$ LysC and 10 ng μL$^{-1}$ sequencing-grade trypsin (Promega) at 37 °C for 16 h. The digested samples were dried using a vacuum evaporator at room temperature and taken up in 200 mM EPPS pH 8.0. The multiplexing TMTpro tags[46] were added at a mass ratio of 5:1 tag/peptide to ~40 mg of peptide per condition and allowed to react for 2 h at room temperature. The reaction was quenched with 1% hydroxylamine (30 min, RT). Samples from all conditions were combined into one tube, acidified with 5% phosphoric acid (pH < 2). The samples are then ultracentrifuged at $100,000 \times g$ at 4 °C for an hour to pellet undigested proteins. The supernatants were dried using a vacuum evaporator at room temperature to remove acetonitrile from the labeling step. Dry samples were taken up in HPLC grade water and subjected to medium pH reverse phase prefractionation. Samples are prefractionated with medium pH reverse-phase HPLC (Zorbax 300 Extend C18, 4.6 × 250 mm column, Agilent) with 10 mM ammonium bicarbonate, pH 8.0, using 5% acetonitrile for 17 min followed by an acetonitrile gradient from 5% to 30%. Each fraction was dried and resuspended in

**Table 4 | The information for all the primers used**

| Name | Sequence (5' – 3') |
|---|---|
| ftsH.Fwd | GCGCTAGAAATGTGTCGTGA |
| ftsH.Rev | GGATGGCTAAGGTCCAGTGA |
| KOftsHpKD4.Fwd | CGCTGTTTTTAACACAGTTGTAATAAGAGGTTAATCCCTTGAGTGACatgTGTGTAGGCTGGAGCTGCTTC |
| KOftsHpKD4.Rev | GTACAAATACAGTCATCTGATGCGGGAACttaCTTGTCGCCTAACTGCTCATGGGAATTAGCCATGGTCC |
| fabZ.Fwd | CTGAACCAGGCGTCTATTCC |
| fabZ.Rev | GACATGGGGTCCAACGATAC |

100 μL of HPLC water. Fractions were acidified to pH < 2 with phosphoric acid and desalted. The samples were resuspended in 1% formic acid to 1 mg mL$^{-1}$ and 1 mg of the total combined sample was analyzed with the TMTproC approach[46]. We used two strategies to ensure that the monoisotopic peak was isolated for each peptide. First, isotopic envelope fitting was used to shift the quad isolation window towards the predicted monoisotopic peak via a diagnostic routine provided by Thermo Fisher Scientific. We added an additional filtering step during data analysis that was then used to remove any erroneously isolated non-$M_0$-peaks once the identity of the peptide was determined. To this end, we simply compared the pseudo-monoisotopic peak mass and charge state with the used isolation window $m/z$ values. The difference between the isolation and monoisotopic $m/z$ will be -0 for a true $M_0$ isolation, and a multiple of $1/z$ if not.

Samples were analyzed on an EASY-nLC 1200 (Thermo Fisher Scientific) HPLC coupled to an Orbitrap Fusion Lumos mass spectrometer (Thermo Fisher Scientific) with Tune version 3.3. Peptides were separated on an Aurora Series emitter column (25 cm × 75 μm ID, 1.6 μm C18) (Ionopticks, Australia) and held at 60 °C during separation using an in-house built column oven, over 90 min for fractionated samples, applying nonlinear-acetonitrile gradients at a constant flow rate of 350 nL min$^{-1}$. MS parameter were set as previously described for TMTproC analysis[46]. Briefly, the mass spectrometer was operated to analyze positively charged ions in a data-dependent MS2-mode, recording centroid data with the RF lens level at 60% and the following settings for full scans: AGC target of 4E5 charges, maximum ion injection time of 50 ms, scan range $m/z$ 350–1400 with wide quadrupole isolation enabled, 120k Orbitrap™ resolution.

Following the survey scan, the following filters were applied for triggering MS2 scans. Ions with $z = 2$ were analyzed if their $m/z$-ratio was between 500 and 1074 and had an intensity greater than 1.9E5. Isolated masses were excluded for 60 s after triggering with a mass tolerance window of ±10 ppm, while also excluding isotopes and different charge states of the isolated species. AGC target was set to 7.5E4 charges and the maximum ion injection times was 123 ms. The Orbitrap™ resolution was 60k in normal mass range mode. The quadrupole was utilized for isolation with an isolation width of 0.4 Th, and ions were fragmented with 30% CID amplitude (10 ms activation time, activation Q of 0.25).

The data was analyzed using the Gygi Lab GFY software licensed from Harvard. The detailed description can be found in ref. 91. Thermo Fisher Scientific raw-files were converted to mzXML using ReAdW.exe (http://svn.code.sf.net/p/sashimi/code/). Assignment of MS2 spectra was performed using the SEQUEST[92] algorithm by searching the data against the combined reference proteomes for *E. coli* acquired from Uniprot on 08/2017 along with common contaminants such as human keratins and trypsin. The target-decoy strategy was used to construct a second database of reversed sequences that were used to estimate the false discovery rate on the peptide level[93]. SEQUEST searches were performed using a 20-ppm precursor ion tolerance with the requirement that both N- and C terminal peptide ends are consistent with the protease specificities of LysC and Trypsin. A peptide level MS2 spectral assignment false discovery rate of 1% was obtained by applying the target-decoy strategy with linear discriminant analysis[94]. Peptides were

assigned to proteins and a second filtering step to obtain a 1% FDR on the protein level was applied[95]. Peptides that matched multiple proteins were assigned to the proteins with the most unique peptides. For all methods, peptides were only considered quantified if the signal-to-FT noise ratio (S:N) across all channels was greater than forty.

Identification of complementary ion peaks, modeling of the isolation window, and deconvolution of the complementary peaks were performed as previously described[46]. Briefly, the complement reporter ion cluster was located, and the observed ratios were extracted. Using the measured shape of the isolation window and measured TMTpro isotopic impurities, the relative abundance and composition of each peak that was isolated from the precursor envelope was determined and used in the deconvolution algorithm.

### Statistics and data reproducibility
All protein turnover rate measurements were replicated for each strain and condition. After collecting replicates for the wild-type strain in carbon limitation at a 6-h doubling time, we determined that the measurements were highly reproducible and that a significant fraction of the proteome could be confidently assigned as degrading more than dilution at a $p$-value cutoff of 0.05 (Figs. 1E and 2B). The batch starvation assay was also initially done in duplicates. Due to the large sample of the technical replication (>1000 proteins quantified at a false discovery rate of 0.5%), we were able to confidently separate the decrease in protein abundance of the cytoplasmic and membrane-bound proteins with a $p$-value < 1E−15 (Fig. 2E). Growth assay comparison of the wild-type strain (NCM3722) and the triple protease knock-out on minimal media with thymidine as the sole nitrogen source were done in triplicate. All triplicates agreed well with each other, so no further replication was deemed to be necessary ($p$-value < 1E−4, Fig. 5D).

### Reporting summary
Further information on research design is available in the Nature Portfolio Reporting Summary linked to this article.

## Data availability
The mass spectrometry proteomics data have been deposited to the ProteomeXchange Consortium via the PRIDE partner repository with the dataset identifier PXD042444. Source data are provided with this paper.

## Code availability
The code used for analyzing these data sets can be found on our lab's Github at https://github.com/wuhrlab/ProteinTurnoverEcoli. The code is also citable using: https://doi.org/10.5281/zenodo.10895828.

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

## Acknowledgements

We would like to thank Markus Basan for the gift of the triple-protease knockout strain. We thank Michaela Eickhoff, Yihui Shen, Josh Rabinowitz, Jonathon O'Brien, Joe Sheehan, and Irina Mikheyeva-Bridges for their advice and discussions. This work was supported by NIH grants R35GM128813 (M.W.), R35GM118024 (T.J.S.), and grant T32-GM007388 (to Princeton University [E.M.H.]), the U.S. Department of Energy, Office of Science, Office of Biological and Environmental Research under award number DE-SC0018420 and DOE grant DE-SC0018260. This work was supported in part by the National Science Foundation through the Center for the Physics of Biological Function (PHY–1734030 [N.S.W.]). We gratefully acknowledge support by the American Heart Association predoctoral fellowship 20PRE35220061 (T.N.), Princeton Catalysis Initiative (M.W.), Eric and Wendy Schmidt Transformative Technology Fund (M.W.), Harold W. Dodds Fellowship (M.G.), NSF Graduate Research Fellowship (ERC), Princeton University's Summer Undergraduate Research Program (E.R.C.).

## Author contributions

M.G., A.N.T.J., T.J.S., N.S.W., Z.G., and M.W. designed the study. M.G., A.N.T.J., E.R.C., E.J.C., R.G., T.N., and M.S. performed experiments and data analysis. S.H.L, B.P.B, and E.M.H helped set up and troubleshoot experiments. T.J.S., N.S.W., Z.G., and M.W. supervised the study and provided funding. M.G., A.N.T.J., and M.W. wrote the manuscript with input from all co-authors.

## Competing interests

The authors declare no competing interests.
