## [Peer Review File · Nature Communications]

Global Protein Turnover Quantification in Escherichia coli Reveals Cytoplasmic Recycling under Nitrogen LimitationREVIEWER COMMENTS

Reviewer #1 (Remarks to the Author):

In this work, Gupta, Johnson and colleagues measured proteome-wide protein turnover in *E. coli*, by using an elegant approach: feeding cells with heavily labelled ammonia for different time points, labeling each time point with a different isobaric tag, and then isolating the monoisotopic peak for fragmentation and measuring the complement reporter ions in the MS2 (seeing a decrease with time, since the monoisotopic peak is reduced with time). The authors apply this approach to 12 different conditions of nutrient limitation. The authors find that upon nitrogen limitation a large proportion of cytoplasmic proteins is actively degraded. This prompts an attempt to identify the proteases responsible for the degradation of each protein by using their approach in knockout mutants of known proteases. This is a challenging task due to the redundancy (and the essentiality) of some of the proteases. The authors then analyze the characteristics of rapidly turning over proteins, how functionally-related proteins turnover and whether protein degradation rates scale with growth rate. The study is well designed and easy to follow, and the novel approach is likely to be useful for the study of protein turnover in general.

I have only minor comments:

1. For the analysis of the mass spectrometry data, and for an accurate quantification of protein half-lives, it seems to me that it is essential that only the monoisotopic peak (M0) is fragmented. Can the authors expand the methods section to clarify how this was achieved and ensured; either by settings on the MS or subsequent data analysis?
2. Can the authors clarify what is the concentration of heavily labeled ammonia in nitrogen-starvation conditions? Did the authors ensure that the doubling rate was the same before and after the switch to heavily labeled ammonia?
3. Apart from the N-terminal amino acid, did the authors look for other sequence characteristics/motifs that could explain how a specific protease recognizes its substrate? Are there known degron tags for these proteases?
4. Did the authors evaluate which proteins are upregulated in nitrogen starvation conditions? Are there hints of other proteases that are upregulated and could be responsible for the degradation of some of proteins?

André Mateus

Reviewer #2 (Remarks to the Author):

In their manuscript, Gupta et al. report a combination of mass spec approaches to obtain high quality data of the dynamics of incorporation of heavy nitrogen into *E. coli* proteins under multiple substrate-limited growth conditions in chemostat cultures.

Based on these data the authors calculate half-lives of individual proteins. The reported half-lives are calculated based on estimates of the proteins' first order degradation rate constants. The estimates are obtained by fitting a model of the dynamics of the monoisotopic peaks to the corresponding data of each protein's peptides.

Based on their analysis of the obtained half-lives, the authors report that under nitrogen limitation *E. coli* actively degrades proteins and recycles the amino acids obtained in this way, while this is not the case under carbon and phosphorus limitation. Furthermore, the authors determine substrates for several of *E. coli*'s ATP dependent proteases by comparing half-lives in different protease knockout experiments.

From their analysis, the authors conclude that the degradation mechanism that is responsible for the active degradation under nitrogen limitation needs further study and characterisation, since the protease responsible for this degradation cannot be unambiguously determined by their experiments.

This study is a beautiful example of how partnering wet lab experiments with theory (or modelling) can make quantities accessible which would not be accessible based on the measurements alone. Unfortunately, the multi-disciplinary approach of the work is not sufficiently reflected in the main text of the manuscript. There is no mention of the model-based nature of the approach in the summary and only very little of it in the rest of the main text. The model is mostly referred to as "theoretical dilution curve" which seems to imply that there is a fundamental theoretical ground truth on what this curve should look like, rather than this being the output of the model developed by the authors.

However, the interpretation of experimental data based on a model of the underlying biological process implies that all quantitative results are inseparably intertwined with the assumptions and uncertainties of the model. The quantity under study, the active degradation constant is not directly measured, but estimated by calibrating (fitting) a differential equation model cast by the authors to the experimental data. This would need to be made evident in the main text.

Since both provided documents start line enumeration with 1, I will refer to lines in the supplement and manuscript in formats S111 and 111, respectively. My mass spec expertise is rather limited and I was specifically asked to review the modelling part of the manuscript.

The model comprises the dynamics of the process of integration of heavy nitrogen into newly synthesized proteins, starting from the moment when in a chemostat culture, that had previously reached steady state, the feed of is switched from light to heavy ammonium. The model includes the recycling of amino acids from degraded proteins, a process that influences the dynamics of heavy nitrogen in the pool of amino acids available for protein synthesis. The dynamics of this pool are crucial to connect the model equations to the measured quantity: the dynamics of the normalised monoisotopic peak. All reactions are modelled with kinetics of maximal first order, which results in a system of linear ordinary differential equations that can be solved analytically. The quantity of interest is the active degradation (or recycling) rate constant in the model, which needs to be estimated by finding the value for this rate constant that minimised the discrepancy between model output and data.

I will start with my main points of criticism regarding the model first and then pick up on more general issues regarding consistency, clarity and completeness of the presentation of the model methodology.

I have doubts about the consistency of the model as pictured in the schematic between S160 and S161 and formalised in several differential equations distributed throughout the supplement. In particular, the reaction describing the recycling of nitrogen from proteins into glutamate only, seems to violate conservation of mass. If state variable NG is only the glutamate that is recycled from a degraded protein, mass conservation of nitrogen is not fulfilled in this reaction. This can be fixed by assuming that the rate constant $k_{d,bulk}$ implicitly contains a factor of a yield coefficient to account for the fact that proteins do not exclusively degrade into glutamates. But then I don't see how NG can be used to calculate the probability that a nitrogen in a newly synthesized peptide will be heavy one (S236), because NG only provides information on the ones that come in the form of glutamate.

Why did the authors choose to distinguish between glutamate and other amino acids? Why not have an all amino acids pool instead? What I noticed is that measured glutamate levels in cells are used to simplify the model (S256). I acknowledge, this would not be possible this if the state variable NG were representing the amount of nitrogen in all amino acids.

I think using a chemostat for this purpose is a very elegant approach. But I am wondering why the authors did not make more use of the classic chemostat model and its form of the equation for the dynamics of the limiting substrate in the vessel.

My second main issue with the model is the described assignment of values to parameters of the model to justify certain model simplification. This part is neither clear nor convincing to me. If I understand correctly, degradation characteristics observed for the different limitation scenarios are used to put reasonable numbers on certain model parameters. But, are those characteristics (like no active degradation/recycling in Plim and Clim) not deduced with the help of the model in the

first place? Or was the full model used first to infer those characteristics?

In general, the methods would need to be revised with respect to clarity regarding the reasoning behind the model simplification. For example, that the M0 profile is only sensitive to a certain range of values of parameter c is claimed (S257) but no reasoning or explanation supporting this claim is provided. How parameter "a" in nitrogen limitation can be calculated from the bar plots in figure 3G (S267) is not mentioned and completely unclear to me. Same for the reasoning behind the deduction of model simplification (S270).

While I understand the merit of assuming reasonable values and making simplifications when one is looking for underlying fundamental principles, I believe when it comes to analysing data to extract quantities with a direct biological meaning, like active degradation rate constants, the consequences of assumptions have to be assessed carefully. The authors advertise the estimated protein half-lives as a resource which can be used in future studies. But in fact, those numbers are not measurements, but estimates depending on the assumption of the model topology as well as the parameter values chosen to be able to simplify the model. Therefore, there needs to be some kind of assessment of the implications of those assumptions. Figure S. 3 shows the agreement between the full and the simplified model, but only for the case that the parameters in the full model take on the values assumed for simplification. It is also not clear to which parameter in the model nitrogen = 15 (S305) corresponds.

The presentation of the model derivation, simplification and parameter estimation would need substantial revising to improve clarity and completeness.

- The respective sections would profit from brief introductory paragraphs on what will be derived and the methodology or statistical test used.
- The presentation of the model would profit from presenting the differential equation system that comprises the model together with the underlying assumptions. Furthermore, a clear definition of the model observable is needed, that is how the model quantities connect to the measured quantities. Now the dynamics of the individual state variables are spread out over several pages and the connection between model and data are not stated explicitly.
- The meaning of all symbols used in the equations need to be explained immediately after the first use of the symbol.
- A symbolic distinction between the time dependent state variables of the model and the values they take on in steady state and at time 0 is needed. At the moment the same symbols for the time dependent state variables as for their steady state and initial values are used.
- Symbols and notation need to be made consistent between Figures, main text and different sections of the methods/supplement.
- References to the equations used to derive an equation seem to be off in some cases

Examples of undefined symbols and equations and other issues:

S156 V, D, F0

S178: dNV/dt misses „= 0“

S229 I guess the NG is the steady state value, needed as a function of the parameters, but I cannot find where this function is given

S234: complete equation undefined

S235: kT

S236: I

S274: NL, NT

More comments:

S298: KD must be kD

S320: KD must be kD

S300: equation (13) misses the I in the denominator

S338: $px8 \rightarrow ix8?$ since p has no numbers assigned to it

S306 To what model parameter does "nitrogen" correspond

S266: outside the vessel \rightarrow inside the vessel, outside the cell?

S247: How can this equation be derived from equation 9?

In general, the manuscript and supplement need to be revised in terms of consistency and clarification of terminology. After spending a considerable amount of time on it, I am still not sure what quantity is exactly referred to by the term half-life in the manuscript. Is it the half-life resulting from active degradation or half-life resulting from total protein turnover (active degradation + dilution)? The caption of Figure 1 says "The half-lives for each protein were calculated from the fits shown in D". Since it is the rate constant for active degradation which is derived by the fit, I would assume half-life means half-life as resulting from active degradation. But what is the meaning of the dilution limit in Fig. 1 panel E then in this context? How does dilution constrain the active degradation? Also in other parts of the manuscript half-life seems to refer to the half-life that results from total turnover. For example in the caption of Figure 2 (254) it says "Half-lives greater than doubling time indicate measurement noise." If the half-life is calculated from the active degradation rate constant, why should it not be greater than dilution? On the contrary, for stable proteins I would assume the half-life based on active degradation approaches infinity.

The manuscript switches between different terminology for the same quantity and same terminology for slightly different quantities: (active) degradation - turnover - replacement - protein decay. Distinction between turnover = (active) degradation + dilution and 'degradation' needs to be more consistent throughout the manuscript. Since here a distinction is made between turnover due to dilution and turnover due to (active) degradation, turnover and degradation should not be used interchangeably.

Comments on section: Protein half-life fitting

S310: Mode of what characteristic of the membrane peptides? Why is later the median used instead of the mode?

S319 What are correction values? What do we need to correct for? Cannot just be pipetting errors.

S327 What is the "a" vector obtained here used to calculate? No calculation using the vector provided.

Although the estimates are provided as 2 replicates, at least asymptotic confidence intervals should be provided for the fitted parameter to see how well the data constrain the fit.

Comments on section "Likelihood ratio calculations for the constant and the scaled model"

This section needs to be completely revised. Aim of the computation not stated. Notation is not consistent with Figure 6. It is not clear what the connection between the different sigmas mentioned there is. Are they all the same? Computations not clear either. RSS undefined.

Final result is the ratio between the likelihoods of the data under the alternative models, but what does the ratio of the likelihoods tell us? Shouldn't there be a significance level defined that the obtained ratio is compared to, to decide whether the Null hypothesis is rejected?

Collection of further comments regarding the main text:

For readers that are less familiar with the modelling of protein dynamics, referencing the exact pages in Uri Alon's book would make the content more accessible. I am also not sure that the term "response time" can be assumed to be general knowledge and used without explanation.

Line 83 and throughout manuscript: Should not be ammonium be correct rather than ammonia? I guess the authors are referring to the ion in solution and not the gas.

Figure 2C seems not to be referenced (should be line 157?) same for Fig. 2D

Line 159 "Due to measurement noise, these estimates are likely lower bounds of the true extent of protein degradation in N-lim." What's the reasoning behind this conclusion?

Figure 6: Notation for ratio r is not consistent between caption and Figure content

393 What makes separation between active degradation rates and dilution rates experimentally difficult for higher dilution rates?

Reviewer #3 (Remarks to the Author):

Review of the paper by Gupta et. Al.

Protein homeostasis or 'proteostasis' is crucial maintaining proper cellular functionality, and cytoplasmic density regulation. Coordination between protein homeostasis and cell division is an important but not yet properly resolved. Proteostasis is a poorly understood process, because precise quantification of protein turnover and determination of protein half-life is a technically challenging process.

In this present study, authors addressed this challenging question and measured proteome wide degradation rates in *E. coli*, by combining isotope labelling and complement reporter ion quantification. Authors precisely quantified half-life of more than 3000 proteins ranging 12 different growth conditions. In this study authors identified cytoplasmic proteins are recycled under nitrogen limitation growth condition, as most of the cellular nitrogen can be found in proteins. Further in this study authors identified cytoplasmic ATP dependent proteases are not involved in cytoplasmic protein degradation in nitrogen limitation conditions, which may indicate the presence of a yet unknown protein degradation pathway in *E. coli*. Surprisingly, authors observed protein degradation rates are independent of cell division rates, which open many interesting biological questions. In contrast with previous models, where protein homeostasis was mostly attributed to cellular growth and dilution, measurements in this paper suggests presence of active protein degradation and recycling in some conditions and for some proteins.

This study is of significant merit and backed up by well-designed experiments. Protein turnover measurements are of high quality and will serve as an important resource for researchers from various disciplines. Authors also took superior approaches to analyze complex mass spectrometry samples (M0 peak utilization).

Overall, we are very satisfied by the quality of work presented in this manuscript and we recommend this study for publication in Nature communication. We have a few minor comments that authors should address.

Minor comments:

- One of the very interesting findings that came out of this study is that half-life of certain protein has a growth rate dependence. For example, authors pointed out LpxC, is rapidly degraded in slower growth conditions. We suggest, authors should add a supplementary section, where proteins that shows growth rate dependent half-life change should be highlighted. Authors already have the data, which can either be presented as a heatmap, where a colormap represent the half-life and columns represent different growth rates, and each row represents a protein. Authors can also come up with any other way of representing the data. This will be useful as many crucial physiological processes are growth rate dependent in *E. coli*.

- Authors labeled samples taken at each timepoint with TMT pro isobaric tags and analyzed the samples by isolating pseudo-monoisotopic (M0) peak for quantification of complement reporter ions in the MS2. We agree this is a better approach while analyzing such complex systems, but authors should include a clear justification in support of this approach. Why this approach is superior to taking conventional approach should be described. In figure 1b, authors demarcated the M0 peak and its further decomposition in MS2 spectrum, but the real advantage of this approach should be mentioned more clearly in the main text.

- Identification of novel substrates for specific protease is one of the very interesting findings of this study. Some of these novel substrates are mentioned and pointed out in figure 3. Although a detailed table is provided for all identified proteins, it will be useful to add a supplementary figure focusing on all newly identified novel protease substrates.

Reviewer #4 (Remarks to the Author):

I co-reviewed this manuscript with one of the reviewers who provided the listed reports. This is part of the Nature Communications initiative to facilitate training in peer review and to provide appropriate recognition for Early Career Researchers who co-review manuscripts

Reviewer #5 (Remarks to the Author):

In this manuscript by Gupta, et al the authors use quantitative mass spectrometry to create a resource of protein lifetimes for a majority of cytoplasmic proteins in *E. coli* under different limiting growth conditions. Using mutant strains, they assign the degradation of some of these proteins to specific proteases, but show under nitrogen limiting conditions, protein degradation occurs independently of these proteases. Combining their measurements, they explore several global features of degradation in *E. coli*, such as exploring the importance of known destabilizing residues, that shorter lifetime proteins tend to be smaller and more disordered, and that proteins involved in the same pathways tend to have more similar lifetimes than unrelated proteins. Finally, they show that degradation rates are not dependent on cell division rates. Overall, this work provides a valuable resource for the *E. coli* community and highlights the impressive power of these types of proteomic approaches.

This is a resource paper rather than a hypothesis driven paper. Often, these types of resource papers are really two stories in one, with the first describing the method, publishing the resource, then showing aspects of that resource. Then, there is a 'deeper' dive for one novel aspect of that resource that is itself a complete other story. By doing so those keen on hypothesis driven science and the simplicity of a single story are satisfied as well as those interested in the resource itself. However, this can distract from the publication of the resource itself, which defeats the purpose of a database as described here. While I would not expect a complete additional story, a demonstration of the biological significance of some aspect of the results seems warranted.

For example, the shorter lifetime of ribosomal r-proteins during P-limitation is attributed to loss of rRNA followed by degradation of unassembled r-proteins. This model could be tested by directly downregulating rRNA and monitoring r-protein loss. Another example is to pursue their initial findings that proteins found in the same complex or same pathways tend to have similar lifetimes, which could be tested focused on a particular pathway using functional assays. Having such studies showcasing utility of their resource would provide more confidence in the value of the work. I want to emphasize that there does not need to be a complete story from these case studies, rather they demonstrate the physiological significance of the dataset within the context of presenting the resource.

The systematic testing of known proteases (ClpP, Lon, HslUV) shows how the lifetimes of many proteins can be assigned to some combination of these proteases. Again, some followup on these observations would add validation to the paper. For example, for the 'additive' degraded proteins such as IbpA, do these multiple pathways for degradation explain phenotypes in the single or double mutants?

Interestingly, the authors show that 40% of actively degrading proteins in N-lim conditions seem to still be degraded in the triple mutant, suggesting additional pathways of degradation during nitrogen starvation. A concern with this conclusion is that in the supplemental table S1 there is a half-life (3 hrs) shown for the triple mutant for the Lon protease (sp|P0A9M0|LON_ECOLI) for one of the replicates. Does this mean that there is some Lon protein derived signal in this triple mutant and thus, still some Lon activity? Given that generating this triple mutant is challenging because clpP and lon are adjacent on the genome, verifying this strain and result is critical, especially if there is still some Lon protease activity present.

Detailed comments:

1. tmRNA mediated trans-translation results in *ssrA*-tagging of a fraction of a large number of proteins. Because ClpXP and ClpAP degrade *ssrA*-tagged proteins, it would be worth investigating how much of the protein turnover observed could be assigned to this pathway by measuring protein turnover in strains lacking *ssrA* or *smpB*. If this is not possible, determining the overlap with the clpP targets and those known to be *ssrA*-tagged (such as that found in <https://pubmed.ncbi.nlm.nih.gov/11373298/>) would be helpful.

2. In the section describing the N-end rule substrates, reference 72 is cited as the source of Edman degradation data confirming these N-termini identify, but this reference is about *Xenopus* embryonic development? This needs to be checked.

3. Regarding N-end rule substrates described in Figure 4 E, based on the Table S4, there are only a handful of proteins identified with destabilizing N-termini. Using the label-free set from the authors, out of the 639 identified N-termini they identified, only five have destabilizing residues (leu/phe). The remainder of the destabilizing residues described in this figure appear to be from an older reference (Link, et al 1997), but even this dataset has only four primary destabilizing N-termini out of the 230 identified proteins. This makes sense because N-end rule substrates are likely present at very low levels due to their degradation, making them challenging to detect. It also seems likely that any N-end degrons unmasked by cleavage would likely be immediately degraded. Therefore, it is possible that the only proteins visible by these methods would be those that have circumvented the rapid degradation elicited by the N-end rule, making the lifetime of these proteins more the exception than the rule. Because of this and the small numbers of destabilizing N-termini used in their conclusion, it is suggested to remove the discussion of N-end rule substrates or describe these caveats in more detail.

4. Ensuring the knockout mutants are correct through resequencing or reconstructing the strains and retesting is critical, especially for the triple-knockout mutant.

5. Once published, this dataset of lifetimes will be incredibly valuable for the *E. coli* community. Ideally, the authors could work with databases such as EcoCyc or Microbesonline so that these datasets are integrated into other information hubs.

6. In the supplemental methods, chloramphenicol is stated as being used at 200 mg/ml. This is likely meant to be 200 micrograms /ml and should be corrected if so.

REVIEWER COMMENTS

Reviewer #1 (Remarks to the Author):

In this work, Gupta, Johnson and colleagues measured proteome-wide protein turnover in *E. coli*, by using an elegant approach: feeding cells with heavily labelled ammonia for different time points, labeling each time point with a different isobaric tag, and then isolating the monoisotopic peak for fragmentation and measuring the complement reporter ions in the MS2 (seeing a decrease with time, since the monoisotopic peak is reduced with time). The authors apply this approach to 12 different conditions of nutrient limitation. The authors find that upon nitrogen limitation a large proportion of cytoplasmic proteins is actively degraded. This prompts an attempt to identify the proteases responsible for the degradation of each protein by using their approach in knockout mutants of known proteases. This is a challenging task due to the redundancy (and the essentiality) of some of the proteases. The authors then analyze the characteristics of rapidly turning over proteins, how functionally-related proteins turnover and whether protein degradation rates scale with growth rate. The study is well designed and easy to follow, and the novel approach is likely to be useful for the study of protein turnover in general.

I have only minor comments:

1. For the analysis of the mass spectrometry data, and for an accurate quantification of protein half-lives, it seems to me that it is essential that only the monoisotopic peak (M_0) is fragmented. Can the authors expand the methods section to clarify how this was achieved and ensured; either by settings on the MS or subsequent data analysis?

We've added more detail to the supplemental methods detailing how we ensured that the monoisotopic peak was fragmented. Copied below for reference:

“We used two strategies to ensure that the monoisotopic peak was isolated for each peptide. First, isotopic envelope fitting was used to shift the quad isolation window towards the predicted monoisotopic peak via a diagnostic routine provided by Thermo Fisher Scientific. We added an additional filtering step during data analysis that was then used to remove any erroneously isolated non- M_0 -peaks once the identity of the peptide was determined. To this end, we simply compared the pseudo-monoisotopic peak mass and charge state with the isolation window m/z values. The difference between the isolation and monoisotopic m/z will be ~ 0 for a true M_0 isolation, and a multiple of $1/z$ if not. “

We would like to note that this diagnostic capability provided by Thermo will be included in the next instrument software update (Tune 4.1) and therefore broadly available to the scientific community. Our main contact at Thermo (Graeme McAlister) expects this software to be released in early 2024.

2. Can the authors clarify what is the concentration of heavily labeled ammonia in nitrogen-

starvation conditions? Did the authors ensure that the doubling rate was the same before and after the switch to heavily labeled ammonia?

We used a concentration of 1.9mM NH₄Cl in the feed for N-lim condition as opposed to 9.5mM NH₄Cl for the feed in P-lim and C-lim conditions. These details are mentioned in the supplement:

“MOPS media (M2120, Teknova) was used with glucose (0.4% w/v, Sigma G8270), ammonium (9.5 mM NH₄Cl, Sigma A9434) and phosphate (1.32 mM K₂HPO₄, Sigma P3786) added separately. For C- and N-limiting media, glucose and ammonium concentrations were reduced by 5-fold (0.08% and 1.9 mM, respectively). The P-limiting medium contains 0.132 mM K₂HPO₄.”

As the reviewer pointed out correctly, it is crucial that the steady state in the chemostat is maintained at all times. This is especially important for the nitrogen limited condition as the heavy and light media cannot be prepared in the same batch as is possible for the C-lim and P-lim. To ensure steady state in nitrogen limited conditions, we carefully measured the OD of the bacteria in the chemostat before and after switching to heavy media. Additionally, to ensure that the pump is delivering media at a steady rate, we carefully measured the amount of fluid collected over different ranges of time throughout the run. Representative readings from one of the runs (nitrogen limited doubling time of 6 hours) is shown below:

Time	OD ₆₀₀ (*4)	Doubling time = Ln(2)* volume in chemostat*time of effluent collection / volume of effluent collected
20 hours before switch	0.127	6.08 h
4 hours before switch	0.124	6.21 h
2 hours before switch Switch at time = 0	0.121	6.32 h
2 hours post switch	0.124	6.32 h
4 hours post switch	0.120	-
20 hours post switch	0.120	6.38 h
40 hours post switch	0.119	-

3. Apart from the N-terminal amino acid, did the authors look for other sequence characteristics/motifs that could explain how a specific protease recognizes its substrate? Are there known degron tags for these proteases?

Thanks for the suggestion. We indeed tried to find sequence patterns correlated with degradation rates. Here is the summary of the things we tried:

We ran Meme (a motif alignment and search tool) using the primary sequences for the rapid degraders as a positive set and non degraders as a negative set. But we could not find any highly significant motif. (E value <0.05).

In addition, the *ssrA* tagging system is a known mechanism for degrading polypeptides whose translation has stalled by *clpXP* and *clpAP* proteases. In response to reviewer 5, we investigated how much of the observed protein degradation could be attributed to the system, we knocked out the *smpB* gene (codes for the protein responsible for marking proteins for degradation) and measured gene-by-gene protein turnover in nitrogen limitation with a 6-hour doubling time in duplicate. We didn't see any significant change in protein turnover compared to the wild-type (scatterplot below).

Figure S10. Protein turnover rate measurements in a *smpB* knock-out strain. To investigate how much of the observed protein degradation in nitrogen limitation could be attributed to the SsrA tagging system, we knocked out *smpB* in the NCM3722 background strain used throughout this study. SmpB is part of the SsrA tagging complex. We measured gene-by-gene protein turnover in nitrogen limitation with a 6-hour doubling time in duplicate. 12 proteins significantly had a significantly longer total half-life in the mutant strain compared to the wild type (marked with a black x) : *nac*, *ynfM*, *gadE*, *ydhR*, *moeA*, *rpsT*, *pyrI*, *ybaK*, *mazF*, *csrA*, *truD*, *yajQ*, and *ppiC*. We did not observe any significant change in bulk cytoplasmic protein turnover compared to the wild type.

It's unclear to us why there isn't a more pronounced effect of removing the *ssrA*-tagging system. It's possible that the primary targets for the *ssrA*-tagging system, incomplete polypeptides in a stalled ribosome, make up a small portion of the overall proteome. Using stable isotope-labeling for measuring protein turnover, these rapidly degraded peptides would then produce a small effect on the incorporation of ^{15}N into the peptide backbone relative to the incorporation into properly translated proteins. Another possibility is that the ArfA-mediated pathway for releasing stalled ribosomes is completely redundant with the *ssrA* system (<https://doi.org/10.1111/j.1365-2958.2011.07607.x>). In the *smpB* knock-out strain, all degradation would be funneled through this pathway instead.

We've added the results from this experiment to the accompanying Supplementary Table of protein half-lives and the scatterplot and discussion above to Supplementary Figure 10.

4. Did the authors evaluate which proteins are upregulated in nitrogen starvation conditions? Are there hints of other proteases that are upregulated and could be responsible for the degradation of some of proteins?

We compiled the list of 74 proteases in *E.coli* obtained from the website (<https://www.uni-due.de/zmb/microbiology/>). This consists of all the different classes of proteases: metalloproteases, serine-proteases, aspartic-proteases and cysteine proteases. We compared the expression of proteases in nitrogen and phosphorus limited conditions using a volcano plot. Out of the 4 ATP dependent proteases, clpP, lon, hslV do not show a significant difference in expression (as highlighted in the plot). ftsH is highly significant, however the effect size is small. Highly significantly upregulated proteases in N-limitation as compared to P-lim are ddpX, dacD, which are marked on the plot. Since the stable proteins have to be unfolded to be degraded, we expect an ATP dependent protease to be a reasonable candidate for the degradation observed under nitrogen limitation. Unfortunately, we do not see any strong candidate being upregulated in the nitrogen limitation condition.

André Mateus

Reviewer #2 (Remarks to the Author):

In their manuscript, Gupta et al. report a combination of mass spec approaches to obtain high quality data of the dynamics of incorporation of heavy nitrogen into *E. coli* proteins under multiple substrate-limited growth conditions in chemostat cultures.

Based on these data the authors calculate half-lives of individual proteins. The reported half-lives are calculated based on estimates of the proteins' first order degradation rate constants. The estimates are obtained by fitting a model of the dynamics of the monoisotopic peaks to the

corresponding data of each protein's peptides.

Based on their analysis of the obtained half-lives, the authors report that under nitrogen limitation *E. coli* actively degrades proteins and recycles the amino acids obtained in this way, while this is not the case under carbon and phosphorus limitation.

Furthermore, the authors determine substrates for several of *E. coli*'s ATP dependent proteases by comparing half-lives in different protease knockout experiments.

From their analysis, the authors conclude that the degradation mechanism that is responsible for the active degradation under nitrogen limitation needs further study and characterisation, since the protease responsible for this degradation cannot be unambiguously determined by their experiments.

This study is a beautiful example of how partnering wet lab experiments with theory (or modelling) can make quantities accessible which would not be accessible based on the measurements alone. Unfortunately, the multi-disciplinary approach of the work is not sufficiently reflected in the main text of the manuscript. There is no mention of the model-based nature of the approach in the summary and only very little of it in the rest of the main text. The model is mostly referred to as "theoretical dilution curve" which seems to imply that there is a fundamental theoretical ground truth on what this curve should look like, rather than this being the output of the model developed by the authors. However, the interpretation of experimental data based on a model of the underlying biological process implies that all quantitative results are inseparably intertwined with the assumptions and uncertainties of the model. The quantity under study, the active degradation constant is not directly measured, but estimated by calibrating (fitting) a differential equation model cast by the authors to the experimental data. This would need to be made evident in the main text.

Response:

We thank the reviewer for pointing out the lack of clarity in our explanation. We have edited the main manuscript and included the statements below and pointed the readers to the supplementary section containing the derivation of the model.

Previously, we had the following statements. "Proteins without active degradation are expected to follow the theoretical dilution curve (dotted curve) based on the chemostat dilution rate. Fitting the measured signal of OmpF peptides with a model for the expected decay of the M_0 peak (solid curve) results in a turnover half-life similar to this expected value."

We have now included the following statements in the main text:

"We built a differential equation model for the dynamics of the peptide monoisotopic peak (M_0) as a function of protein active degradation rate (k_D), the number of nitrogens in the peptide (f), and the chemostat dilution rate (D) (Supplementary information). For each protein, the total turnover rate (k_D+D) is obtained by fitting the model to the experimentally measured decay of

the M_0 peaks. We also predict the dynamics of M_0 decay if the protein was infinitely stable ($k_D=0$) for a peptide with the median number of nitrogens, which we refer to as the dilution curve (dotted). Fitting k_D to the measured signal of OmpF peptides yields $k_D+D \sim D$ which indicates that this protein is not actively degrading.”

We agree with the reviewer that there is no theoretical ground truth and our estimates are based on the model assumptions, therefore, we have omitted the word “theoretical” in the main text and Figure 1.

Since both provided documents start line enumeration with 1, I will refer to lines in the supplement and manuscript in formats S111 and 111, respectively. My mass spec expertise is rather limited and I was specifically asked to review the modelling part of the manuscript.

The model comprises the dynamics of the process of integration of heavy nitrogen into newly synthesized proteins, starting from the moment when in a chemostat culture, that had previously reached steady state, the feed of is switched from light to heavy ammonium. The model includes the recycling of amino acids from degraded proteins, a process that influences the dynamics of heavy nitrogen in the pool of amino acids available for protein synthesis. The dynamics of this pool are crucial to connect the model equations to the measured quantity: the dynamics of the normalised monoisotopic peak. All reactions are modelled with kinetics of maximal first order, which results in a system of linear ordinary differential equations that can be solved analytically. The quantity of interest is the active degradation (or recycling) rate constant in the model, which needs to be estimated by finding the value for this rate constant that minimised the discrepancy between model output and data.

I will start with my main points of criticism regarding the model first and then pick up on more general issues regarding consistency, clarity and completeness of the presentation of the model methodology.

I have doubts about the consistency of the model as pictured in the schematic between S160 and S161 and formalised in several differential equations distributed throughout the supplement. In particular, the reaction describing the recycling of nitrogen from proteins into glutamate only, seems to violate conservation of mass. If state variable NG is only the glutamate that is recycled from a degraded protein, mass conservation of nitrogen is not fulfilled in this reaction. This can be fixed by assuming that the rate constant $k_{d,bulk}$ implicitly contains a factor of a yield coefficient to account for the fact that proteins do not exclusively degrade into glutamates. But then I don't see how NG can be used to calculate the probability that a nitrogen in a newly synthesized peptide will be heavy one (S236), because NG only provides information on the ones that come in the form of glutamate.

Why did the authors choose to distinguish between glutamate and other amino acids? Why not have an all amino acids pool instead? What I noticed is that measured glutamate levels in cells

are used to simplify the model (S256). I acknowledge, this would not be possible if the state variable NG were representing the amount of nitrogen in all amino acids.

Our understanding is that in *E.coli*, ammonia is first assimilated into glutamate before being transformed into the other amino acids ¹. The kinetics of isotope incorporation in tracer experiments are influenced by the rate of disappearance of the unlabeled metabolite and the pool size². At steady state, when the nitrogen fluxes are balanced, the labeling kinetics can be written down such that only the pool sizes are relevant, as derived in equations 8-12 in the newly revised detailed model. The intracellular concentration of glutamate is 96 mM, which is approximately 65% of all amino acids. Because the nitrogen in the other amino acids are derived from glutamate and their pool sizes are relatively small, they incorporate ¹⁵N rapidly after glutamate². Therefore, the kinetics of labeling of intracellular glutamate will largely determine label incorporation rates into proteins. This is the reason the schematic was drawn in such a way.

However, we agree that only showing glutamate is confusing and does not satisfy the mass balance. As pointed out correctly by the reviewer, $k_{d,bulk}$ is for all the amino acids. We have corrected the schematic and fully revised the annotation to indicate that we are referring to all amino acid pools and not just glutamate.

The relevant parameter that is affected by this change is the concentration of nitrogen is $c = \frac{N_P}{N_{AA}}$ $\sim 1000/147 \sim 6.7$. Determined using the literature values of pool sizes ³. The nitrogen present in the form of proteins (N_P) is ~ 1000 mM, and the pool size of nitrogen in the form of free amino acids N_{AA} is ~ 150 mM. $\Rightarrow 1000/150 = 6.67$. The range of c is $[1, \infty)$.

As demonstrated in the newly provided sensitivity analysis, M_0 is insensitive to the exact value of this parameter. Figure S3 shows various test cases that assess the model's sensitivity to different parameter values. Below we show one of the examples. As we see in the figure, while the N_{AA} changes drastically with different values of c (6, 60), its effect on M_0 (the quantity of interest) is practically insignificant.

A,C) Case 1: Model P-lim/C-lim for a peptide which is not actively degrading such that $\lambda = 0$, $f = 8$, and $D = 0.05 \text{ hr}^{-1}$, doubling time 6 hours.

2. I think using a chemostat for this purpose is a very elegant approach. But I am wondering why the authors did not make more use of the classic chemostat model and its form of the equation for the dynamics of the limiting substrate in the vessel.

The mass balance used in our equation is similar to that used for the classical chemostat equations, but the quantity of interest that is being modeled is modeled at different times in the chemostat (before steady-state, after steady state), hence the exact functional form is irrelevant. Below, we try to explain this in more detail with the equations.

Classical chemostat equation on limiting substrate: The equation below models the concentration of limiting substrate (C) as a function of time until the chemostat reaches a steady state. At steady state, $\frac{dC}{dt} = 0$.

If F is the constant volumetric flow rate, C_r is the concentration of limiting nutrient in the media, W is biomass, V is the volume of liquid in the chemostat vessel and C is the concentration of limiting substrate, S is the stoichiometric constant for the conversion of nutrient to biomass, D is the dilution rate:

$$\frac{dC}{dt} = \boxed{DCr} - \boxed{DC} - \boxed{S\mu \frac{W}{V}}$$

Inflow with the feed
Dilution of the chemostat vessel
Conversion of the substrate to the biomass

$$\frac{dW}{dt} = \mu W - DW$$

However, growth rate μ is not a constant and depends on the concentration of the limiting substrate. Monod law can relate growth rate to the limiting substrate concentration, $\mu = \mu_{max} \frac{C}{K_S + C}$.

Together, these equations can be used to solve the dynamics of the substrate concentration and the biomass before it reaches the steady state.

chemostat equations used in this study: In this study, our system is already at a steady state. Once it reaches the steady state, we switch the feed to introduce the heavy form of the nutrient into the system. Therefore, at all times our system is in a steady state. To model the dynamics of M_0 (the monoisotopic fraction of the peptide) over time, we need to model the dynamics of composition of light ammonium N_{VL} in the vessel,

Equation used in this study:

$$\frac{d(N_{VL})}{dt} = \boxed{N_{FL} F_0} - \boxed{DN_{VL}} - \boxed{k_1 N_{VL}}$$

Inflow with the feed=0
Dilution of the chemostat vessel
Conversion of the substrate to the biomass

Here, F_0 is volumetric flow rate into the chemostat, N_{FL} is Nitrogen in the form of ammonium in the feed and k_1 is the rate of consumption of nitrogen in the vessel by bacteria. The parameter, k_1 is constant and is estimated.

My second main issue with the model is the described assignment of values to parameters of the model to justify certain model simplification. This part is neither clear nor convincing to me.

In the paragraphs below, the reviewer is raising reasonable arguments regarding,

- the lack of clarity on how the parameters for detailed and simplified models were deduced.
- Lack of assessment of the consequences of the parameter assumptions to the quantity of interest (M_0)

- lack of clarity in delineating the rationale behind the simplification of the model.

We completely agree with the criticism and have now clearly defined the following:

- What are all the parameters in the model, what they represent, and how they were calculated. **(SECTION1 – see further below)**
- Next, we show that the dynamics of M_0 - the quantity of interest- is insensitive to the exact values of these parameters. **(SECTION2– see further below)**
- Next, we clearly present the rationale behind the insensitivity of the M_0 to the exact values of parameters. **(SECTION3– see further below)**
- Furthermore, we explain the rationale behind why the simplified model (with only one parameter K) is a reasonable approximation to the detailed model. **(SECTION4– see further below)**

The sections are detailed below in this document. These details have been now included in the revised supplement.

If I understand correctly, degradation characteristics observed for the different limitation scenarios are used to put reasonable numbers on certain model parameters. But, are those characteristics (like no active degradation/recycling in Plim and Clim) not deduced with the help of the model in the first place? Or was the full model used first to infer those characteristics?

SECTION 1 clearly describes what are all the parameters, what do they mean in the context of the model and how were they calculated. The term " $k_{d,bulk}/D$ " which represents the percentage recycling was indeed deduced from the simplified model. However, in SECTION 2 and SECTION3 we show that the exact values of the parameters are practically insignificant for the dynamics of M_0 - the quantity of interest. They are kept in the analysis to provide a reasonable guess for the parameter values.

In general, the methods would need to be revised with respect to clarity regarding the reasoning behind the model simplification. For example, that the M_0 profile is only sensitive to a certain range of values of parameter c is claimed (S257) but no reasoning or explanation supporting this claim is provided. How parameter "a" in nitrogen limitation can be calculated from the bar plots in figure 3G (S267) is not mentioned and completely unclear to me. Same for the reasoning behind the deduction of model simplification (S270).

The sensitivity analysis in SECTION2 and describes how the exact values of the parameters are practically insignificant for the dynamics of M_0 and SECTION3 provides the intuitive rationale behind it. Specifically, we have now deleted line S257 and replaced it with the sensitivity analysis. SECTION1 now describes clearly how the parameter "a" is calculated from figure 3G. SECTION4 describes the rationale behind model simplification.

While I understand the merit of assuming reasonable values and making simplifications when one is looking for underlying fundamental principles, I believe when it comes to analysing data to extract quantities with a direct biological meaning, like active degradation rate constants, the

consequences of assumptions have to be assessed carefully. The authors advertise the estimated protein half-lives as a resource which can be used in future studies. But in fact, those numbers are not measurements, but estimates depending on the assumption of the model topology as well as the parameter values chosen to be able to simplify the model.

Therefore, there needs to be some kind of assessment of the implications of those assumptions. Figure S. 3 shows the agreement between the full and the simplified model, but only for the case that the parameters in the full model take on the values assumed for simplification.

Since our sensitivity analysis (SECTION2) shows that the exact value of the parameters in the detailed model are practically irrelevant to the dynamics of M_0 . SECTION4 shows the agreement between the simplified model and the detailed model. Together they convey that the equivalence of the detailed and simplified model holds over a large range of parameter values.

It is also not clear to which parameter in the model nitrogen = 15 (S305) corresponds

We apologize for the lack of consistency in our notation, nitrogen = 15 corresponds to the quantity f in the model. This is a quantity which is constant and determines the number of nitrogen atoms in a peptide, completely determined by the composition of the peptide of interest. This has now been corrected.

SECTIONS BEGIN HERE:

What are all the parameters in the model, what they represent, and how they were calculated. (SECTION1)

The detailed model has three parameters (c , d , a) and two constants (f and D) and one unknown (k_d), which is to be determined. The constant D represents the dilution rate set in the chemostat. For a doubling time of 6 hours, $D = \log(2)/6$. The constant f represents the number of Nitrogens in the peptide for which M_0 is computed. k_d , an unknown represents the active rate of protein-turnover.

Parameter	Estimation
$c = N_p/N_{AA}$	Determined using the literature values of pool sizes ³ . The nitrogen present in the form of proteins (N_p) is ~ 1000 mM, and the pool size of nitrogen in the form of free amino acids N_{AA} is ~150 mM. => $1000/ 150 = 6.67$. The range of c is $[1, \infty)$
$d = N_v/N_{AA}$	For N-lim, this parameter is approximately 0 because all the nitrogen in the vessel is taken up by the bacteria.

	For the P-lim and C-lim conditions, we estimate this quantity as follows: The concentration of nitrogen in the feed for P-lim and C-lim chemostats is 9.5 mM and for N-lim chemostats is 1.9 mM. Under N-lim, the cells consume all the nitrogen provided and grow to approximately the same OD₆₀₀ as in P and C-lim. Therefore, we assume that the amount of nitrogen consumed by the bacteria in P-lim and C-lim compared to the total nitrogen in the vessel, are in the ratio of 1.9:9.5 i.e. $\frac{1.9}{9.5} = \frac{N_p + N_{AA}}{N_p + N_{AA} + N_V}$. Using this equation and the ratio of $c = N_p/N_{AA}$, d comes out to 28. The range of d is $[0, \infty)$
$a = (k_{d,bulk} + D)/D$	For P-lim and C-lim, since most of the proteins are stable, it is assumed that the recycling flux is negligible: $k_{d,bulk} = 0$, hence $a = 1$ For N-lim, Fig. 3G shows the percentage of active turnover per hour calculated by multiplying the molecular weight and its abundance to the active half-life calculated using the simplified model (detailed in the subsequent section). The exact formula is presented in the supplement under the section “Percentage of active proteome turnover per unit hour”. In N-lim, 4.5% of the proteome is actively turned over each hour. $a = (1 + 0.045 / (\log(2)/6)) = 1.89$ The range of a is $[1, \infty)$

Dynamics of M_0 - the quantity of interest- is insensitive to the exact values of these parameters. (SECTION2)

For each case, we plot the solutions for N_{AAL} , N_{AAL}^f , M_0 for the estimated values of c , d , and a along with the solutions when each parameter is changed by an order of magnitude. We excluded values which are outside the possible range for each parameter. Finally, we plot the solutions when all parameters are changed by an order of magnitude simultaneously.

Figure S3. Plots of N_{AAL} , N_{AAL}^f , M_0 with time for range of parameter values

A-C) Case 1: Model P-lim/C-lim such that $N_V \neq 0$, for a peptide which is not actively degrading such that $\rightarrow k_D=0$, $f = 8$, and $D = \log(2)/6$

D-F) Case 2: Model P-lim/C-lim such that $N_V \neq 0$, for a peptide which is actively degrading with the active half-life of 1 hour, such that $\rightarrow k_D = \log(2)/1$, $f = 8$, and $D = \log(2)/6$

G-I) Case 3: Model N-lim such that $N_V = 0$, for a peptide which is only diluting such that $\rightarrow k_D=0$, $f = 8$, and $D = \log(2)/6$

J-L) Case 4: Model N-lim such that $N_V = 0$, for a peptide which is actively degrading such that $\rightarrow k_D = \log(2)/1$, $f = 8$, and $D = \log(2)/6$

Rationale behind the insensitivity of the M_0 to the exact values of parameters. (SECTION3)

Fig. S3, shows the profile of three quantities, N_{AAL} , N_{AAL}^f , M_0 with respect to time for different ranges of c , d , a . These plots show that N_{AAL} is sensitive to the exact choice of parameter values. However, since the dynamics of N_{AAL} only affect M_0 through N_{AAL}^f , M_0 is insensitive to the exact parameter values.

For a shotgun proteomics experiment, tryptic peptides must have at least seven amino acids to be considered. Since tryptic peptides end in the amino acids R or K, each peptide will have a minimum of eight nitrogen atoms. As the number of nitrogen atoms in a peptide increases, the impact of N_{AAL} dynamics on M_0 decreases. Hence, the above analysis is conducted for peptides with eight nitrogen atoms. Even for this length, the M_0 dynamics are not impacted by the changes in parameter values.

Rationale behind why the simplified model (with only one parameter K) is a reasonable approximation to the detailed model. (SECTION4)

Given that the exact profile of N_{AAL} has little impact on M_0 , we propose that the dynamics of nitrogen available to translate protein can be simplified to an exponential with one parameter. This makes the integration analytical and simplifies the fitting procedure.

$$\frac{N_L}{N_T} = e^{-DKt}$$

where $K \in [1, \infty)$. K determines how fast the nitrogen available to make the proteins is exchanged. Figure S4 shows how M_0 is affected by different values of K .

Finally, solving the simplified gives us:

$$\frac{M_0}{PI} = \frac{(k_D + D) e^{-fDKt} - fDK e^{-(k_D + D)t}}{(k_D + D) - fDK} \quad (A5)$$

Solving equation A5 requires an estimate for $K \in [1, \infty)$, which describes how quickly nitrogen is exchanged within the cell. Our initial parameter estimates were $K=100$ for N-lim cells, since there is little free nitrogen in the system, and $K=1$ for P-lim and C-lim cells, indicating that most nitrogen is present outside the cells and is only exchanged by dilution out of the vessel and not by bacterial consumption. As seen in the Fig. S4 the M_0 profiles are robust to the exact choice of K , particularly when K is large.

Finally, Fig. S5 shows the equivalence of simplified and detailed model by comparing the M_0 profiles for all three conditions for different k_D . The simplified model is a reasonable approximation for the detailed model.

Figure S5. Demonstration of equivalence of the detailed and simplified model by comparing M_0 relative levels plotted with time. The number of nitrogens $f=8$ and $D = \log(2)/6$

- A) Dilution-only curve such that the active degradation rate $k_D=0$ for P-lim and C-lim.
- B) $k_D = \log(2)/1$ for P-lim and C-lim
- C) Dilution-only curve such that the active degradation rate $k_D=0$ for N-lim
- D) $k_D = \log(2)/1$ for N-lim

7. The presentation of the model derivation, simplification and parameter estimation would need substantial revising to improve clarity and completeness.

We apologize for the lack of clarity in the way the entire model is written. Most of the confusion indicated by the reviewer stems from the lack of clarity in the model presentation.

Revised model derivation is pasted here:

In this section, we will derive the dynamics of M_0 , the monoisotopic peak for a peptide with f nitrogen atoms, as a function of time after switching the feed from light to heavy ammonium.

The light peak (M_0) decays over time after we switch the media from $^{14}\text{NH}_4^+$ to $^{15}\text{NH}_4^+$. The mass balance on M_0 is given by:

$$\frac{dM_0}{dt} = \text{Rate of synthesis of } M_0 - \text{Rate of removal of } M_0$$

The rate of synthesis of M_0 is given by the total peptide synthesis rate k_T multiplied by the fraction of that peptide that contributes to the monoisotopic peak. Proteins derive their nitrogen from the amino acid pools N_{AA} , where the fraction of light nitrogen in the amino acid pool is given by $\frac{N_{AAL}}{N_{AA}}$. Since we are only considering the light peak M_0 , every nitrogen atom in the peptide) must be light. Hence, k_T is weighted by $\left(\frac{N_{AAL}}{N_{AA}}\right)^f$, where f is the number of nitrogens in the peptide. Finally, k_T is also multiplied by the probability I that all other non-nitrogen atoms in the peptide are light.

$$\text{Rate of synthesis of } M_0 = k_T \left(\frac{N_{AAL}}{N_{AA}}\right)^f I$$

We assume that the degradation of all peptides (including M_0) to follow a first-order decay, originating from the active degradation of the protein with the rate k_D and the dilution of the vessel at the rate D , due to the division of the *E. coli*.

$$\text{Rate of removal of } M_0 = k_D M_0 + D M_0 = (k_D + D) M_0$$

Putting together the two terms, we get,

$$\frac{dM_0}{dt} = k_T \left(\frac{N_{AAL}}{N_{AA}}\right)^f I - (k_D + D) M_0$$

We will now integrate M_0 using the integrating factor $e^{(k_D+D)t}$

$$\int_0^{M_0(t)} d(e^{(k_D+D)t} M_0) = \int_0^t k_T I \left(\frac{N_{AAL}}{N_{AA}}\right)^f e^{(k_D+D)t} dt \quad (1)$$

In equation 1, k_T is a constant that can be determined by writing the mass balance of the total amount of peptide P . Since the chemostat is at steady state, P does not change with time.

$$\frac{d(P)}{dt} = k_T - (k_D + D)P = 0$$

$$k_T = (k_D + D)P \quad (2)$$

At time $t=0$, when we assume there is no heavy nitrogen in the system, M_0 is determined by the natural isotopic abundance of each element.

$$M_0(t = 0) = PI \quad (3)$$

Using equations 1, 2, and 3 we can write the equation for M_0 as follows:

$$\frac{M_0}{PI} = e^{-(k_D+D)t} + (k_D+D) e^{-(k_D+D)t} \int_0^t \left(\frac{N_{AAL}}{N_{AA}}\right)^f e^{(k_D+D)t} dt \quad (4)$$

Next, we determine the time dependence of the fraction of light nitrogen in the amino acid pool $\frac{N_{AAL}}{N_{AA}}$. While the feed is instantaneously switched from $^{14}\text{NH}_4^+$ to $^{15}\text{NH}_4^+$, there is a delay for nitrogen in amino acids to be exchanged for ^{15}N . Therefore, to determine the composition of M_0 , we need to determine the dynamics of $\frac{N_{AAL}}{N_{AA}}$.

As presented in the schematic below, all nitrogen in the system is present in three forms: ammonium in the vessel (N_V), free amino acids in *E. coli* (N_{AA}), and proteins (N_P). Each of these nitrogen pools can be either ^{14}N (L) or ^{15}N (H). Ammonium from the feed (concentration N_F) replenishes ammonium in the vessel (N_V) at the volumetric flow rate of F_0 . The ammonium is assimilated by bacteria into amino acids with a specific conversion rate of k_1 . Amino acids are also replenished by recycling from degraded proteins at a degradation rate of $k_{d,bulk}$. Nitrogen from amino acids is translated to proteins at a rate k_2 . Each form of nitrogen gets depleted from the vessel at a dilution rate, D .

The mass balance of the light nitrogen in amino acids can be written as follows.

$$\frac{d(N_{AAL})}{dt} = k_1 N_{VL} + k_{d,bulk} N_{PL} - k_2 N_{AAL} - D N_{AAL} \quad (5)$$

The dynamics of N_{AAL} depend on N_{VL} and N_{PL} which are also both time-dependent. Hence, we need to determine their expressions to be able to calculate the dynamics of N_{AAL} . The mass balance for N_{VL} is:

$$\frac{d(N_{VL})}{dt} = N_{FL} F_0 - k_1 N_{VL} - D N_{VL}$$

Since the feed is instantaneously switched to heavy nitrogen, $N_{FL}(t = 0) = 0$. Therefore, the above equation can be simplified.

$$\frac{d(N_{VL})}{dt} = -(k_1 + D) N_{VL}$$

Integrating and using the boundary condition $N_{VL}(t = 0) = N_V$

$$N_{VL} = N_V e^{-(k_1+D)t} \quad (6)$$

Similarly, the mass balance of light nitrogen in proteins is:

$$\frac{d(N_{PL})}{dt} = k_2 N_{AAL} - k_{d,bulk} N_{PL} - D N_{PL} \quad (7)$$

Simultaneously solving equation 5, 6, 7 will give us the desired equation for N_{AAL} . But the above equations involve the undetermined rate parameters F_0 , k_1 , k_2 , and $k_{d,bulk}$. These parameters need to be determined to be able to solve the system.

At steady state, when nitrogen fluxes are balanced, the labeling kinetics can be written down such that only the pool sizes are relevant instead of the rate parameters F_0 , k_1 , k_2 , and $k_{d,bulk}$. It is easier to estimate the pool sizes instead of the rate parameters. Using the mass balance on total pools (heavy plus the light) of all the nitrogen forms, we will rewrite the equations 5, 6, 7.

$$\text{Let } N = N_{AA} + N_P + N_V.$$

At steady state, total balance on N gives us:

$$N_F F_0 = D N \quad (8)$$

At steady state, total balance on N_V gives us:

$$N_F F_0 = k_1 N_V + D N_V \quad (9)$$

Similarly, at steady state, total balance on N_P gives us:

$$k_2 N_{AA} = k_{d,bulk} N_P + D N_P \quad (10)$$

Let us define some parameters in the form of pool sizes:

$$c = N_P / N_{AA} ; d = N_V / N_{AA} ; a = (k_{d,bulk} + D) / D$$

Using equations 8, 9, 10 and the definition of c, d, and a:

$$k_1 = D(c+1)/d \quad (11)$$

$$k_2 = Dac \quad (12)$$

Using equations 5, 6, 11, and 12, the dynamics of N_{AAL} can now be written as:

$$\frac{d(N_{AAL})}{dt} = k_1 N_V e^{-(k_1+D)t} + D(a-1)N_{PL} - D(ac+1)N_{AAL} \quad (13)$$

Using equations 7, 11, and 12, the dynamics of N_{PL} can now be written as:

$$\frac{d(N_{PL})}{dt} = DacN_{AAL} - DaN_{PL} \quad (14)$$

Equations 13 and 14 form a coupled system of equations that is analytically solvable. The box below represents the details of solution for N_{AAL} .

Solving the coupled system of equations 13 and 14 to get the dynamics of N_{AAL}

$$\frac{d(N_{AAL})}{dt} = k_1 N_V e^{-(k_1+D)t} + D(a-1)N_{PL} - D(ac+1)N_{AAL} \quad (13)$$

$$\frac{d(N_{PL})}{dt} = DacN_{AAL} - DaN_{PL} \quad (14)$$

First solve the homogenous part of the equation. The matrix of interest is

$$\begin{pmatrix} -D(ac+1) & D(a-1) \\ Dac & -Da \end{pmatrix}$$

Eigen values and the corresponding eigen vectors are:

$$\lambda_1 = -D, v_1 = \begin{bmatrix} 1 \\ ac/(a-1) \end{bmatrix}; \lambda_2 = -Da(c+1), v_2 = \begin{bmatrix} 1 \\ -1 \end{bmatrix}$$

$$N_{AAL} = M_1 e^{-Da(c+1)t} + M_2 e^{-Dt} + M_3 e^{-(k_1+D)t}$$

$$N_{PL} = -M_1 e^{-Da(c+1)t} + \frac{M_2 a c e^{-Dt}}{a-1}$$

Using boundary conditions, $N_{AAL}(t=0) = N_{AA}$ and $N_{PL}(t=0) = N_P$, we can determine the values of M_1 and M_2 using the boundary conditions. M_3 can be determined using the method of undetermined coefficients.

Finally,

$$M_3 = \frac{N_{AA}(c+1)}{dac-c-1}; M_2 = \left(\frac{N_{AA}(c+1)(a-1)}{ac+a-1} \right) \left(\frac{dac-c-2}{dac-c-1} \right); M_1 = N_{AA} - M_3 - M_2$$

$$N_{AAL} = M_1 e^{-Da(c+1)t} + M_2 e^{-Dt} + M_3 e^{-(k_1+D)t} \quad (15)$$

Equations 4 and 15 together represent the coupled system we need to solve. In order to solve the above system of equations, we need to estimate the parameters:

$$c = N_P/N_{AA}; d = N_V/N_{AA}; a = (k_{d,bulk} + D)/D$$

The following table details how each parameter was initially estimated. A sensitivity analysis follows in the next section to determine the effect that errors in our estimates have on the dynamics of N_{AAL} and M_0 .

Parameter estimation for the detailed model

Parameter	Estimation
-----------	------------

$c = N_P/N_{AA}$	Determined using the literature values of pool sizes³. The nitrogen present in the form of proteins (N_P) is ~ 1000 mM, and the pool size of nitrogen in the form of free amino acids N_{AA} is ~150 mM. => 1000/ 150 = 6.67. The range of c is [1, ∞)
$d = N_V/N_{AA}$	For N-lim, this parameter is approximately 0 because all the nitrogen in the vessel is taken up by the bacteria. For the P-lim and C-lim conditions, we estimate this quantity as follows: The concentration of nitrogen in the feed for P-lim and C-lim chemostats is 9.5 mM and for N-lim chemostats is 1.9 mM. Under N-lim, the cells consume all the nitrogen provided and grow to approximately the same OD_{600} as in P and C-lim. Therefore, we assume that the amount of nitrogen consumed by the bacteria in P-lim and C-lim compared to the total nitrogen in the vessel, are in the ratio of 1.9:9.5 i.e., $\frac{1.9}{9.5} = \frac{N_P+N_{AA}}{N_P+N_{AA}+N_V}$. Using this equation and the ratio of $c = N_P/N_{AA}$, d comes out to 28. The range of d is [0, ∞)
$a = (k_{d,bulk} + D)/D$	For P-lim and C-lim, since most of the proteins are stable, it is assumed that the recycling flux is negligible: $k_{d,bulk} = 0$, hence $a = 1$ For N-lim, Fig. 3G shows the percentage of active turnover per hour calculated by multiplying the molecular weight and its abundance to the active half-life calculated using the simplified model (detailed in the subsequent section). The exact formula is presented in the supplement under the section "Percentage of active proteome turnover per unit hour". In N-lim, 4.5% of the proteome is actively turned over each hour. $a = (1 + 0.045 / (\log(2)/6)) = 1.89$ The range of a is [1, ∞)

Given these parameter estimates, under N-lim, since the pools of nitrogen outside the cell and inside the vessel are 0, $N_V \rightarrow 0$ and equation 13 simplifies to

$$\frac{d(N_{AAL})}{dt} = D(a-1)N_{PL} - D(ac+1)N_{AAL}$$

This simplifies the solution, equation 15, by eliminating the third term with M_3 .

$$N_{AAL} = M_1 e^{-Da(c+1)t} + M_2 e^{-Dt} \quad (16)$$

Therefore, the equation 15 represents the dynamics of N_{AAL} for nitrogen rich P-lim and C-lim conditions whereas the equation 16 represents the dynamic of N_{AAL} in N-lim conditions.

• The respective sections would profit from brief introductory paragraphs on what will be derived and the methodology or statistical test used.

Every section now starts with a detailed description of the goal of that section and what will that help us achieve. For example:

- When starting the derivation of M_0 , we have included the following paragraph.

“In this section, we will derive the dynamics of M_0 , the monoisotopic peak for a peptide as a function of time after switching the feed from light to heavy ammonium. The light peak (M_0) decays over time as we switch the media from 14NH_4^+ to 15NH_4^+ . For a peptide containing f number of Nitrogens, balance on the mono-isotopic peak (M_0) is given by: “

- On starting the derivation of N_{AAL} , we now include the following lines. The lines below indicate why it is necessary to derive N_{AAL}

“Time dependence of the fraction of light Nitrogen in the amino acid pool $\frac{N_{AAL}}{N_{AA}}$: In the chemostat the feed is instantaneously switched from 14NH_4^+ to 15NH_4^+ . However, the Nitrogen present in the form of amino acids that in turn makes the protein is only slowly getting exchanged with 15N . Therefore, to determine the composition of M_0 , we need to determine the dynamics of $\frac{N_{AAL}}{N_{AA}}$ ”

• The presentation of the model would profit from presenting the differential equation system that comprises the model together with the underlying assumptions. Furthermore, a clear definition of the model observable is needed, that is how the model quantities connect to the measured quantities. Now the dynamics of the individual state variables are spread out over several pages and the connection between model and data are not stated explicitly.

We have now included a section under the parameter fitting, that links the measured quantity to model parameters.

For a protein p with i peptides each containing f_i nitrogen atoms, we construct a theoretical matrix of $(i \times 8)$ values. Accordingly, the corresponding matrix of experimental values is created.

The k_D for each protein is obtained by minimizing the least square difference between the theoretical value (Eq. A5 with a free parameter $k_D + D$) and the experimentally observed values.

- The meaning of all symbols used in the equations need to be explained immediately after the first use of the symbol.

We have now included a table with variables and their description. The table is shown below. In addition, in the introductory paragraphs to every derivation we have clearly defined the description of variables used in that section. As an example, we are pasting the following paragraph here:

“As presented in the schematic below, all Nitrogen (N) in the system is present in three forms: ammonium in the vessel (N_V), amino acids that have been assimilated by *E. coli* (N_{AA}), and proteins (N_P). Ammonium from the feed (concentration N_F) replenishes ammonium in the vessel (N_V) at the volumetric flow rate of F_0 . The ammonium is assimilated by bacteria in the form of amino acids (N_{AA}) with a specific conversion rate of k_1 . Amino acids are also replenished by recycling from degraded proteins at a degradation rate of $k_{d,bulk}$. Nitrogen from amino acids enter the Nitrogen in proteins (N_P) at a rate k_2 . Each form of nitrogen gets depleted from the vessel at a dilution rate, D .”

Variable	Unit	Description
M_0	mol	Monoisotopic peak intensity, the MS peak where all the elements are light
f	Unitless	Number of nitrogen atoms in the peptide
N_{AA}	mol	Nitrogen in the amino acid pools
N_{AAL}	mol	Light nitrogen in the amino acid pool
P	mol	Total amount of peptide (sum of all MS peaks ($M_i, i = 0, \dots, \infty$) for the peptide)
k_T	mol hr ⁻¹	Rate of synthesis of total peptide P
I	Unitless	Probability of all the elements being light due to naturally occurring isotopic composition of each element.

k_D	hr^{-1}	Rate of active decay of peptide
D	hr^{-1}	Dilution rate out of the chemostat (F_0/V)
N	mol	Total amount of nitrogen in the vessel $N_V + N_{AA} + N_P$
N_V	mol	Nitrogen in the form of ammonium in the vessel
N_P	mol	Nitrogen in the form of proteins inside the bacteria
F_0	L hr^{-1}	Volumetric flow rate into the chemostat
N_F	mol L^{-1}	Nitrogen concentration in the form of ammonium in the feed
k_1	hr^{-1}	Rate of assimilation of ammonium by the bacteria in the form of amino acids (N_{AA})
k_2	hr^{-1}	Rate of conversion of nitrogen from amino acids into the nitrogen in proteins (N_P)
$k_{d,\text{bulk}}$	hr^{-1}	Rate of conversion of nitrogen in from proteins into amino acids
c	Unitless	N_P/N_{AA}
d	Unitless	N_V/N_{AA}
a	Unitless	$(k_{d,\text{bulk}} + D)/D$

- A symbolic distinction between the time dependent state variables of the model and the values they take on in steady state and at time 0 is needed. At the moment the same symbols for the time dependent state variables as for their steady state and initial values are used.

We agree with the confusion caused by the same notation used for state variables and their value at time 0. We have fixed this by specifying in the parentheses the ($t=0$) whenever the variable describes its specific value at time 0. We are pasting an example:

At time $t=0$, when we assume there is no heavy nitrogen in the system, M_0 is determined by the natural isotopic abundance of each element.

$$M_0 (t = 0) = PI \quad (3)$$

- Symbols and notation need to be made consistent between Figures, main text and different sections of the methods/supplement.

We have carefully checked this and made sure that symbols and notations are now consistent throughout the manuscript.

- References to the equations used to derive an equation seem to be off in some cases

We have carefully checked the equation numbers and their references used to derive more equations.

Examples of undefined symbols and equations and other issues:

We have included a table that describes the variables and their description. Moreover, these variables are also defined at their first occurrence in the text.

S156 V, D, F0:

In the modified form V is no longer needed,

F_0	Volumetric flow rate into the chemostat
D	Dilution rate out of the chemostat

S178: dNV/dt misses „= 0“. Thanks for pointing this out - this is fixed.

S229 I guess the NG is the steady state value, needed as a function of the parameters, but I cannot find where this function is given.

With the changes suggested by the reviewer, NG is now replaced with N_{AA} . The final set of coupled equation is:

$$\frac{M_0}{PI} = e^{-(k_D+D)t} + (k_D + D) e^{-(k_D+D)t} \int_0^t \left(\frac{N_{AAL}}{N_{AA}} \right)^f e^{(k_D+D)t} dt$$

$$M_3 = \frac{N_{AA}(c+1)}{dac-c-1}; M_2 = \left(\frac{N_{AA}(c+1)(a-1)}{ac+a-1}\right)\left(\frac{dac-c-2}{dac-c-1}\right); M_1 = N_{AA} - M_3 - M_2$$

$$N_{AAL} = M_1 e^{-Da(c+1)t} + M_2 e^{-Dt} + M_3 e^{-(k_1+D)t}$$

In these equations the relevant quantity of interest is $\frac{N_{AAL}}{N_{AA}}$, therefore we are not interested in the actual value of N_{AA} . All the coefficients M_1, M_2, M_3 that appear in the solution of N_{AAL} have N_{AA} as a multiple.

S234: complete equation undefined: We have now defined the variables associated with this equation. This equation derives the dynamics of composition of M_0 . The following paragraph is included in the supplement:

“We will derive the dynamics of M_0 , the monoisotopic peak for a peptide as a function of time after switching the feed from light to heavy ammonium. The light peak (M_0) decays over time as we switch the media from $^{14}\text{NH}_4^+$ to $^{15}\text{NH}_4^+$. For a peptide containing f number of Nitrogen, balance on the mono-isotopic peak (M_0) is given by:

$$\frac{d(M_0)}{dt} = \text{Rate of synthesis of } M_0 - \text{Rate of removal of } M_0$$

Term 1: Rate of synthesis of M_0 is the rate of synthesis of peptide multiplied by the fraction of that peptide that contributes to the monoisotopic peak. Since the proteins derive its Nitrogen from the amino acid pools (N_{AA}), the fraction of light Nitrogen in the amino acid pool is given by $\frac{N_{AAL}}{N_{AA}}$. Since we are considering the only light peak, every nitrogen (of the total f nitrogens) must be light, hence the relevant quantity is $\left(\frac{N_{AAL}}{N_{AA}}\right)^f$

Rate of synthesis of M_0 = Rate of synthesis of total peptide (P) * fraction of light peak
 = k_T * Probability of Nitrogens being light *
 Probability of elements being light acc. to composition in earth's crust

$$= k_T \left(\frac{N_{AAL}}{N_{AA}}\right)^f I$$

Term 2: Rate of removal of M_0 : We assume that the degradation follows a first-order decay. The peptide is actively decaying with the rate k_D and Diluting of the vessel at the rate D .

Rate of removal of M_0 = First order active decay of M_0 + Rate of dilution out of the vessel
 = $k_D M_0 + D M_0 = (k_D + D) M_0$ “

S235: k_T

k_T	Rate of synthesis of total peptide P
-------	--------------------------------------

S236: I

I	Probability of all the elements being light due to isotopic composition of each element in the earth's crust.
---	---

S274: NL, NT:

These are used to denote the light form and total Nitrogen (N) available to translate protein as shown in the snippet below.

Simplified model

Given that the exact profile of N_{AAL} has little impact on M_0 , we propose that the dynamics of nitrogen available (N) to translate protein can be simplified to an exponential with one parameter. This makes the integration analytical and simplifies the fitting procedure.

$$\frac{N_L}{N_T} = e^{-DKt}$$

where $K \in [1, \infty)$. K determines how fast the nitrogen available to make the proteins is exchanged.

More comments:

S298: KD must be k_D Thx, this now reads " $e^{(k_D+D)t}$ "

S320: KD must be k_D . Thx, this now reads " $k_D=0$ "

S300: equation (13) misses the I in the denominator.

Thx, this now reads"

$$\frac{M_0}{PI} = \frac{(k_D+D) e^{-fDKt} - fDK e^{-(k_D+D)t}}{(k_D+D) - fDK} \text{ „}$$

S338: px8 -> ix8? since p has no numbers assigned to it. Apologies for the typo, it is now fixed.

S306 To what model parameter does "nitrogen" correspond?

It should be f which represents the

f	Number of Nitrogen in the peptide.
---	------------------------------------

S266: outside the vessel -> inside the vessel, outside the cell? Thank you, this now reads:

“Since the pools of nitrogen outside the cell and inside the vessel”

S247: How can this equation be derived from equation 9?

We are pasting the rationale behind why and how the equation for the available nitrogen pools to make proteins can be simplified:

Given that the exact profile of N_{AAL} has little impact on M_0 , we propose that the dynamics of nitrogen available to translate protein can be simplified to an exponential with one parameter. This makes the integration analytical and simplifies the fitting procedure.

$$\frac{N_L}{N_T} = e^{-DKt}$$

where $K \in [1, \infty)$. K determines how fast the nitrogen available to make the proteins is exchanged.

Fig. S5 shows the equivalence of simplified and detailed model by comparing the M_0 profiles for all three conditions for different k_D . The simplified model is a reasonable approximation for the detailed model.

Figure S5. Demonstration of equivalence of the detailed and simplified model by comparing M_0 relative levels plotted with time. The number of nitrogens $f=8$ and $D= \log(2)/6$

- A) Dilution-only curve such that the active degradation rate $k_D=0$ for P-lim and C-lim.
- B) $k_D=\log(2)/1$ for P-lim and C-lim
- C) Dilution-only curve such that the active degradation rate $k_D=0$ for N-lim
- D) $k_D=\log(2)/1$ for N-lim

In general, the manuscript and supplement need to be revised in terms of consistency and clarification of terminology. After spending a considerable amount of time on it, I am still not sure what quantity is exactly referred to by the term half-life in the manuscript. Is it the half-life resulting from active degradation or half-life resulting from total protein turnover (active degradation + dilution)? The caption of Figure 1 says “The half-lives for each protein were calculated from the fits shown in D”. Since it is the rate constant for active degradation which is derived by the fit, I would assume half-life means half-life as resulting from active degradation. But what is the meaning of the dilution limit in Fig. 1 panel E then in this context? How does dilution constrain the active degradation? Also in other parts of the manuscript half-life seems to refer to the half-life that results from total turnover. For example in the caption of Figure 2 (254) its says “Half-lives greater than doubling time indicate measurement noise.” If the half-life is calculated from the active degradation rate constant, why should it not be greater than dilution? On the contrary, for stable proteins I would assume the half-life based on active degradation approaches infinity.

Half-life throughout the manuscript refers to the total half-life resulting from active protein degradation and dilution. We use the differential equation model to fit the total turnover rate from which the total half-life is calculated. We’ve updated all instances of the term “half-life” to “total half-life” to help with this confusion.

The manuscript switches between different terminology for the same quantity and same terminology for slightly different quantities: (active) degradation - turnover – replacement - protein decay. Distinction between turnover = (active) degradation + dilution and ‘degradation’ needs to be more consistent throughout the manuscript. Since here a distinction is made between turnover due to dilution and turnover due to (active) degradation, turnover and degradation should not be used interchangeably.

Thank you for pointing out this inconsistency. We’ve removed the few instances of “replacement” and “decay” from the manuscript to reduce confusion. We’ve also replaced all instances of “degradation rate” with “turnover rate” where we refer to the total contribution of active degradation and dilution.

Comments on section: Protein half-life fitting

S310: Mode of what characteristic of the membrane peptides? Why is later the median used instead of the mode?

We apologize for the lack of clarity in the description. We use the median of “the set of stable membrane peptides” to normalize the data. The median is simply more robust to measurement

noise than the mode. Below we've pasted the revised description of the entire normalization strategy and how we obtain the set of stable membrane peptides:

Normalization strategy.

The experimentally measured values of M_0 needs to be normalized for pipetting errors and for aligning deviations of the measured chemostat dilution times to desired dilution time (e.g. 5.7, 6.3, or 5.9 hours to 6.0 hours). We assume that the majority of membrane peptides (annotations from Uniprot) are stable in each chemostat condition, i.e., the median experimentally measured M_0 should align with the model predictions for no active turnover ($k_D = 0$). To ensure this, we use a set of "stable membrane peptides" to find a normalizing constant for each time point. The M_0 values for every protein are then divided by their corresponding constant to normalize the data.

To remove membrane peptides which are actively degrading, we applied the k-means clustering algorithm to cluster the peptides according to their M_0 decay profiles over time (Figure S6). The top panel shows k-means clustered membrane peptide profiles before normalization for an example P-lim condition. The numbers in the legend for each color represent the number of peptides in each cluster. Most peptides from membrane proteins fall in the purple cluster, which agrees well, but not perfectly, with the theoretical prediction for a protein with $k_D=0$. All membrane peptides except those that clustered into the most rapidly degrading profiles were used for normalization (in this example, the red cluster in the top plot in Fig. S6).

The correction values for each time point were obtained by minimizing the least square difference between the theoretical value (Eq. A5 with $k_D=0$) and the median values of the experimentally observed M_0 for all the g groups and 8 time points at once. Gradient descent (curve_fit function in python) was used to find the minima. |

Figure S6. Clusters of M_0 relative levels plotted with time. Clustering is done using k means algorithm. The dotted black curve is the dilution curve. The numbers in the legend correspond to the number of peptides in a cluster. Top) K means clustering before normalization for all the membrane peptides. Bottom) K means clustering after normalization for all the peptides containing 12 nitrogens.

S319 What are correction values? What do we need to correct for? Cannot just be pipetting errors.

The observed data for experimentally measured values of M_0 needs to be normalized for pipetting errors and for aligning deviations of the disparate chemostat dilution times to desired dilution time (for eg. 5.7h, 6.3h, 5.9 h to 6 hours). See the response to the comment above for a full explanation of the normalization strategy.

S327 What is the "a" vector obtained here used to calculate? No calculation using the vector provided.

The following sentences are now included in the supplement:

Vector \hat{a} contains the normalizing constants for each time point. The M_0 values for each time point of the entire dataset (experimentally measured values of M_0 for all peptides) are then divided by their corresponding constants to normalize the data.

Although the estimates are provided as 2 replicates, at least asymptotic confidence intervals should be provided for the fitted parameter to see how well the data constrain the fit.

Thank you for the suggestion. Supplementary table S7 contains 95% asymptotic confidence intervals for all the 13 conditions measured in this paper.

Comments on section “Likelihood ratio calculations for the constant and the scaled model”

This section needs to be completely revised. Aim of the computation not stated.

We have included the following statements.

“The goal of this section is to determine which of the models, constant vs scaled from Figure 6 fit the data better. To discriminate between the two models, we calculate the likelihood of observing the total $T_{1/2}$ in condition i given the model and its parameters.

Both the constant and the scaled model describe the relationship between $T_{1/2}$ total for bacteria doubling with 12 hours and a shorter doubling time. The index p denotes individual proteins.”

Notation is not consistent with Figure 6.

Sorry for not explaining the connection between the equations in Figure 6 and the supplementary section. This is now corrected. Below is an example snippet from revised supplement. We start out with the formula from the Figure and then work our way to the derivation of likelihood.

Scaled model:

From Fig. 6.

$$\frac{k_{\text{total},p,x=j}}{k_{\text{total},p,x=i}} = \frac{r_{i,j} k_{\text{dilution},i} + r_{i,j} k_{\text{active},p,i}}{k_{\text{dilution},i} + k_{\text{active},p,i}} = r_{i,j}$$

Let j be the condition where bacteria doubles with 12 hours:

$$\frac{k_{\text{total},p,x=12}}{k_{\text{total},p,x=i}} = r_{i,12}$$

Since $k_{\text{total}} = \ln 2 / T_{1/2}$

$$T_{1/2, p, i} = r_{i, 12} T_{1/2, p, 12}$$

We use regression-based approach to fit the above model to our data by assuming $T_{1/2, 12}$ as our independent variable.

$$T_{1/2, p, i} = r_{i, 12} T_{1/2, p, 12} + \epsilon_{\text{scaled}}$$

We assume that ϵ_{scaled} has a Gaussian distribution denoted by $N(0, \sigma_{\text{scaled}}^2)$. Therefore, we can write,

$$p(\epsilon_{\text{scaled}} | T_{1/2, p, i}, T_{1/2, p, 12}, \sigma_{\text{scaled}}) = N(0 | \mu_p = (T_{1/2, p, i} - r_{i, 12} T_{1/2, p, 12}), \sigma_{\text{scaled}}^2)$$

It is not clear what the connection between the different sigmas mentioned there is. Are they all the same?

There are two different sigmas, one for each model,

$$\sigma_{\text{constant}} \text{ and } \sigma_{\text{scaled}}$$

Each sigma represents the inherent standard deviation of the dependent variable from the mean. As in regression models, we assume homoscedasticity i.e. constant standard deviation over different values of the independent variable. We try to find the sigma's by maximizing the likelihood of each of the models with respect to sigmas.

Computations not clear either. RSS undefined.

Apologies for the lack of clarity. We have now included intermediate steps to complete the derivation and improve clarity in the computation.

We have provided.

1. The definition of likelihood for each of the models assuming standard deviation as a parameter.
2. A generalized derivation on how to estimate the standard deviation from maximizing the likelihood.
3. Using the estimated standard deviation to compute the likelihood for each model.

Below I am pasting a snippet from the supplement as an example.

Below we present a generalized derivation for estimating the value of σ for a regression-based modeling.

Estimating the parameter σ by maximizing the probability functions with respect to it (Maximum likelihood estimator):

If n is the total number of proteins, likelihood (L) is given by:

$$L = \prod_{p=1}^n p(0 | \mu_p, \sigma^2)$$
$$\text{Log}(L) = \frac{-n}{2} \log(2\pi) - n \log(\sigma) - \frac{1}{2\sigma^2} \sum_{i=1}^n \mu_p^2$$

Taking the derivative wrt to σ and equating to zero:

$$\hat{\sigma}^2 = \frac{\sum_{p=1}^n \mu_p^2}{n} = \frac{\text{RSS}}{n}$$

We have now clearly defined that the RSS stands for residual sum of squares, and the formula representing the RSS in the above snippet.

Final result is the ratio between the likelihoods of the data under the alternative models, but what does the ratio of the likelihoods tell us? Shouldn't there be a significance level defined that the obtained ratio is compared to, to decide whether the Null hypothesis is rejected?

We apologize for not providing the p-values. We have now conducted an F test to compare the variances from two models and using that the p-values for all the comparisons are highly significant (p -value $< 10^{-16}$). We have added the above statements and the calculations to the supplement and the caption of the main figure.

Collection of further comments regarding the main text:

1. For readers that are less familiar with the modelling of protein dynamics, referencing the exact pages in Uri Alon's book would make the content more accessible.

We have added the reference to Uri Alon's book with specific page numbers.

"Alon, U. An Introduction to Systems Biology: Design Principles of Biological Circuits pg. 18-21. (CRC Press, 2006)"

I am also not sure that the term "response time" can be assumed to be general knowledge and used without explanation.

Thank you for the comment. We've removed the term "response time" and more specifically state what we are referring to. Here is the relevant portion of the introduction for reference:

“As environments change, modification of degradation rates can rapidly adapt protein abundances to desired levels. Even if protein levels are modulated via transcription or translation, the time it takes for a protein to reach its new steady-state level is set by its turnover rate.”

2. Line 83 and throughout manuscript: Should not be ammonium be correct rather than ammonia? I guess the authors are referring to the ion in solution and not the gas.

Thanks for pointing this out. All the instances of ammonia are replaced with ammonium throughout the supplement and the main text.

3. Figure 2C seems not to be referenced (should be line 157?) same for Fig. 2D

Thanks for pointing this out. The reference is now correctly provided.

4. Line 159 “Due to measurement noise, these estimates are likely lower bounds of the true extent of protein degradation in N-lim.” What’s the reasoning behind this conclusion?

For proteins with little active degradation, the estimated quantity $k_D + D$ approaches D . Measurement noise is roughly constant across the range, making the discrimination between true active degradation and noise more difficult. For low sample sizes such as these, we expect that we are statistically underpowered and therefore many cytoplasmic proteins with increased degradation in nitrogen limitation are not detected. Hence, these estimates are likely lower bounds of the true extent of the number of actively degraded cytoplasmic proteins in nitrogen limitation.

We’ve amended this sentence with additional detail in the manuscript.

“Due to measurement noise and low sample sizes, we expect to be statistically underpowered and that these estimates are likely lower bounds of the true extent of protein degradation in N-lim.”

5. Figure 6: Notation for ratio r is not consistent between caption and Figure content

Thank you for catching this inconsistency. We’ve updated the figure caption to match the figure content.

6. 393 What makes separation between active degradation rates and dilution rates experimentally difficult for higher dilution rates?

Estimates for active protein degradation using stable isotope labeling are bounded by the time scales of amino acid incorporation (fast) and dilution due to growth (slow). When active degradation approaches either of these bounds, sensitivity to changes in active degradation

decreases and measurement noise begins to dominate. The closer these time scales are to one another (as is the case in exponential growth), the range of accurate degradation rate estimates decreases. In our experiments, for proteins that are moderately decaying (active half-life between the exponential phase growth rate and the chemostat growth rate), measurement noise hinders the determination of active degradation when cells are growing exponentially.

Reviewer #3 (Remarks to the Author):

Review of the paper by Gupta et. Al.

Protein homeostasis or 'proteostasis' is crucial maintaining proper cellular functionality, and cytoplasmic density regulation. Coordination between protein homeostasis and cell division is an important but not yet properly resolved. Proteostasis is a poorly understood process, because precise quantification of protein turnover and determination of protein half-life is a technically challenging process.

In this present study, authors addressed this challenging question and measured proteome wide degradation rates in *E. coli*, by combining isotope labelling and complement reporter ion quantification. Authors precisely quantified half-life of more than 3000 proteins ranging 12 different growth conditions. In this study authors identified cytoplasmic proteins are recycled under nitrogen limitation growth condition, as most of the cellular nitrogen can be found in proteins. Further in this study authors identified cytoplasmic ATP dependent proteases are not involved in cytoplasmic protein degradation in nitrogen limitation conditions, which may indicate the presence of a yet unknown protein degradation pathway in *E. coli*. Surprisingly, authors observed protein degradation rates are independent of cell division rates, which open many interesting biological questions. In contrast with previous models, where protein homeostasis was mostly attributed to cellular growth and dilution, measurements in this paper suggests presence of active protein degradation and recycling in some conditions and for some proteins.

This study is of significant merit and backed up by well-designed experiments. Protein turnover measurements are of high quality and will serve as an important resource for researchers from various disciplines. Authors also took superior approaches to analyze complex mass spectrometry samples (M0 peak utilization).

Overall, we are very satisfied by the quality of work presented in this manuscript and we recommend this study for publication in Nature communication. We have a few minor comments that authors should address.

Minor comments:

- One of the very interesting findings that came out of this study is that half-life of certain protein has a growth rate dependence. For example, authors pointed out LpxC, is rapidly degraded in slower growth conditions. We suggest, authors should add a supplementary section, where proteins that shows growth rate dependent half-life change should be highlighted. Authors already have the data, which can either be presented as a heatmap, where a colormap

represent the half-life and columns represent different growth rates, and each row represents a protein. Authors can also come up with any other way of representing the data. This will be useful as many crucial physiological processes are growth rate dependent in *E. coli*.

Thank you for the suggestion. While we found that the vast majority of proteins were actively degraded at a rate independent of growth rate, some were more rapidly degraded in fast or slow growth, which could be of interest to the community. We've added the following heat map to the supplementary figures to highlight the proteins whose half-life was dependent on growth rate.

Figure S8. Proteins whose active degradation was dependent on growth rate. We tested whether each protein's active degradation rate was correlated with growth rate using Pearson's

correlation coefficient. P-values were corrected for multiple hypothesis testing using the q-value package in R. Proteins shown in the heat map above met the significance threshold with a false-discovery rate of 0.01. Text values on each tile are the active degradation rate in units of hr^{-1} and are capped at 1. Proteins were excluded if they did not have a total half-life significantly less than the dilution-only half-life in at least one condition (T-test with a significance threshold of 0.05), or if they did not have an active degradation rate of more than 4% per hour (approximate bulk active degradation) in at least one condition.

- Authors labeled samples taken at each timepoint with TMT pro isobaric tags and analyzed the samples by isolating pseudo-monoisotopic (M_0) peak for quantification of complement reporter ions in the MS2. We agree this is a better approach while analyzing such complex systems, but authors should include a clear justification in support of this approach. Why this approach is superior to taking conventional approach should be described. In figure 1b, authors demarcated the M_0 peak and its further decomposition in MS2 spectrum, but the real advantage of this approach should be mentioned more clearly in the main text.

We're glad that you appreciated the approach we used to analyze these challenging samples. We've added significantly more detail to the section of the paper discussing the advantages of the approach to make this more clear for readers (copied below).

“With the knowledge of a peptide's chemical composition and the fraction of heavy isotopes over time, we can calculate the degradation rate of the corresponding protein. In practice, however, obtaining such measurements of isotopic envelopes in the MS1 spectrum is quite challenging, particularly at later time points when the isotopic envelopes spread out and overlap with those of other peptides. MS1-based label-free quantification relies on accurate assignment of peaks to peptide elution profiles, which is made more challenging by the complexity of spectra. Additionally, traditional search algorithms struggle to identify peptides with a significant fraction of heavy isotopes, so missing values between at later time points are a severe limitation of such approaches.⁴² To overcome these limitations, we labeled samples at each of the acquired eight time points with TMTpro isobaric tags and combined them for co-injection into the mass spectrometer.^{46,47} While MS1 spectra are still extremely complex using this approach, by combining isotope envelopes of peptides with and without ^{15}N , we ensure that the pseudo-monoisotopic peak (M_0) is always present for isolation and fragmentation in the MS2, increasing peptide identification rates and alleviating the missing value problem. Multiplexing experiments with highly complex MS1 spectra inevitably increase co-isolation of multiple peptides in a single MS2 spectra. Analyzing these extremely complex samples with standard low m/z reporter ion quantification would lead to severe ratio distortion and measurement artifacts.^{37,38,48} We overcame this limitation by isolating the pseudo-monoisotopic peak (M_0) for fragmentation and quantifying the balancer-peptide conjugates (complement reporter ions) that remain after reporter ions are cleaved off in the MS2 spectra. Complement ions (Fig. 1C) have peptide-dependent m/z ratios that are typically slightly different than co-isolated peptides. Therefore, using the complementary ions for quantification reduces ratio distortion effects compared to both MS2 and multi-notch MS3 reporter ion quantification.⁴⁶ (Fig. 1C)”

- Identification of novel substrates for specific protease is one of the very interesting findings of this study. Some of these novel substrates are mentioned and pointed out in figure 3. Although a detailed table is provided for all identified proteins, it will be useful to add a supplementary figure focusing on all newly identified novel protease substrates.

Thank you for the suggestion. We've added a new Supplementary Figure 7 to showcase all single protease substrates as well as the redundantly and additively degraded proteins.

ClpP only
 HslIV only
 Redundant

Lon only
 Additive
 Actively degrading in Triple KO

clpA	mdlB	sbcC	mukB	glpD	recA	pcnB	clpX	dps	cysN
ribB	ydhQ	intA	sixA	aspA	rutA	putA	dnaB	patA	fadE
dksA	mnmG	lldD	cyaA	nfo	ftsZ	leuA	dnaK	rpoC	znuC
yheS	oxyR	cysH	thiC	ppx	rpoS	rpoB	alaC	comR	exuR
uvrD	phoH	obgE	dnaE	fnr	cysD	mutS	otsA	hrpB	aroL
rlmN	rpsA	yebC	parC	def	yegW	yheO	era	uvrB	crl
yfcZ	ypfH	sufB	ispU	narP	topB	intF	tag	nemA	metR
ibaG	ydcl	priC	fhlA	rpoD	adhP	nadB	trxC	uhpA	yggX
kbp	yihD	ycaR	chaB	yaiA	mazF	ppiC	ibpA	helD	yajQ
ybaB	yieE	acpP	thiE	proQ	yiiQ	yaeP	iscR	rhlB	erpA
parE	yibA	yfhH	glnD	pspA	yjgA	grcA	yedW	minE	csiR
uvrY	srmB	thiL	astA	rarA	dnaA	aroK	nadA	treF	iscA
murQ	astE	yiaU	elaA	yihI	cobT	phnO	radA	ispH	radD
ybeZ	rsuA	yhgF	sufD	ydeP	mazG	yfcL	yecA	nanR	greA
thiF	argA	frr	hscA	hscB	iscS	flkB	recJ	metH	nfuA
cspE	hflX	pcm	ydiU	recD	chbB	ilvE	sdaA	ybcJ	relE
abgA	yjjU	ybhA	ugd	lipA	thiH	epmB	miaB	ydiF	ycgB
ybjl	yigl	dosP	arnA	priF	dinG	cysI	thrA	gadE	dnaQ
ftsK	gsiA	cfa	lpxC	iscU	ydbK	nudL	cysJ	leuC	ispG
astD	ubiC	birA	pflB	acnB	dnaX	dnaC	metAS	greB	thiG
clsB	gdhA	ilvB	iaaA	ruvC	yeaH	folX	cspC	guaD	serA
ydhS	leuD	ycfH	astB	bisC	rng	yegQ	nfsB	ahpF	lysA
asnA	tonB	mioC	aroD	ftsQ	tyrA	avtA	dedD	ackA	dadA
dapA	foIE	lexA	metK	trpB	yfcN	yaaA	fbp	aceA	trxB
yacC	aroG	bolA	ccmH	dedA	yehF	ftsI	ahpC	tabA	rnk
secG	ubiA	srkA	bioB	flk	aroE	selD	narW	fadB	gabT
pckA	garK	hypE	hmp	amiB	gcvT	tktA	wecB	nrdD	hemE
gph	mqo	dctR	bcsG	mdoB	azoR	uxaA	rplX	ydfZ	yocA
ynfC	ybiU	ybjX	sad	yegU	thiM	sufS	astC	prpR	yciV
csdA									

Figure S7. Protease substrate assignments. We assign each protease's contribution to active protein turnover of the rapidly degrading proteins, as in Figure 3. Each protein was then assigned to one of six categories as described in the section “Assigning substrates to proteases”.

Reviewer #4 (Remarks to the Author):

I co-reviewed this manuscript with one of the reviewers who provided the listed reports. This is part of the Nature Communications initiative to facilitate training in peer review and to provide appropriate recognition for Early Career Researchers who co-review manuscripts

Reviewer #5 (Remarks to the Author):

In this manuscript by Gupta, et al the authors use quantitative mass spectrometry to create a resource of protein lifetimes for a majority of cytoplasmic proteins in *E. coli* under different limiting growth conditions. Using mutant strains, they assign the degradation of some of these proteins to specific proteases, but show under nitrogen limiting conditions, protein degradation occurs independently of these proteases. Combining their measurements, they explore several global features of degradation in *E. coli*, such as exploring the importance of known destabilizing residues, that shorter lifetime proteins tend to be smaller and more disordered, and that proteins involved in the same pathways tend to have more similar lifetimes than unrelated proteins. Finally, they show that degradation rates are not dependent on cell division rates. Overall, this work provides a valuable resource for the *E. coli* community and highlights the impressive power of these types of proteomic approaches.

This is a resource paper rather than a hypothesis driven paper. Often, these types of resource papers are really two stories in one, with the first describing the method, publishing the resource, then showing aspects of that resource. Then, there is a 'deeper' dive for one novel aspect of that resource that is itself a complete other story. By doing so those keen on hypothesis driven science and the simplicity of a single story are satisfied as well as those interested in the resource itself. However, this can distract from the publication of the resource itself, which defeats the purpose of a database as described here. While I would not expect a complete additional story, a demonstration of the biological significance of some aspect of the results seems warranted.

For example, the shorter lifetime of ribosomal r-proteins during P-limitation is attributed to loss of rRNA followed by degradation of unassembled r-proteins. This model could be tested by directly downregulating rRNA and monitoring r-protein loss.

Thank you for the suggestion; this hypothesis is quite interesting. Unfortunately, downregulating rRNA is quite complex in *E. coli*. Ribosomal RNA is transcribed from 7 different operons, with a significant degree of overlap and redundancy and is subject to complex regulatory networks (see for example, Schneider, Ross, and Gross, "Control of rRNA expression in *Escherichia coli*", 2003, *Current Opinion in Microbiology*). The suggested experiment is therefore probably not feasible and/or would require unreasonable amounts of time and effort to try. Instead, we pursued two other avenues that you suggested below to further demonstrate the biological significance of our results.

Another example is to pursue their initial findings that proteins found in the same complex or same pathways tend to have similar lifetimes, which could be tested focused on a particular

pathway using functional assays. Having such studies showcasing utility of their resource would provide more confidence in the value of the work. I want to emphasize that there does not need to be a complete story from these case studies, rather they demonstrate the physiological significance of the dataset within the context of presenting the resource. The systematic testing of known proteases (ClpP, Lon, HslUV) shows how the lifetimes of many proteins can be assigned to some combination of these proteases. Again, some followup on these observations would add validation to the paper. For example, for the 'additive' degraded proteins such as IbpA, do these multiple pathways for degradation explain phenotypes in the single or double mutants?

Thank you for the suggestion to include examples of physiological significance of the results detailed in the manuscript. We've expanded the manuscripts with two findings that we think those keen on hypothesis-driven research will appreciate and further highlight the usefulness of the resource.

We were interested in validating two findings from the manuscript that you suggested: 1) that many proteins are additively/redundantly degraded and 2) that related genes (e.g. pathways) tend to have similar degradation rates. One pathway that demonstrates both facets is the *rut* pathway for pyrimidine degradation. The pathway is transcribed by a single operon of seven genes (*rutABCDEFG*) and has one adjacent transcriptional repressor, *rutR*. In our database, we detected five of the seven proteins plus the repressor. Of these five proteins, four are rapidly degraded in the wild-type strain and the single protease knock-outs (the fifth protein, *rutG*, is found in the membrane, which likely explains its stability). However, when we knock out *lon*, *clpP*, and *hslV* simultaneously, the four proteins are all significantly stabilized.

The paper that originally characterized the pathway showed that when the *rut* genes are transcriptionally upregulated, *E. coli* were able to grow on thymidine as the sole nitrogen source in minimal media at room temperature (PMID: 16540542). We reasoned that by removing their degradation, we could similarly increase protein concentrations and observe the same growth on thymidine. Indeed, the triple protease knock-out strain grows well on thymidine while the wild-type has difficulty growing.

We've added the following two panels to the manuscript along with discussion to showcase this finding.

B) The *rut* operon consists of seven genes and one repressor. We measured total half-lives for five of the seven genes, four of which were rapidly degraded in the wild-type strain and single protease knock-outs (*clpP*, *lon*, and *hsIV*). These four proteins were stabilized when the three proteases were knocked out simultaneously (TKO - triple knock-out). **C)** The wild-type and TKO strain were grown on glucose minimal media with thymidine as the sole nitrogen source. The mutant strain grows to a higher maximum OD600 than the wild-type strain.

The second demonstration of biological significance that we've added to the manuscript concerns the finding that cytoplasmic protein degradation increases when nitrogen is limiting. We reasoned that this significant investment of energy must be advantageous for the cell and that its downregulation would have a fitness cost.

Knocking out three of the four cytoplasmic, ATP-dependent proteases in *E. coli* significantly decreased cytoplasmic protein degradation in nitrogen limitation. While much degradation remains, we predicted that this downregulation might be enough to observe a fitness cost. As protein turnover is critical in adapting to new conditions, we hypothesized that if nitrogen-starved cells were given a new source of nutrients in the form of a nutrient upshift, protein turnover could provide amino acids for the translation of proteins needed in the new condition.

We measured growth curves following nutrient upshift from nitrogen starvation to LB media for both the wild-type strain and the mutant lacking *lon*, *clpP*, and *hslV*. We find that the triple knock-out is less able to adapt to the new conditions and has a significant growth delay (~80 minutes) compared to the wild-type. An important control in this experiment is the behavior of the cells when starved of carbon. We repeated the experiment with both strains, but this time the cells were shifted from carbon starvation to LB. Here, the growth of the triple protease knock-out matches the wild-type strain.

We've added the following panel and appropriate discussion to the manuscript highlighting this finding.

Figure S9. A triple protease knock-out strain has delayed growth when shifting from nitrogen starvation to rich media. We measured growth curves following nutrient upshift from starvation in minimal media to LB media for both the wild-type strain and the mutant lacking *lon*, *clpP*, and *hslV*. We find that the triple knock-out (TKO, brown) is less able to adapt to the new conditions and has a significant growth delay (~80 minutes) compared to the wild-type (blue) in

nitrogen limitation. When the cells were shifted from carbon starvation to LB, the growth of the triple protease knock-out matches the wild-type strain.

Interestingly, the authors show that 40% of actively degrading proteins in N-lim conditions seem to still be degraded in the triple mutant, suggesting additional pathways of degradation during nitrogen starvation. A concern with this conclusion is that in the supplemental table S1 there is a half-life (3 hrs) shown for the triple mutant for the Lon protease (sp|P0A9M0|LON_ECOLI) for one of the replicates. Does this mean that there is some Lon protein derived signal in this triple mutant and thus, still some Lon activity? Given that generating this triple mutant is challenging because *clpP* and *lon* are adjacent on the genome, verifying this strain and result is critical, especially if there is still some Lon protease activity present.

We are impressed by how careful this reviewer must have read the paper. We indeed were worried about this as well before we first submitted the manuscript. However, we convinced ourselves, and can hopefully also convince the reviewer, that this is due to one misidentified peptide for Lon in the knockout strain:

In all proteomics experiments, peptides are identified based on probabilistic models. We control the false discovery rate by using linear discriminant analysis to separate potential true peptides and known false peptides. We create a false peptide library by reversing the sequence of each protein in the .fasta file and searching each MS2 spectra against both forward and reverse sequences. Proteomics datasets are typically published with a 1% FDR (we also used this threshold in this study), meaning that approximately ~30 proteins per experiment are erroneously identified (typically these are proteins identified based on a single peptide). We are convinced that the strains were generated correctly, and that the Lon protein identification was part of the 1% false hits. This is based on the following evidence:

- 1) First, Lon is a high abundant protein, yet in the protease triple knock-out data sets we only detected one single peptide with low signal (6.2E7). In the corresponding wild-type experiments, we found 185 Lon peptides with a total signal of 2.9E10.
- 2) We PCR amplified an internal 2.3 kb region of the *lon* gene of the triple protease knock-out and the wild-type. The wild-type strain gave the expected ~2.3 kB band while the knock-out did not amplify. The blank lane does not include any cells.

Detailed comments:

1. tmRNA mediated trans-translation results in *ssrA*-tagging of a fraction of a large number of proteins. Because ClpXP and ClpAP degrade *ssrA*-tagged proteins, it would be worth investigating how much of the protein turnover observed could be assigned to this pathway by measuring protein turnover in strains lacking *ssrA* or *smpB*. If this is not possible, determining the overlap with the *clpP* targets and those known to be *ssrA*-tagged (such as that found in <https://pubmed.ncbi.nlm.nih.gov/11373298/>) would be helpful.

We knocked out the *smpB* gene in the NCM3722 background strain used throughout this study and measured gene-by-gene protein turnover in nitrogen limitation with a 6-hour doubling time in duplicate. We didn't see any large-scale change in protein turnover compared to the wild-type.

Figure S10. Protein turnover rate measurements in a *smpB* knock-out strain. To investigate how much of the observed protein degradation in nitrogen limitation could be attributed to the SsrA tagging system, we knocked out *smpB* in the NCM3722 background strain used throughout this study. SmpB is part of the SsrA tagging complex. We measured gene-by-gene protein turnover in nitrogen limitation with a 6-hour doubling time in duplicate. 12 proteins significantly had a significantly longer total half-life in the mutant strain compared to the wild type (marked with a black x) : *nac*, *ynfM*, *gadE*, *ydhR*, *moeA*, *rpsT*, *pyrI*, *ybaK*, *mazF*, *csrA*, *truD*, *yajQ*, and *ppiC*. We did not observe any large-scale change in bulk cytoplasmic protein turnover compared to the wild type.

It's unclear to us why there isn't a more pronounced effect of removing the *ssrA*-tagging system. It's possible that the primary targets for the *ssrA*-tagging system, incomplete polypeptides in a stalled ribosome, make up a small portion of the overall proteome. Using stable isotope-labeling for measuring protein turnover, these rapidly degraded peptides would then produce a small effect on the incorporation of ^{15}N into the peptide backbone relative to the incorporation into properly translated proteins.

We've added the results from this experiment to the accompanying Supplementary Table of protein half-lives and the scatterplot and discussion above to Supplementary Figure 10.

2. In the section describing the N-end rule substrates, reference 72 is cited as the source of

Edman degradation data confirming these N-termini identify, but this reference is about *Xenopus* embryonic development? This needs to be checked.

Apologies for the incorrect reference, we have now corrected it to:

Link, A.J., Robison, K., and Church, G.M. (1997). Comparing the predicted and observed properties of proteins encoded in the genome of *Escherichia coli* K-12. *Electrophoresis* 18, 1259-1313. 10.1002/elps.1150180807.

3. Regarding N-end rule substrates described in Figure 4 E, based on the Table S4, there are only a handful of proteins identified with destabilizing N-termini. Using the label-free set from the authors, out of the 639 identified N-termini they identified, only five have destabilizing residues (leu/phe). The remainder of the destabilizing residues described in this figure appear to be from an older reference (Link, et al 1997), but even this dataset has only four primary destabilizing N-termini out of the 230 identified proteins. This makes sense because N-end rule substrates are likely present at very low levels due to their degradation, making them challenging to detect. It also seems likely that any N-end degrons unmasked by cleavage would likely be immediately degraded. Therefore, it is possible that the only proteins visible by these methods would be those that have circumvented the rapid degradation elicited by the N-end rule, making the lifetime of these proteins more the exception than the rule. Because of this and the small numbers of destabilizing N-termini used in their conclusion, it is suggested to remove the discussion of N-end rule substrates or describe these caveats in more detail.

Thanks for the suggestion. We have updated the main text as follows:

“One obvious sequence feature to investigate is the N-end rule, which relates a protein's stability to its amino-terminal residue.⁴ Amino-terminal arginine, lysine, leucine, phenylalanine, tyrosine, tryptophan, and formylated N-terminal methionine (fMet) are believed to be destabilizing residues, whereas the other residues are believed to be stabilizing.^{4,5} First, we determined the in vivo N-terminal residue of ~600 proteins using a label-free proteomics data set (Table S4). MS2 spectra were searched with the Sequest algorithm⁶ considering all possible N-terminal tryptic subfragments for a protein. Encouragingly, when we compared a small subset of the identified N-termini with previous data in literature obtained using Edman Degradation,⁷ we found nearly perfect agreement (55/61 proteins)(Fig. 4D). Surprisingly, however, our rapidly degrading proteins showed no enrichment for the previously reported destabilizing N-terminal amino residues (Fig. 4E). Interestingly, few proteins detected in either dataset had destabilizing N-terminal residues. It is possible that proteins with destabilizing N-termini are immediately degraded and therefore difficult to detect. Proteins which had destabilizing residues exposed via cleavage would similarly be short-lived and low abundant. In this case, the main-determinant of protein half-life would be the rate at which destabilizing N-termini are exposed. Either way, our results suggest that the N-terminus of *E. coli* proteins is not the primary determinant of proteins' in vivo half-life.”

4. Ensuring the knockout mutants are correct through resequencing or reconstructing the strains and retesting is critical, especially for the triple-knockout mutant.

As noted above, we believe that the Lon peptide identified in the triple-knockout mutant to be a false hit. We've confirmed that the *lon* gene is correctly deleted by PCR amplifying the *lon* gene using internal primers and showing that the triple-knockout strain does not amplify this gene anywhere on the genome.

In addition to the PCR results from each strain, we can also use the proteomic data to validate the knockouts. The table below shows the number of peptides identified in each strain in nitrogen limited chemostats from each of the proteases as well as the total number of identified peptides. There are no peptides identified from knocked-out proteases.

	Wild-type NCM3722	$\Delta clpP$	Δlon	$\Delta hslV$	$\Delta clpP \Delta lon$ $\Delta hslV$
Total peptides	75,548	58,887	68,548	54,859	67,021
ClpP peptides	11	0	7	5	0
Lon peptides	185	159	0	108	1
HslV peptides	16	11	28	0	0

5. Once published, this dataset of lifetimes will be incredibly valuable for the *E. coli* community. Ideally, the authors could work with databases such as EcoCyc or Microbesonline so that these datasets are integrated into other information hubs.

Thank you for suggesting this. We've been in contact with Peter Karp at EcoCyc about integrating our resource into their database. We are cautiously optimistic that our measurements will soon be available via EcoCyc.

6. In the supplemental methods, chloramphenicol is stated as being used at 200 mg/ml. This is likely meant to be 200 micrograms /ml and should be corrected if so.

Thank you for correcting this error. We've updated the concentration to 200 ug/ml.

1. Reitzer, L. (2003). Nitrogen assimilation and global regulation in *Escherichia coli*. *Annu Rev Microbiol* 57, 155-176. 10.1146/annurev.micro.57.030502.090820.
2. Yuan, J., Fowler, W.U., Kimball, E., Lu, W., and Rabinowitz, J.D. (2006). Kinetic flux profiling of nitrogen assimilation in *Escherichia coli*. *Nat Chem Biol* 2, 529-530. 10.1038/nchembio816.
3. Bennett, B.D., Kimball, E.H., Gao, M., Osterhout, R., Van Dien, S.J., and Rabinowitz, J.D. (2009). Absolute metabolite concentrations and implied enzyme active site occupancy in *Escherichia coli*. *Nat Chem Biol* 5, 593-599. 10.1038/nchembio.186.
4. Tobias, J.W., Shrader, T.E., Rocap, G., and Varshavsky, A. (1991). The N-end rule in bacteria. *Science* 254, 1374-1377. 10.1126/science.1962196.
5. Piatkov, K.I., Vu, T.T., Hwang, C.S., and Varshavsky, A. (2015). Formyl-methionine as a degradation signal at the N-termini of bacterial proteins. *Microb Cell* 2, 376-393. 10.15698/mic2015.10.231.
6. Eng, J.K., McCormack, A.L., and Yates, J.R. (1994). AN APPROACH TO CORRELATE TANDEM MASS-SPECTRAL DATA OF PEPTIDES WITH AMINO-ACID-SEQUENCES IN A PROTEIN DATABASE. *J. Am. Soc. Mass Spectrom.* 5, 976-989. 10.1016/1044-0305(94)80016-2.
7. Bell, P., and Scheer, U. (1999). Developmental changes in RNA polymerase I and TATA box-binding protein during early *Xenopus* embryogenesis. *Exp Cell Res* 248, 122-135. 10.1006/excr.1999.4411.

REVIEWER COMMENTS

Reviewer #1 (Remarks to the Author):

The authors have addressed all of my comments.

Reviewer #2 (Remarks to the Author):

The authors have addressed all of my concerns and questions and revised the manuscript and supplement accordingly. I have only some minor comments and suggestions.

Only when I looked at the confidence intervals in TableS7, I realized that when "Model fitting resulted in half-lives much greater than dilution rate; total half-life was set to ceiling for that condition". If I understand correctly, those estimated half-lives were replaced in Table S1 by a value that is specific for the condition (the ceiling or I would call it a right-censored value). I am not sure whether this is mentioned somewhere explicitly. But even if so, I am wondering whether this could easily be overlooked by other people using your data for follow up analyses, i.e. that those values would be taken as they are, without noticing that they basically just mean that the protein is stable. I am not sure what could be a good measure to prevent this. Maybe an indicator vector in parallel to the half-lives indicating whether a half-live is "genuine" or a right censored value (we only know that the real value is larger than the recorded value)?

I am still struggling a bit with the notation which does not distinguish between time dependent functions (state variables, $x(t)$) and the value those functions assume in steady state. Especially for N_V , N_P and N_{AA} which are kind of introduced as state variables (functions of time) in the kinetic scheme in (206) and appear later as the values in steady state. And P in equation (2). Adding this information to the variable table would probably make this already a bit clearer.

269 Table 1 – Parameter estimation for the detailed model: Is it possible the value of 28 mentioned for the parameter d should be something like 30.67? That's what I calculate and what seems also to be used in the sensitivity analysis.

N_L and N_T in S307 seem to be undefined and not in the variable table, N_{VL} and N_{PL} seem also to be missing in the variable table, which might also be sorted alphabetically for the readers benefit.

If I understand correctly, the estimated parameter is the sum of the dilution and the active degradation rate constant, so the left hand side of the equation in S392 could be changed accordingly, to avoid confusion.

A mention on how the asymptotic confidence intervals for the half-lives were calculated would be nice.

I found the reference 10.1038/nchembio816 in the rebuttal very helpful and think it could be helpful for others to reference it in the Supplementary as well.

Thank you for providing the code to estimate the half-lives on github. I see the point of not putting large csv files in a repository, but storing them on google drive is suboptimal. I would suggest to add the csv files when creating a DOI for the code, e.g. on Zenodo. A more detailed description on which folders in the repository contain which analysis would be appreciated. A short readme file would do. Only having the experimental condition cryptically in the folder name is a bit little information.

I would also appreciate the code of other quantitative analysis parts of the manuscript being made available, The authors have addressed all of my concerns and questions and revised the manuscript and supplement accordingly. I have only some minor comments and suggestions.

Only when I looked at the confidence intervals in TableS7, I realized that when "Model fitting

resulted in half-lives much greater than dilution rate; total half-life was set to ceiling for that condition". If I understand correctly, those estimated half-lives were replaced in Table S1 by a value that is specific for the condition (the ceiling or I would call it a right-censored value). I am not sure whether this is mentioned somewhere explicitly. But even if so, I am wondering whether this could easily be overlooked by other people using your data for follow up analyses, i.e. that those values would be taken as they are, without noticing that they basically just mean that the protein is stable. I am not sure what could be a good measure to prevent this. Maybe an indicator vector in parallel to the half-lives indicating whether a half-life is "genuine" or a right censored value (we only know that the real value is larger than the recorded value)?

I am still struggling a bit with the notation which does not distinguish between time dependent functions (state variables, $x(t)$) and the value those functions assume in steady state (X^*). Especially for N_V , N_P and N_{AA} which are kind of introduced as state variables (functions of time) in the kinetic scheme in (206) and appear later as the values in steady state. And P in equation (2). Adding this information to the variable table would probably make this already a bit clearer.

269 Table 1 – Parameter estimation for the detailed model: Is it possible the value of 28 mentioned for the parameter d should be something like 30.67? That's what I calculate and what seems also to be used in the sensitivity analysis.

N_L and N_T in S307 seem to be undefined and not in the variable table, N_{VL} and N_{PL} seem also to be missing in the variable table, which might also be sorted alphabetically for the readers benefit.

If I understand correctly, the estimated parameter is the sum of the dilution and the active degradation rate constant, so the left hand side of the equation in S392 could be changed accordingly, to avoid confusion.

A mention on how the asymptotic confidence intervals for the half-lives were calculated would be nice.

I found the reference 10.1038/nchembio816 in the rebuttal very helpful and think it would be helpful for others to reference it in the Supplementary as well.

Thank you for providing the code to estimate the half-lives on github. I see the point of not putting large csv files in a repository, but storing them on google drive is suboptimal. I would suggest to add the csv files when creating a DOI for the code, e.g. on Zenodo. A more detailed description on which folders in the repository contain which analysis would be appreciated. Only having the experimental condition cryptically in the folder name is a bit little information.

I would also appreciate the code of other quantitative analysis parts of the manuscript, like the content of Figure 6, being made available.

Congratulations, this is really nice work!

Reviewer #3 (Remarks to the Author):

The authors addressed all our concerns. We recommend this paper for publication. It's a great work.

Reviewer #4 (Remarks to the Author):

I co-reviewed this manuscript with another reviewer and the review report is submitted by other reviewer.

Reviewer #5 (Remarks to the Author):

This revised manuscript by Gupta et al has largely addressed the concerns from the initial submission. My major comments were the need to experimentally test some new biological insight from the impressive proteome wide measurements and a concern about whether the Lon protease was still present in the triple mutant.

For the first, the authors have fully addressed any concerns with additional experiments demonstrating two separate new biological insights from their measurements and should be commended for their efforts. Both the thymidine utilization and the exit from starvation results add excellent perspectives and insights from their proteomic findings.

For the second, the authors make a case explaining that the single peptide corresponding to Lon is likely a false hit. Combined with the negative PCR, the authors have generally addressed this concern, although it is surprising that routine whole genome resequencing of this strain was not performed.

In addition to addressing the experimental concerns, the authors are thanked for their efforts to make the data resource and pipeline fully available to all those in the community.

Reviewer #2 (Remarks to the Author):

The authors have addressed all of my concerns and questions and revised the manuscript and supplement accordingly. I have only some minor comments and suggestions.

Only when I looked at the confidence intervals in TableS7, I realized that when “Model fitting resulted in half-lives much greater than dilution rate; total half-life was set to ceiling for that condition”. If I understand correctly, those estimated half-lives were replaced in Table S1 by a value that is specific for the condition (the ceiling or I would call it a right-censored value). I am not sure whether this is mentioned somewhere explicitly. But even if so, I am wondering whether this could easily be overlooked by other people using your data for follow up analyses, i.e. that those values would be taken as they are, without noticing that they basically just mean that the protein is stable. I am not sure what could be a good measure to prevent this. Maybe an indicator vector in parallel to the half-lives indicating whether a half-live is “genuine” or a right censored value (we only know that the real value is larger than the recorded value)?

Thank you for the suggestion. We have updated Table S1 to make it more clear when total half-lives were set to the ceiling. A note at the top of Table S1 notes “* **Protein total half-life was set to the ceiling for this dilution rate.**” The “*” symbol has then been added to the appropriate cells in Table S1.

A mention on how the asymptotic confidence intervals for the half-lives were calculated would be nice.

We have added the following statements in the supplement:

“ Curve- fit function in python along with returning the parameter estimate also returns the variance estimate of the fitted parameter. To get the one standard deviation error on the parameter, we take the square root of the estimated variance. This is the error with 67% confidence interval. To get the 95% CI, we should look up for the Z-table. However, this is only valid if the parameters follow a normal distribution, which is not the case for small sample size. This is where the t-distribution comes in handy. Therefore we evaluate, Percent point function (inverse of cdf) at a value of 0.975 given degrees of freedom (dof= Number of data points – number of estimated parameters = $i \times 8 - 1$) (Table S7).”

I am still struggling a bit with the notation which does not distinguish between time dependent functions (state variables, $x(t)$) and the value those functions assume in steady state. Especially for N_V , N_P and N_{AA} which are kind of introduced as state variables (functions of time) in the kinetic scheme in (206) and appear later as the values in steady state. And P in equation (2). Adding this information to the variable table would probably make this already a bit clearer.

In the table pasted below, we have added a new column that indicates what all variable names represent a constant steady state or time-dependent variable.

N_L and N_T in S307 seem to be undefined and not in the variable table, N_VL and N_PL seem also to be missing in the variable table, which might also be sorted alphabetically for the readers benefit.

We thank the reviewer for pointing out these mistakes. We have now included all the four missing variables in the table below. Additionally, we noticed another one (K) was missing, which was now added. We have also sorted the table to facilitate better reading.

Variable	Unit	Constant or time dependent?	Description
D	hr ⁻¹	Constant	Dilution rate out of the chemostat (F_0/V)
F_0	L hr ⁻¹	Constant	Volumetric flow rate into the chemostat
I	Unitless	Constant	Probability of all the elements being light due to naturally occurring isotopic composition of each element.
k_1	hr ⁻¹	Constant	Rate of assimilation of ammonium by the bacteria in the form of amino acids (N_{AA})
k_2	hr ⁻¹	Constant	Rate of conversion of nitrogen from amino acids into the nitrogen in proteins (N_p)
$k_{d,bulk}$	hr ⁻¹	Constant	Rate of conversion of nitrogen in from proteins into amino acids
k_D	hr ⁻¹	Constant for a peptide	Rate of active decay of peptide
k_T	mol hr ⁻¹	Constant	Rate of synthesis of total peptide P
M_0	mol	Time dependent	Monoisotopic peak intensity, the MS peak where all the elements are light
N_{AA}	mol	Constant	Nitrogen in the amino acid pools
N_{AAL}	mol	Time dependent	Light nitrogen in the amino acid pool
N_F	mol L ⁻¹	Constant	Nitrogen concentration in the form of ammonium in the feed

N_{FL}	mol L ⁻¹	Time dependent	Light nitrogen concentration in the form of ammonium in the feed
N_p	mol	Constant	Nitrogen in the form of proteins inside the bacteria
N_{pL}	mol	Time dependent	Light nitrogen in proteins inside bacteria
N_V	mol	Constant	Nitrogen in the form of ammonium in the vessel
N_{VL}	mol	Time dependent	Light nitrogen in the form of ammonium in the vessel
N	mol	Constant	Total amount of nitrogen in the vessel $N_V + N_{AA} + N_p$
P	mol	Constant	Total amount of peptide (sum of all MS peaks ($M_i, i = 0, \dots, \infty$) for the peptide)
a	Unitless	Constant	$(k_{d,bulk} + D)/D$
c	Unitless	Constant	N_p/N_{AA}
d	Unitless	Constant	N_V/N_{AA}
f	Unitless	Constant	Number of nitrogen atoms in the peptide
Simplified Model Parameters			
N_T	mol	Constant	Nitrogen available to translate protein (from simplified model)

N_L	mol	Time dependent	Light nitrogen available to translate protein (from simplified model)
K	Unitless	Constant	K determines how fast the nitrogen available to make the proteins is exchanged. (A parameter in the simplified model)

269 Table 1 – Parameter estimation for the detailed model: Is it possible the value of 28 mentioned for the parameter d should be something like 30.67? That's what I calculate and what seems also to be used in the sensitivity analysis.

We apologize for the typo and commend the reviewer for their attention to detail. Thanks for pointing this out. We have now corrected the value to 30.67

If I understand correctly, the estimated parameter is the sum of the dilution and the active degradation rate constant, so the left hand side of the equation in S392 could be changed accordingly, to avoid confusion.

This is now corrected, and it reads:

$$\widehat{k_{D,p} + D} \triangleq \arg \min_{k_{D,p}} \sum_I \sum_{t=0}^7 (\overline{M}_{0,\text{theoretical}, p, i, t}(k_{D,p}) - \overline{M}_{0,\text{experimental}, p, i, t})^2$$

I found the reference 10.1038/nchembio816 in the rebuttal very helpful and think it could be helpful for others to reference it in the Supplementary as well.

This reference is added.

Thank you for providing the code to estimate the half-lives on github. I see the point of not putting large csv files in a repository, but storing them on google drive is suboptimal. I would suggest adding the csv files when creating a DOI for the code, e.g. on Zenodo. A more detailed description on which folders in the repository contain which analysis would be appreciated. A short readme file would do. Only having the experimental condition cryptically in the folder name is a bit little information.

Thanks for the suggestion. The files are now uploaded to Zenodo. Here is the link:

<https://zenodo.org/records/10672712>

The github also contains a table with each of the file names corresponding to a condition.

I would also appreciate the code of other quantitative analysis parts of the manuscript, like the content of Figure 6, being made available.

Figure 3 and Figure 6 in the paper contain additional quantitative analysis. This code to reproduce Figure 3 and Figure 6 is now added to the GitHub repository.

REVIEWERS' COMMENTS

Reviewer #2 (Remarks to the Author):

The authors have addressed all of my comments.